# Graphical Abstract

## **CLAQC v1.0 – Country Level Air Quality Calculator. An empirical modeling approach.**

Stefania Renna,Francesco Granella,Lara Aleluia Reis,Paulina Schulz-Antipa

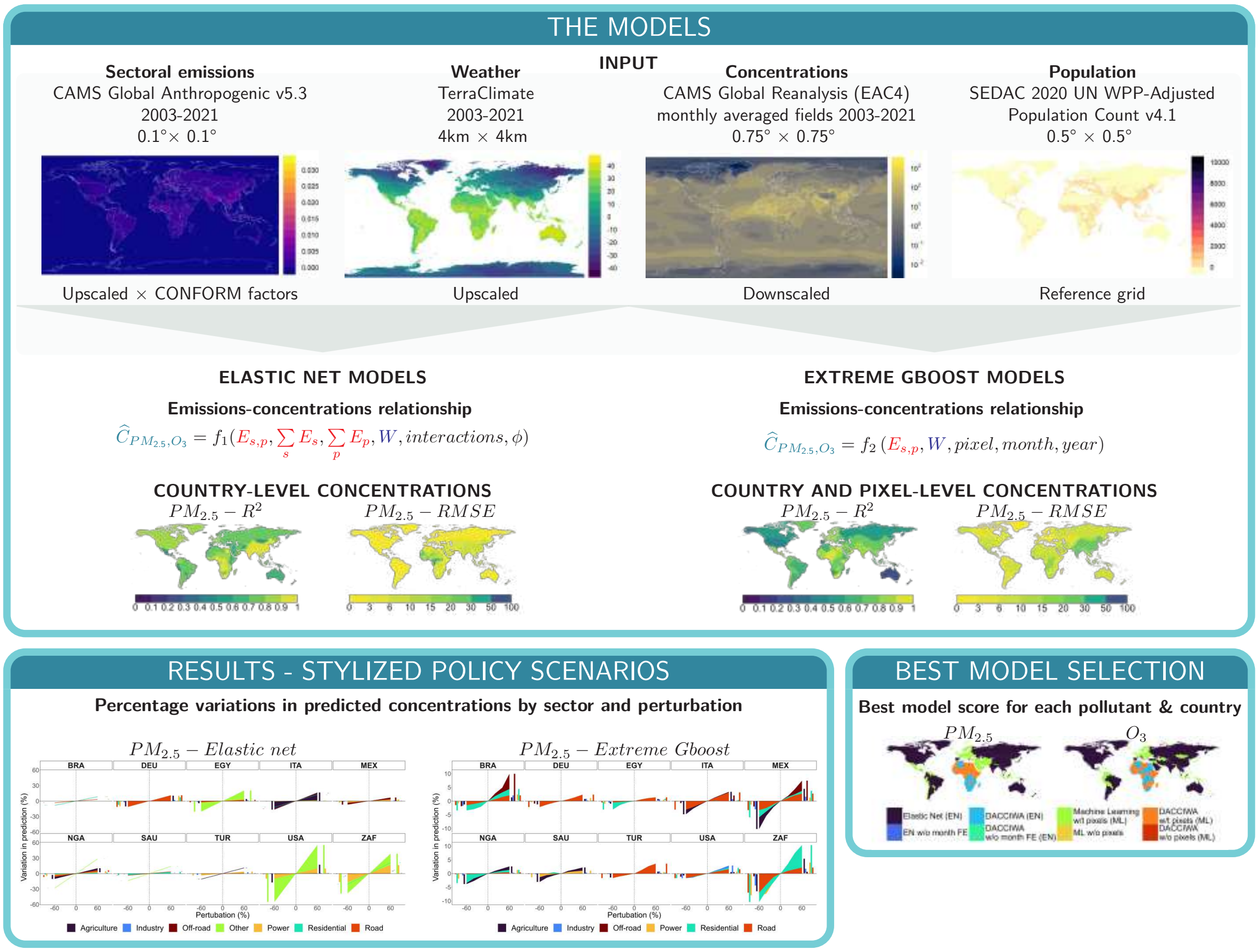

# Highlights

## **CLAQC v1.0 – Country Level Air Quality Calculator. An empirical modeling approach.**

Stefania Renna,Francesco Granella,Lara Aleluia Reis,Paulina Schulz-Antipa

- We develop a country-level air quality calculator based on empirical models to predict $PM_{2.5}$ and $O_3$ concentrations from sectoral emissions and meteorology.
- We use the CAMS emissions and reanalysis products to provide global coverage.
- We include the COVID-19 lockdown period for its disruptive emission and concentration levels.
- We use multiple criteria to select the best model.
- We find that country-level models perform well on annual averages, while some empirical methods are not yet robust for sectoral attribution.

# CLAQC v1.0 – Country Level Air Quality Calculator. An empirical modeling approach.★

Stefania Renna[a,b,c,*], Francesco Granella[a,b,d], Lara Aleluia Reis[a,b] and Paulina Schulz-Antipa[e]

[a] *CMCC Foundation - Euro-Mediterranean Center on Climate Change, Lecce, Italy*
[b] *RFF-CMCC European Institute on Economics and the Environment, Milan, Italy*
[c] *Department of Management, Economics and Industrial Engineering, Politecnico di Milano, Milan, Italy*
[d] *Bocconi University, Milan, Italy*
[e] *World Bank, Washington, DC, United States*

ARTICLE INFO

*Keywords*:
air pollution
emissions-concentrations function
$PM_{2.5}$
$O_3$
sectoral emission
country-level

ABSTRACT

The Country Level Air Quality Calculator (CLAQC) is an open-source modeling tool that utilizes national sectoral emissions and weather data to forecast monthly and annual concentrations of $PM_{2.5}$ and $O_3$. CLAQC leverages the recent advancements in the CAMS system, employing CAMS global gridded emissions and CAMS reanalysis pollutant concentrations to improve the accuracy of its predictions. One of the notable strengths of CLAQC is its ability to provide country-specific and sectoral information. We have developed two methodological approaches, namely elastic net modeling and extreme gradient boosting regressor, that can effectively predict annual average concentrations for nearly all countries. Although both methods show good performance for the country's yearly average, the sectoral contributions are not robust enough for the elastic net models. The tool can simulate a vast range of policy scenarios and can be integrated into national policy assessment and optimization frameworks. Finally, we present a method selection framework for each country to optimize performance, and an online tool displaying model results.

## 1. Introduction

Exposure to air pollution is a significant global health concern (Murray et al., 2020), recognized by the World Health Organization (WHO) as the first environmental health risk factor (World Health Organization, 2021). In 2019, ambient air pollution was responsible for approximately 8% of worldwide deaths, amounting to 4.51 million deaths. The majority of these deaths (92%) were caused by fine particulate matter (particles with a diameter less than 2.5 μm, $PM_{2.5}$), while the rest were due to tropospheric ozone ($O_3$) (Fuller et al., 2022; Institute for Health Metrics and Evaluation (IHME), 2019).

Policies targeting energy and environmental sectors impact airborne pollutants, leading to both co-benefits and trade-offs in air pollution (Eastham, Monier, Rothenberg, Paltsev, & Selin, 2023). Therefore, it is essential to consider the impact on air pollution when designing global and national policies (Reis, Drouet, & Tavoni, 2022). There is a clear need for tools that quantify the impacts associated with such policies in the field of integrated assessment modeling (IAMs) and national to global policy scenario assessment.

IAMs are analytical tools that aim to comprehend the interactions between the earth and human systems to assist policymakers in devising effective and cost-efficient greenhouse gas and air pollution control policies. By estimating the

★This document is the result of a research project funded by the World Bank. The CLAQC tool was developed as an input for the IMF-World Bank Climate Policy Assessment tool (CPAT).

*Corresponding author

**Principal corresponding author

stefania.renna@cmcc.it (S. Renna); francesco.granella@cmcc.it (F. Granella); lara.aleluia@cmcc.it (L. Aleluia Reis); pschulzantipa@worldbank.org (P. Schulz-Antipa)

https://github.com/stfnrnn/,https://www.eiee.org/member/stefania-renna/ (S. Renna); https://francescogranella.github.io/,www.eiee.org/team-member/francesco-granella/ (F. Granella); https://laleluia.github.io/page/,https://www.eiee.org/team-member/lara-aleluia-reis/ (L. Aleluia Reis)

ORCID(s): https://orcid.org/0000-0001-8096-6320 (S. Renna); https://orcid.org/0000-0002-2349-0132 (F. Granella); https://orcid.org/0000-0002-6676-7007 (L. Aleluia Reis)

contributions of various sectors, it becomes possible to focus resources on problematic areas and prioritize regulations or incentives that promote cleaner production or consumption practices.

Considered cutting-edge for their physicochemical representation detail, chemistry-transport models (CTMs) are tools for calculating the impact of emissions on pollutant concentration levels. However, they can be computationally heavy and challenging to use. Reduced-form CTMs address such limitations in trading accuracy and process detail for computational efficiency and can be used to assess multiple scenarios and incorporated into policy optimization frameworks. The most commonly used models to evaluate air pollution policies, such as the Greenhouse Gas - Air Pollution Interactions and Synergies (GAINS) model (Amann et al., 2011; Kiesewetter et al., 2015), the SHERPA tool (Thunis, Degraeuwe, Pisoni, Ferrari, & Clappier, 2016), the TM5-FAst Scenario Screening Tool (TM5-FASST) model (Dingenen et al., 2018), and the Air Control Toolbox (ACT) tool (Colette, Rouïl, Meleux, Lemaire, & Raux, 2022), rely on a variety of methods.[1]

Recent efforts to provide more policy-specific contributions to air pollution have also emerged, such as the Climate Action Planning-Air Quality (CAP-AQ) framework proposed by Kleiman et al. (2022). This approach emulates a CTM using 100 United States sites, although limited to the city level, and integrates sectoral climate-related greenhouse gas emission reductions, air quality policy, and health-related co-benefits in policy planning. Eastham et al. (2023) employ a CTM to estimate local-specific response functions that may be used to derive sector-specific contributions, although using a lower underlying resolution than the TM5-FASST model. The Global Intervention Model for Air Pollution (Global InMAP) is a global-scale reduced-form air quality modeling tool that simulates $PM_{2.5}$ concentrations resulting from different source heights and sites at a heterogeneous grid size (Thakrar et al., 2022). The latter one is the most detailed, up-to-date reduced-form air pollution model on a global scale. Yet, even at its coarser temporal and spatial resolution, its run time is still not compatible with easy-to-implement fast policy assessment. Other studies have compared several air pollution health impact assessment tools (Anenberg et al., 2016), applied similar reduced modeling methods for local-scale air pollution modeling (Oxley et al., 2022), or assessed the sectoral and fuel-specific contributions of $PM_{2.5}$ to mortality at various geographic scales (McDuffie et al., 2021) using CTMs in simulation mode. However, only the Response Surface Model by Eastham et al. (2023) is able to simultaneously provide annual sectoral atmospheric contributions to both $PM_{2.5}$ and $O_3$ concentrations at the national and country level with global coverage.

All in all, the available reduced-form CTMs allow the assessment of multiple scenarios, ultimately contributing to better policy design. However, they are based on CTMs and are therefore bounded by their resolution and scope. They are also known to be less robust for highly non-linear processes, such as secondary $O_3$ formation and secondary PM formation (Dingenen et al., 2018; Thunis et al., 2019). Furthermore, they are limited by the number of underlying scenarios in their training set. The Country-Level Air Quality Calculator (CLAQC) is a statistical model that aims at filling this gap by learning from coupled historical variations in concentrations of $PM_{2.5}$ and $O_3$, sectoral emissions of multiple precursors, and meteorology. Similarly to deterministic reduced-form models, empirical models are also bounded, but by the underlying observed data. Unlike in the source-receptor models, the perturbation level is not set (Thunis et al., 2019), as the model learns from all past variations. Provided with adequate data, a statistical model can learn from a broad spectrum of variations in emissions, concentrations, and atmospheric conditions, akin to simulating a large number of training scenarios in process-based models. CLAQC is built over 19 years of data spanning the entire global landmass and is provided with arguably a wide spectrum of input variables. The large drop in emissions induced by COVID-19 lockdowns and disruptions worldwide further expands the range of training variables. Additionally, CLAQC allows for more flexibility than the deterministic reduced-form models by relying on the global gridded Copernicus Atmosphere Monitoring Service (CAMS) reanalysis products. As new and better data come in every year, the emulator can be updated, and a higher detail level may be possible at lower trade-off costs. For example, the sub-national detail can assist with better placement of energy, transport, and industrial infrastructure. CLAQC complements the above-mentioned models by providing an easy-to-use global tool with country and sectoral details.

The next section discusses the data used. We then present in section 3 the different methodological approaches that have been followed: elastic net models (section 3.3), and machine learning models (section 3.4). While in section 3.5, we compare the two methodological approaches. In section 4, we present the validation of results and sensitivity analyses. In section 5, we discuss the limitations of our tool. Finally, we draw conclusions in section 6.

[1] *E.g.*, GAINS is a fully IAM using emissions-concentrations relationships from the Unified EMEP Eulerian model and CHIMERE. Similarly, the SHERPA tool emulates scenario results from EMEP and CHIMERE (Menut et al., 2021), while TM5-FASST is a source-receptor model based on the chemical transport model TM5 (Dingenen et al., 2018).

# 2. Data

Monitoring stations are used for air quality assessment. However, the lack of a spatially consistent large ground monitoring network in a given area is a strong constraint to achieving this objective. Despite the recent harmonization and open-access advancements in air pollution data, most publicly available global ground-level monitoring databases (*e.g.*, *OpenAQ* (2024)) provide reasonable territorial coverage of the population only in developed countries, in particular in the United States and Europe. The network of ground-level monitors is growing in emerging economies such as China and India, yet urban and rural areas are largely unmonitored in middle- and low-income countries. The uneven ground-level monitoring geographical coverage is problematic, as factors driving the emissions-concentrations relationship differ between monitored and unmonitored areas (e.g., population density, distance from industrial sources, gross domestic product per capita). Emissions-to-concentrations relationships learned using ground-level monitoring yield estimates that are biased toward richer countries.

To meet the CLAQC objectives of global coverage, an alternative option to ground-level monitoring data is using reanalysis data. Global, gridded reanalysis data combine and harmonize satellite air pollution measurements and CTM output with ground-level monitors. While maintaining the quality of the monitor data at the location of the monitors, they bridge the gap in ground-level monitoring networks with satellite observations and models that span the entire globe. This principle, called data assimilation, is based on the method used in numerical weather prediction and air quality forecasting, where a previous forecast is combined with newly available observations in an optimal way to produce a new best estimate of the state of the atmosphere. Reanalysis does not have the constraint of timely forecasts, allowing time to collect observations and allowing for the integration of improved versions of the original observations, raising the quality of the reanalysis product.

Using gridded data has many other advantages, allowing for:

- Weighting reductions in concentrations by population, obtaining changes in exposure to pollutants;
- Better identifying the interactions of emissions with meteorology and topography;
- Increasing statistical power without compromising the estimation of country-specific emissions-to-concentrations functions;
- Reducing rigidity on the spatial scope (not limited to administrative borders) and keeping the sub-national modeling option flexible;
- Global coverage, even in areas without ground-level monitoring.

Among the disadvantages, the need to homogenize different grids in terms of spatial resolution may lead to approximations during the data manipulation process.

## 2.1. Emissions

### *2.1.1. Precursors*

The precursors of $PM_{2.5}$ included in the models are: BC, OC, $NH_3$, $NO_x$, $SO_2$, and NMVOC. All are expected to increase $PM_{2.5}$ concentrations at the country level, although local decreases on the secondary fraction may happen (Clappier, Thunis, Beekmann, Putaud, & de Meij, 2021). Data on OC and BC is almost perfectly collinear: thus, emissions from these precursors are summed into Total Carbon (TC) in the machine learning models where both are available.

The precursors of $O_3$ included in the models are its main precursors: NMVOC, $NO_x$, and $SO_2$. NMVOC are expected to increase $O_3$, whereas the relationship between $NO_x$ and $SO_2$ to $O_3$ may be negative (Dingenen et al., 2018). We follow the approach of most reduced-form models, leaving out the CO precursor for its relatively minor importance (Amann et al., 2011).

### *2.1.2. CAMS emissions*

Emission data is provided by CAMS Global Anthropogenic v5.3, with a monthly temporal resolution and a spatial resolution of 0.1°. The data is originally expressed in Teragrams ($Tg$) and is converted into kilograms ($kg$).The CAMS emission data is based on existing available databases, including nationally reported emissions, the Joint Research Center's (JRC) Emissions Database for Global Atmospheric Research (EDGAR) (Crippa et al., 2018; Huang et al., 2017), Evaluating the Climate and Air Quality Impacts of Short-Lived Pollutants (ECLIPSE) (Stohl et al., 2015), and

the Community Emissions Data System (CEDS) databases (Hoesly et al., 2018; McDuffie et al., 2020). It has the advantage of providing global gridded monthly emissions from 2000 up to 2021, although emission estimates of most recent years (2015-2021) are extrapolated by applying CEDS 2014-2019 country-level trends to gridded EDGAR v5 data and therefore are associated with higher uncertainty. See Granier et al. (2019, 2021) for more detail on CAMS-GLOB-ANT data harmonization and sectoral definitions.

CAMS emission inventory is based on business-as-usual emissions and does not take into account lockdown measures and restrictions put in place to tackle the COVID-19 pandemic. To correct this, we apply to 2020 emissions the COvid-19 adjustmeNt Factors fOR eMissions (CONFORM) constructed by Doumbia et al. (2021) for the following sectors: Power, Industry, Residential, Public and Commercial, Transport[2].

#### *2.1.3. DACCIWA emissions*

Data quality in low-income countries can be comparably poorer due to the scarcity of measurements. Therefore, we additionally consider the Dynamics-aerosol-chemistry-cloud interactions in West Africa (DACCIWA) emission data set as model input, an Africa-specific regional emission inventory developed for providing more accurate estimations for African countries, employing updated emission factors based on in-situ measurements (Keita et al., 2021). DACCIWA covers major human-related emission sources characterizing the African continent such as charcoal production, wood stove combustion, and open-air garbage combustion, and classifies emissions into the following sectors: traffic, energy, residential, industry, other, and waste. It has the same spatial resolution as CAMS emissions and covers 1990 to 2015.

#### *2.1.4. Sectoral aggregation*

Our focus is on identifying sectors that are likely to be directly impacted by policies aimed at reducing the use of fossil fuels. However, emissions from the various sectors, with the exception of the agricultural one, are highly collinear, making it difficult to distinguish the contribution of each individual sector to the total pollution levels (see section A.1 for further details). To overcome this challenge, we group together sectors with similar emission patterns, reducing the complexity of the data and improving the accuracy of our analyses. At the same time, we try to keep sectoral relevance for policy models such as the IMF-World Bank Climate Policy Assessment Tool (CPAT) (Black, Parry, Mylonas, Vernon, & Zhunussova, 2023). We do not include biogenic and sectoral emissions from shipping and aviation. Furthermore, natural emissions, such as desert dust and sea salt, are not taken into account since they are less likely to be subject to policy interventions. We build the following 7 sectors from CAMS-GLOB-ANT data: Agriculture, Industry, Other (including the emissions not considered in the other sectors), Off-road transportation, Energy power generation, Road transportation, and Residential (including buildings, commercial and services). Note that the Off-road transportation sector includes railways and other types of non-road transports not typically used on public roads, such as agricultural machinery, construction equipment, and certain types of off-road vehicles used in industrial operations (e.g., tractors, telehandlers, excavators). See Table 1 for the correspondence between the CAMS-GLOB-ANT sectors and the CLAQC ones.

Note that CAMS and DACCIWA apply slightly different sectoral classifications. For instance, we retain two transport-related sectors from CAMS after aggregation, whereas DACCIWA only includes one transport sector. Moreover, the latter does not consider agriculture in the original data. As a result, we aggregate the available sectors into the following categories: Transport, Power, Industry, Residential, and Other (with the latter one containing waste as well).

### 2.2. Meteorology

All meteorological data, with the exception of wind direction, comes from TerraClimate (Abatzoglou, Dobrowski, Parks, & Hegewisch, 2018). TerraClimate has a wide variety of meteorological variables, good temporal coverage (1958 to 2022), high spatial resolution (4 km × 4 km), and a monthly time resolution. The following atmospheric variables are used as inputs to the models: accumulated precipitation in $mm$; maximum 2-m temperature in degrees Celsius ($°C$); minimum 2-m temperature in $°C$; 10-m wind speed in $m/s$; mean vapor pressure deficit in $kPa$. The wind direction in $degrees$ comes from the European Centre for Medium-Range Weather Forecast's (ECMWF) ERA5 Reanalysis Monthly Means product by Copernicus Climate Change Service (C3S) (Copernicus Climate Change Service, 2019).

[2]CONFORM data were downloaded from the ECCAD portal at `https://eccad.aeris-data.fr`.

**Table 1**
Correspondence between CAMS-GLOB-ANT and CLAQC sectoral aggregation.

| CAMS-GLOB-ANT sector | CLAQC sector |
|---|---|
| Agriculture livestock (AGL)<br>Agriculture soils (AGS)<br>Agriculture waste burning (AWB) | Agriculture |
| Industrial processes (IND) | Industry |
| Solvents application and production (SLV)<br>Solid waste and wastewater handling (SWD) | Other, including the emissions not considered in the other sectors |
| Non-road transportation (TNR) | Off-road transportation |
| Power generation (ENE)<br>Fugitives emissions from solid fuels (FEF)<br>Refineries (REF) | Energy power generation |
| Road transportation (TRO) | Road transportation |
| Residential (SER) | Residential, including buildings, commercial and services |

## 2.3. Concentrations

We obtain the ground-level ambient concentration data for $PM_{2.5}$ and $O_3$ air pollutants from the ECMWF's Atmospheric Composition Reanalysis 4 (EAC4) Monthly Averaged Fields (Inness et al., 2019). The data cover the period from 2003 to 2021, which is the shortest time domain of all the available data sets. Consequently, all other data sets are limited to this time period. The original spatial resolution of the data is 0.75°, which is down-scaled to 0.5°. $PM_{2.5}$ and the mixing ratio of surface-level ozone (obtained from the GEMS Ozone model level 60) are originally expressed in $kg/m^3$ and $kg/kg$, respectively. To facilitate analysis, we convert them to micrograms per cubic meter ($\mu g/m^3$). As an example, Figure 1 shows EAC4 concentration levels of $PM_{2.5}$ and $O_3$ for January 2018 and July 2018, respectively.

To transform country-level, monthly $PM_{2.5}$ concentrations into population-weighted exposure, $Exp_{k,m}$, for country $k$ and month $m$, we use the 2020 UN WPP-Adjusted Population Count, v4.11, at 30 arc-second spatial resolution, from the Center for International Earth Science Information Network (CIESIN) (Center For International Earth Science Information Network-CIESIN-Columbia University, 2018). We calculate monthly country-level exposure, $Exp_{k,m}$, by summing over grid cells $i$ the population weights, $\frac{pop_i}{pop_k}$, times grid-level, monthly concentrations, $C_{i,m}$ (Equation 1). We only use population data referring to one year, 2020, to avoid introducing another source of variation in the models. To give a sense of the data, we display 2018 country-level weighted concentrations of $PM_{2.5}$ and $O_3$ in Figure 2.

$$Exp_{k,m} = \sum_{i=1}^{n} \frac{pop_i}{pop_k} \cdot C_{i,m} \tag{1}$$

## 2.4. Grid definition

All gridded data sources are rescaled to the same $0.5° \times 0.5°$ coordinate grid through linear interpolation, based on the population grid, and merged into a single data set. For instance, concentration data originally at $0.75° \times 0.75°$ spatial resolution are downscaled to $0.5° \times 0.5°$, generating intermediate values that align with the reference grid. The interpolation is implemented using the `interp_like` function from the `xarray` Python package. Notice that some cells may be attributed to multiple countries in case the centroid falls exactly on the countries' borders. This should not be a source of concern as models are independently run country by country.

(a) $PM_{2.5}$

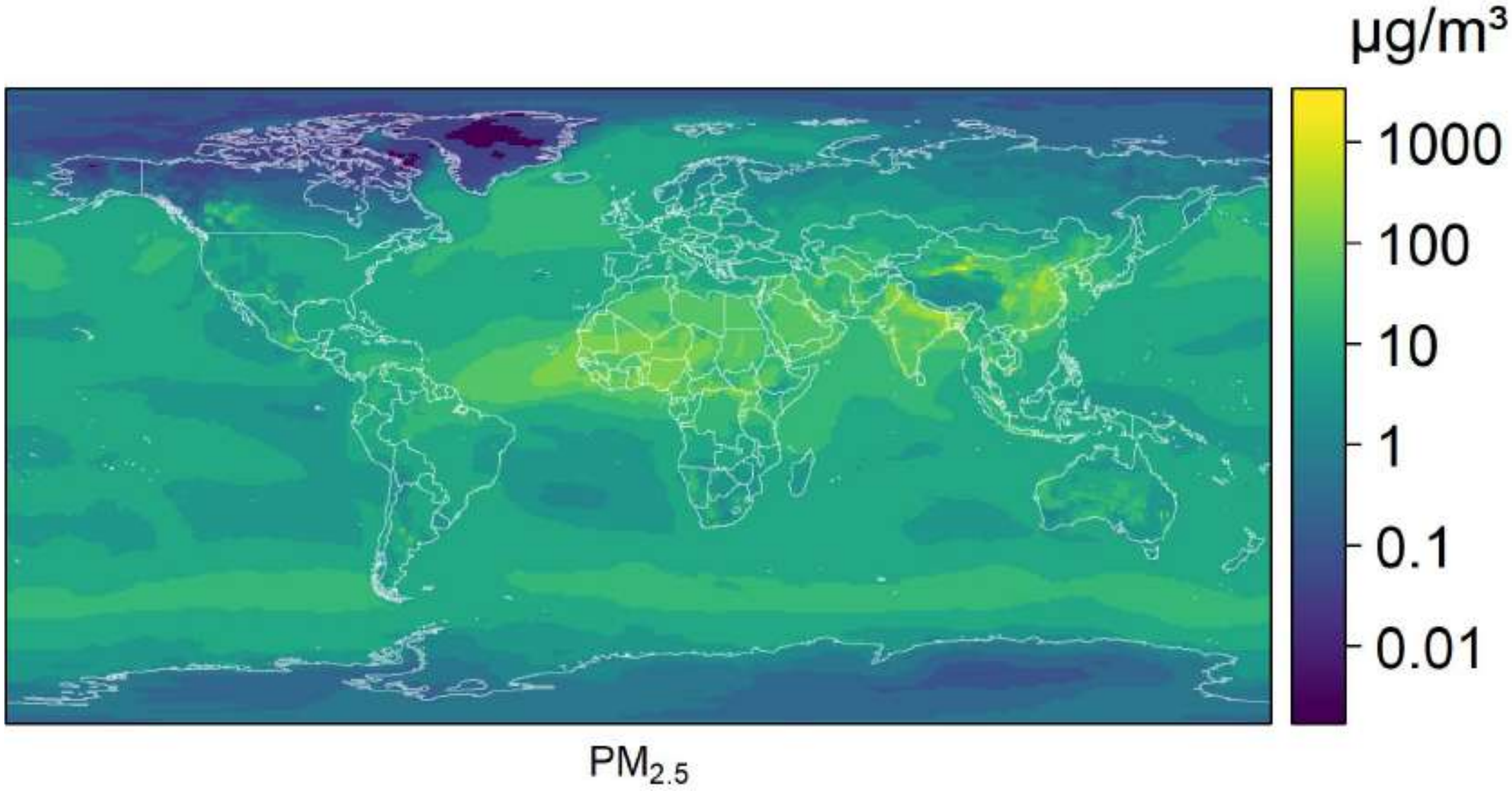

(b) $O_3$

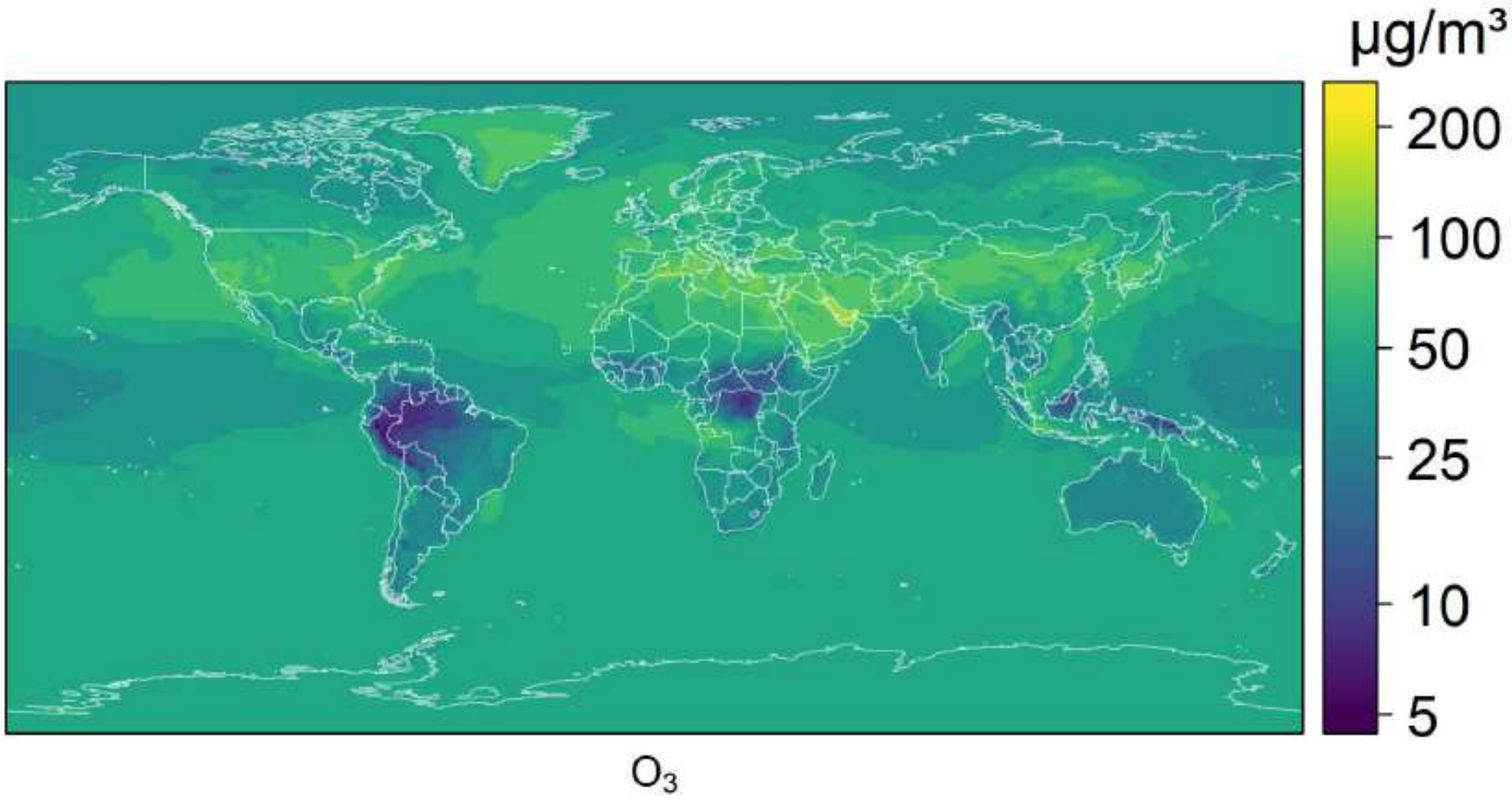

**Figure 1:** Level plots of EAC4 concentrations of $PM_{2.5}$ (January 2018) and $O_3$ (July 2018) in $\mu g/m^3$ with color bar.

# 3. Methods

## 3.1. CLAQC rationale

We are interested in estimating the relationship between emissions $E$ of major ambient air pollutants and the respective ground-level concentrations $C$ of major pollutants $c$ ($PM_{2.5}$, $O_3$). Denote such relationship $f$, so that

$$C_c = f(E) \tag{2}$$

The formation, transport, and dispersion of pollutants are complex natural phenomena that are highly dependent on emissions, weather $W$, and other local characteristics such as topography. Hence, the design of pollution abatement

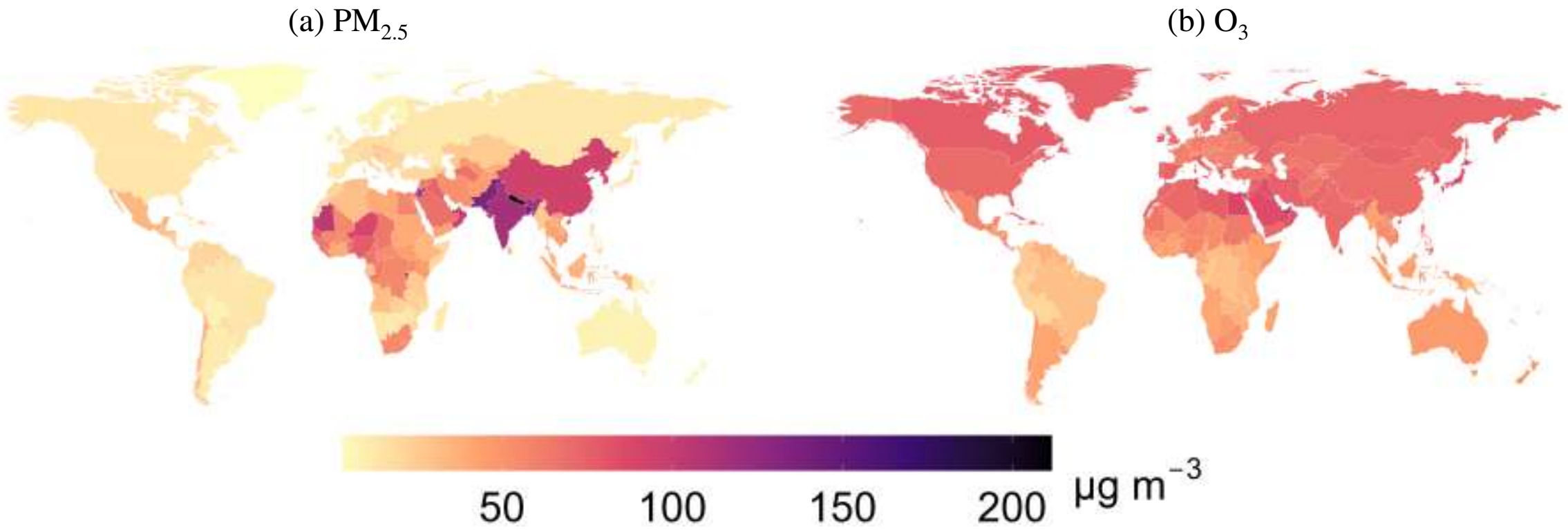

**Figure 2:** EAC4 concentration inputs of $PM_{2.5}$ weighted by the population and $O_3$ in $\mu g/m^3$ aggregated at the country level (2018).

policies in the country $k$ can benefit from the estimation of a country-specific emissions-to-concentrations function (a country-wide population-weighted average) that accounts for the interplay between emissions and weather:

$$C_{c,k} = f_k(E_k, W_k) \tag{3}$$

However, environmental and fiscal policies have heterogeneous effects across the main sectors of emissions and precursors, for instance by inducing a rearrangement in the energy mix. Therefore, it is helpful to establish how country-wide changes in the emissions of precursors $p$, from a given sector $s$, alter ambient concentrations. The emitting sectors $s$ include energy production, industries, buildings, transport, and agriculture, among others. Pollutant precursors $p$ include black carbon (BC), organic carbon (OC), ammonia ($NH_3$), non-methane volatile organic compounds (NMVOC), nitrogen oxides ($NO_x$), and sulphur dioxide ($SO_2$). We are thus interested in estimating the following relationship:

$$C_{c,k} = f_{k,s,p}(E_{k,s,p}, W_k). \tag{4}$$

We identify two methods to empirically derive $\widehat{f}_{k,s,p}$, that trade off simplicity and transparency with prediction power, as we explain in what follows. The first method relies on elastic net models, a penalized linear regression amenable to a large number of predicting variables while preserving an intelligible structure. $\widehat{f}_{k,s,p}$ is modeled as a linear function of emissions and weather variables that can be easily reproduced.

The second method relies on machine learning algorithms that are better suited than linear models to learn highly non-linear relationships, such as those between precursors and weather conditions. Better performance comes, however, at the cost of interpretability, as machine learning algorithms typically do not return simple predictor-target functions. For this reason, we also provide approximate emissions-to-concentrations relationships with functions that are suitable for simpler spreadsheet-style use.

We follow two approaches that trade off interpretability and predictive performance. We first estimate an Elastic Net model, a linear model with selection and shrinkage. The linear form allows for easy interpretation of coefficients, whereas selection and shrinkage give more weight to the variables of the highest importance and address multicollinearities in the data. Second, we fit an extreme gradient boosting regressor, a decision tree-based machine learning algorithm. Figure 3 shows the schematic representation of the CLAQC workflow.

## 3.2. Coefficient constraints

We impose monotonic constraints on certain model coefficients to align with expected physicochemical relationships. These constraints specify how input variables should affect the target, ensuring interpretable and physically plausible results. For instance, a positive monotonic constraint enforces a non-negative relationship, ensuring that as an input variable increases, the predictor output does not decrease.

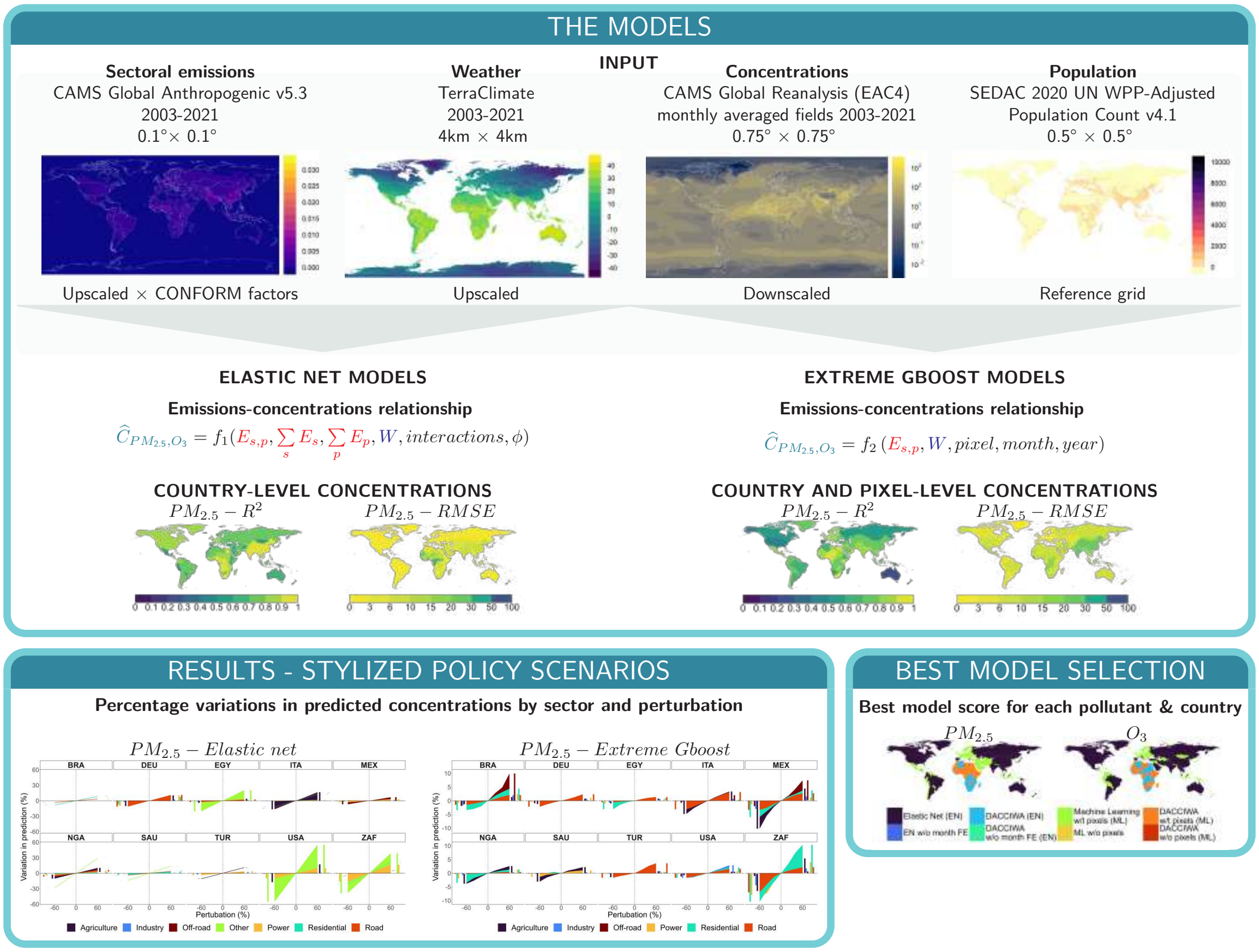

**Figure 3:** Methodological abstract.

In the presence of noise, complex interactions in the data, or predictor cross-correlation, models may otherwise learn patterns that are not realistic or physically plausible. Additionally, monotonic constraints help prevent overfitting, enhancing robustness when input data are limited or uncertain. For example, it is not expected that an increase in BC emissions would lead to a decrease in $PM_{2.5}$ concentrations.

While at the local scale, reducing certain precursors of secondary inorganic aerosols might not always lead to a decrease in $PM_{2.5}$ levels — due to nonlinear atmospheric reactions noted by Ding et al. (2021); Thunis et al. (2019) — our national-scale models focus on broader trends. To avoid giving undue importance to cases where local emissions reductions might result in increased levels of inorganic $PM_{2.5}$, we apply monotonic constraints between emissions and concentrations.

Rather than directly including secondary inorganic aerosols, the models incorporate interactions between PM precursors — specifically $NH_3$, $NO_x$, and $SO_2$ — as proxies for secondary reactions.

It is crucial to understand that in situations where secondary reactions substantially affect the overall mass of $PM_{2.5}$ within a country, our models are designed to omit these precursors from the list of predictors, thereby not reflecting a decrease in $PM_{2.5}$ levels.

We further require that greater precipitations and temperatures decrease $PM_{2.5}$. Precipitation lowers $PM_{2.5}$ by wet deposition, while temperature is a proxy for inversion layer height, *i.e.*, high temperature generally means high inversion layer heights and therefore less concentration (John H. Seinfeld, 2016). Although wind speed normally facilitates pollutant dispersion, we impose no constraint on its role, as long-distance transportation of suspended particles may increase $PM_{2.5}$. All other coefficients are unbound.

Regarding $O_3$, similarly, we impose that emissions of NMVOC increase its concentrations while leaving emissions of $NO_x$ unconstrained, allowing for non-monotone relationships with $O_3$ (Ding et al., 2021). We also constrain temperature to increase $O_3$ concentrations (Jhun, Coull, Schwartz, Hubbell, & Koutrakis, 2015; Lu, Zhang, & Shen, 2019). $O_3$ is a photo-chemical secondary pollutant (John H. Seinfeld, 2016), which increases with intensifying solar radiation. Temperature is therefore used as a proxy.

We include the following variables in both models: sectoral emissions, precipitation, minimum temperature, maximum temperature, vapor pressure deficit, wind speed, and wind direction. In the case of EN, we also add monthly emission sectoral totals (*i.e.*, $\sum_s E_{k,s,m}$) and monthly emission pollutant totals (*i.e.*, $\sum_p E_{k,p,m}$) to increase the chances that the models capture variations in emissions, plus monthly fixed effects, and interaction terms (see section 3.3 for further details on EN model specifications).

## 3.3. Elastic net models

Due to the high multicollinearity among predictors, as shown in Appendix A, ordinary least squares (OLS) regression may fail to yield reliable parameter estimates. Penalized linear regression maintains the interpretability of coefficients of linear models while selecting the variables with the greatest predictive power. We use elastic net models (Zou & Hastie, 2005), a method suitable to identify the subset of best predictors obtaining a parsimonious model. It solves the following minimization problem for the model parameters $\beta_0$ and $\beta$, where $\beta_0$ is the model's intercept and $\beta$ represents the coefficients of the input variables:

$$\min_{\beta_0,\beta} \frac{1}{2N} \sum_{1}^{N} (y_i - \beta_0 - x_i^T \beta)^2 + \lambda[\frac{(1-\alpha)||\beta||_2^2}{2} + \alpha||\beta||_1] \tag{5}$$

Combining the penalty elements of the Least Absolute Selection and Shrinkage Operator (LASSO) regression ($||\beta||_1$) and Ridge regression ($||\beta||_2^2$) on the basis of the alpha ($\alpha$) parameter, the penalization parameter lambda ($\lambda$) selects variables like the former and shrinks them as the latter. It regularizes the model coefficients, improving the model's accuracy and interpretability by decreasing the input variables' space. This prevents our models from being volatile to extreme variations and outliers. Such a technique avoids large errors on the one hand, and, on the other, it results in more conservative estimations of the concentrations obtained from the emissions reductions.

We perform elastic net modeling in R Statistical language (R Core Team, 2020), version 4.0.2 (2020-06-22) on Windows 10 x64 (build 22621). To allow reproducing the R environment, we employ the `renv` package (Ushey, 2022). The elastic net workflow is represented in Figure 4 and follows the steps below. For each country:

1. To ensure reproducibility, a seed is set with the `set.seed` R function.
2. We average the gridded monthly data set to the country-year-month level.[3]
3. We then identify and exclude outliers by applying the interquartile range rule and listwise deletion.
4. We randomly split the 2003-2021 data into training (84% of observations) and test set (16%) stratifying by month.[4] We run sensitivity tests on the splitting ratio, obtaining robust results across splittings: for further insights see section A.5.
5. We apply a k-fold cross-validation algorithm for tuning the $\lambda$ regularization parameter by using the `cv.glmnet` function from the `glmnet` R package (J. Friedman, Hastie, & Tibshirani, 2010). We apply the following specifications: 30 folds, $\alpha = 0.5$ corresponding to elastic net regularization with no optimization of the alpha parameter, 'deviance' type.measure for specifying the mean squared error loss function, and 'gaussian' family (J. H. Friedman et al., 2020).
6. Monotonic constraints are imposed for certain predictors. See details in subsection 3.2.
7. We train the model on the training set by applying the `glmnet` function from the `glmnet` R package.[5]
8. We evaluate its performance on the test set, *i.e.*, on data not used to build the model itself. We report the out-of-sample R-Squared ($R^2$) and Root Mean Square Error (RMSE), calculated as in Equations 6 and 7. $y_i^{test}$ is the test set actual value for observation $i$, $\hat{y}_i^{test}$ is the test set predicted value for observation $i$, $\bar{y}^{test}$ is the mean value of the test set actual values, and $n_{test}$ is the number of observations in the test set.

[3]While we sum up sectoral emissions, we average weighted concentrations and meteorology variables. We treat wind direction as a `circular` variable through the `circular` function from the `circular` R package (Agostinelli & Lund, 2022).

[4]Using the `stratified` function from the `splitstackshape` R package (Mahto, 2019). Note that an 80-20 train-test splitting is applied for models based on DACCIWA emissions.

[5]Notice that the `glmnet` function standardizes by default all the variables, removing the influence of their scales.

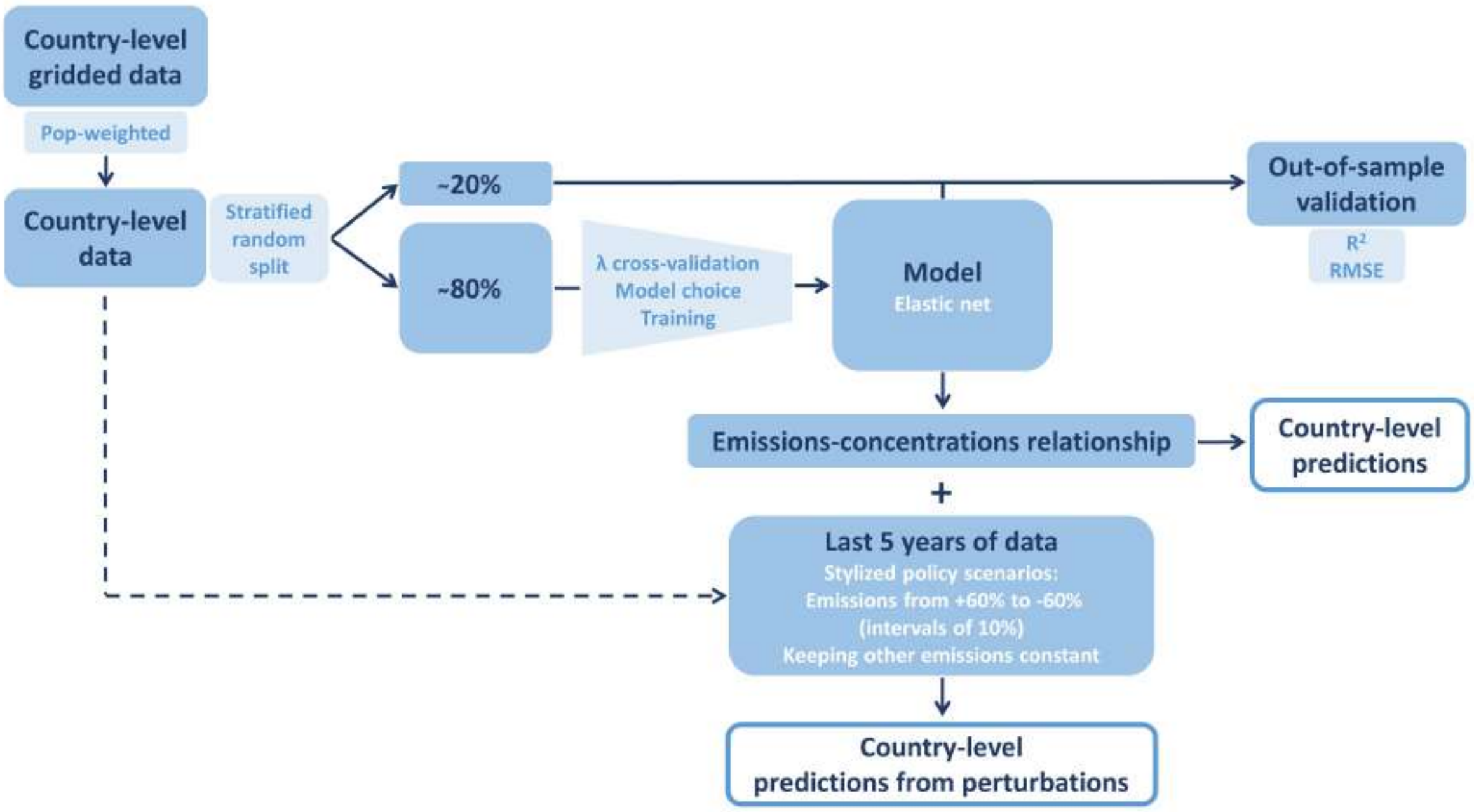

**Figure 4:** Visual representation of the elastic net models' workflow.

9. Finally, we predict concentrations for varying levels of emissions and derive empirical emissions-to-concentrations relationships. More specifically, we simulate perturbations of emissions from -60% to +60% at 10% steps based on the last 5 years of data. This timeframe is selected to reflect recent trends, offering more policy-relevant insights into the empirical relationship between emissions and concentrations. Notice that the user could choose another time period for simulations.

$$R^2 = 1 - \frac{\sum_{i=1}^{n}(y_i^{\text{test}} - \hat{y}_i^{\text{test}})^2}{\sum_{i=1}^{n}(y_i^{\text{test}} - \bar{y}^{\text{test}})^2} \tag{6}$$

$$RMSE = \sqrt{\frac{\sum_{i=1}^{n}(y_i^{\text{test}} - \hat{y}_i^{\text{test}})^2}{n_{\text{test}}}} \tag{7}$$

The elastic net linear regression models take the following form for each country (Equations 8 and 9) (John H. Seinfeld, 2016):

$$\begin{aligned} PM_{2.5t} = \alpha + \sum_{s,p_1} \beta_{s,p_1} E_{s,p_1,t} + \gamma_1 PPT_t + \gamma_2 TMIN_t + \gamma_3 TMAX_t + \gamma_4 VPD_t + \gamma_5 WS_t + \gamma_6 WD_t + \\ + \sum_s \delta_s E_{s,t} + \sum_{p_1} \lambda_{p_1} E_{p_1,t} + \mu E_{NO_x,t} \cdot E_{NH_3,t} + \nu E_{SO_2,t} \cdot E_{NH_3,t} + \xi E_{SO_2,t} \cdot E_{NO_x,t} + \\ + \sum_s \theta_s E_{s,t} \cdot WS_t \cdot WD_t + \phi_t + \varepsilon_t \end{aligned} \tag{8}$$

$$\begin{aligned} O_{3t} = \alpha + \sum_{s,p_2} \beta_{s,p_2} E_{s,p_2,t} + \gamma_1 PPT_t + \gamma_2 TMIN_t + \gamma_3 TMAX_t + \gamma_4 VPD_t + \gamma_5 WS_t + \gamma_6 WD_t + \\ + \sum_s \delta_s E_{s,t} + \sum_{p_3} \lambda_{p_3} E_{p_3,t} + \mu E_{NOx,t} \cdot E_{NMVOC,t} + \nu E_{SO_2,t} \cdot E_{NMVOC,t} + \xi E_{SO_2,t} \cdot E_{NO_x,t} + \end{aligned} \tag{9}$$

$$+\sum_s \theta_s E_{s,t} \cdot WS_t \cdot WD_t + \phi_t + \varepsilon_t$$

where:

$s \in \{Agriculture,\ Industry,\ Other,\ Off\text{-}road\ transportation,\ Energy\ power\ generation,\ Road\ transportation,\ Residential\}$
$p_1 \in \{BC,\ NH_3,\ NMVOC,\ NO_x,\ OC,\ SO_2\}$
$p_2 \in \{NMVOC,\ NO_x\}$
$p_3 \in \{NMVOC,\ NO_x,\ SO_2\}$
$PM_{2.5t}$ = Monthly concentration of $PM_{2.5}$ in $\mu g/m^3$ (population-weighted)
$O_{3t}$ = Monthly concentration of $O_3$ in $\mu g/m^3$
$E_{s,p_1,t}$ = Monthly emissions of sector $s$ and pollutant $p_1$ in $kg$
$E_{s,p_2,t}$ = Monthly emissions of sector $s$ and pollutant $p_2$ in $kg$
$E_{s,p_3,t}$ = Monthly emissions of sector $s$ and pollutant $p_3$ in $kg$
$PPT_t$ = Monthly accumulated precipitation in *mm*
$TMIN_t$ = Monthly minimum 2-m temperature in $°C$
$TMAX_t$ = Monthly maximum 2-m temperature in $°C$
$VPD_t$ = Monthly mean vapor pressure deficit in $kPa$
$WS_t$ = Monthly 10-m wind speed in $\frac{m}{s}$
$WD_t$ = Monthly wind direction in $degrees$
$E_{p_1,t}$ = Monthly composite index from the sum of total emissions of pollutant $p_1$ in $kg$
$E_{p_2,t}$ = Monthly composite index from the sum of total emissions of pollutant $p_2$ in $kg$
$E_{p_3,t}$ = Monthly composite index from the sum of total emissions of pollutant $p_3$ in $kg$
$E_{s,t}$ = Monthly composite index from the sum of total emissions of sectors $s$ in $kg$
$\phi_t$ = Monthly fixed effects
$\varepsilon_t$ = Error term

In Equations 8 and 9, $t$ indicates time, $s$ is the emission sector, and $p_{[n]}$ refers to the sector-related emitted pollutants in their respective models. $PM_{2.5}$ and $O_3$ concentration values obtained from the models in $\mu g/m^3$ are country-level monthly concentration averages indexed by time $t$, just as all the other parameters in the equation; notice that $PM_{2.5}$ levels are weighted by population as explained in subsection 2.3; $\alpha$ is the model intercept; $\lambda$, $\beta$, $\gamma_i$, $\delta$, $\mu$, $\nu$, $\epsilon$, $\theta$ are the predictors' coefficients; $E_{s,p_{[n]}}$ are emissions of sector $s$ and pollutant $p_{[n]}$, respectively; $E_p$ and $E_s$ are total emissions of pollutant $p$ and of sector $s$, respectively; all emission variables are expressed in $kg$; $PPT_t$ stands for accumulated precipitation in *mm*; $TMIN_t$ and $TMAX_t$ are minimum 2-m temperature and maximum 2-m temperature, respectively, in $°C$; $VPD_t$ is mean vapor pressure deficit in $kPa$; $WS_t$ is 10-m wind speed in $m/s$; $WD_t$ is average wind direction in $degrees$; $\phi_t$ are month fixed effects; finally, $\varepsilon_t$ is the stochastic term. Note that the emission terms in the Equations differ due to their different atmospheric reactions. In both Equations, we include multiple emission terms to increase the chances that models capture variations in emissions. In Equation 8, to model the secondary inorganic aerosol formation, we interact total emissions of $NO_x$ and $NH_3$, $SO_2$ and $NH_3$, and $NO_x$ and $SO_2$, respectively. Similarly, in Equation 9, we interact total emissions of $NO_x$ and NMVOC, $SO_2$ and NMVOC, and $SO_2$ and $NO_x$. As before, this attempts to capture the reactions between the precursors of $O_3$, since the presence of at least two of these precursors is necessary for its formation. While NMVOC and $NO_x$ are $O_3$ main precursors, reacting in the presence of solar radiation, $SO_2$ plays an indirect role in $O_3$ formation (Baird & Cann, 2013; John H. Seinfeld, 2016). $SO_2$ is typically emitted by industrial sources. It is involved in secondary PM formation, which can reduce the radiative properties and oxidative capacity of the atmosphere, indirectly affecting $O_3$ formation. In both Equations, we also interact sectoral emissions with wind speed and direction to proxy transport and dispersion of pollutants. We include total sectoral emissions to reflect that sector-specific policies typically impact multiple pollutants through dedicated emission offset protocols. Additionally, we consider total emissions from individual pollutants because variations in total pollutant emissions may result not only from specific sectors but also from inter-sector changes, transported emissions, and chemical reactions. Refer to section A.2.2 for the EN model specifications with DACCIWA emissions, and to section A.4 for EN model implementation.

In order to have non-negative predicted values for $y$, we add a non-negativity constraint that selects the maximum value between zero and the EN model prediction, $\hat{y}_{EN}$. Moreover, in order to have non-extreme predicted values for $y$, e.g., due to input data divergence, we apply a second safety function that caps predicted values to three times the observed country-level concentrations under no perturbations, $3y$:

$$\hat{y}^* = \min\left\{\max\left\{0,\ \hat{y}_{EN}\right\}, 3y\right\} \tag{10}$$

*I.e.*, the final prediction $\hat{y}^*$ is equal to 0 if $\hat{y}_{EN} < 0$, and equal to $\hat{y}_{EN}$ or at most $3y$ if $\hat{y}_{EN} \geq 0$.

## 3.4. Machine learning models

Emissions-to-concentrations functions might not be sufficiently well approximated by a linear function due to the non-linearities of topography and secondary pollution formation (Thunis et al., 2019). Machine learning models are powerful tools that can reproduce highly nonlinear relationships like the complex natural phenomena behind air pollution formation, transport, and dispersion. Importantly, they do not require the user to impose a functional form. Differently from the modeling with elastic net, data are not spatially aggregated. In addition to the pre-processing steps described in section 2, we perform specific data processing.

1. Given the very high level of collinearity between BC and OC emissions data, we sum the two precursors into a variable called Total Carbon (TC).

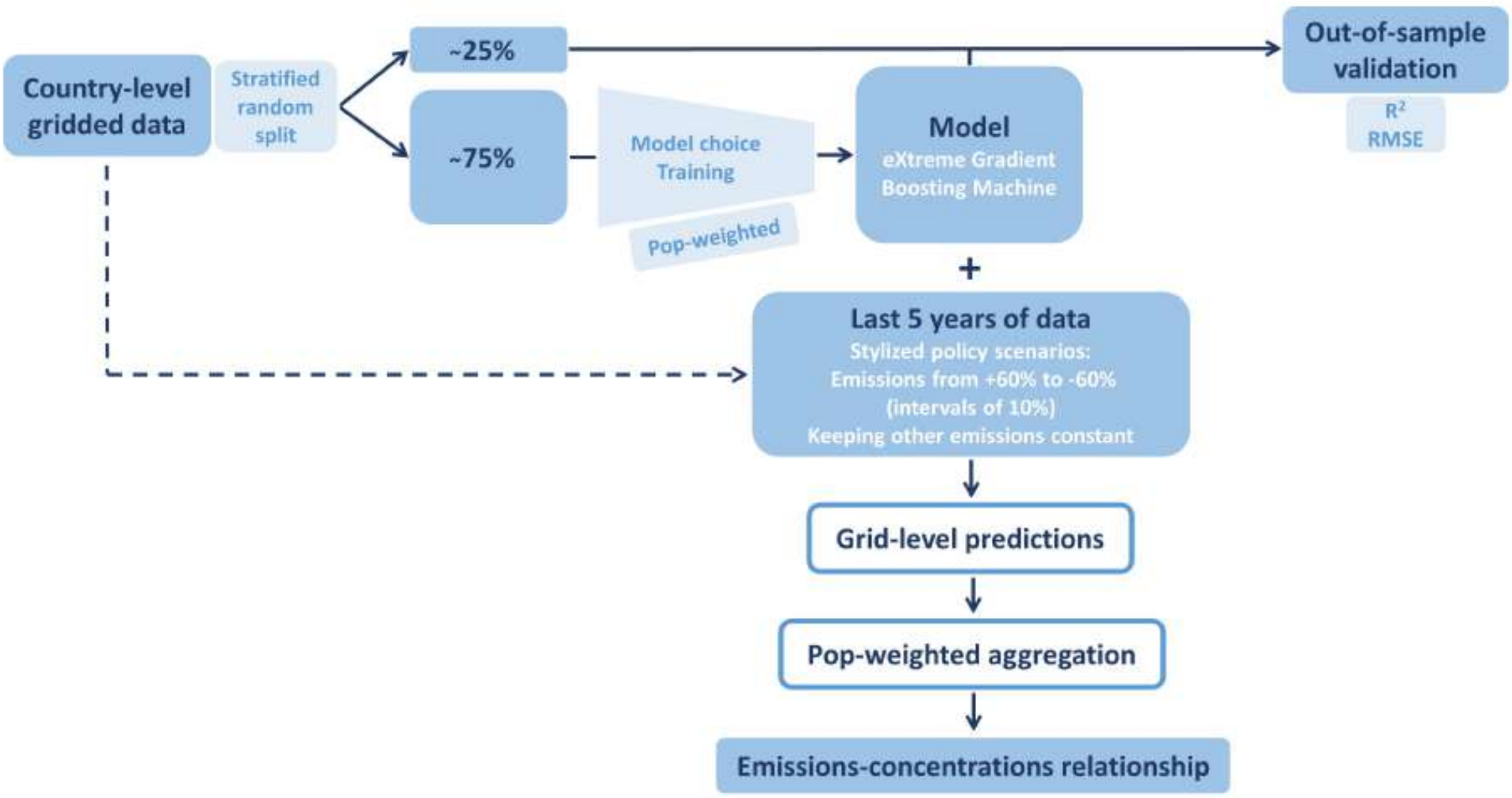

**Figure 5:** Visual representation of the extreme gradient boosting regressor models workflow.

2. Emissions from the Other sector are excluded. These emissions are frequently missing or otherwise highly correlated with other emissions. Moreover, their informative content is very low.
3. In addition to year and month-of-the-year, we include an identifier of grid cells as a predictor variable.

We use extreme gradient boosting regressor (Chen & Guestrin, 2016), a tree-based algorithm that has been shown to perform very well in supervised tasks with structured data (*e.g.*, Ma et al. (2020) in the context of air pollution).

The process is represented in Figure 5 and is as follows.

1. For each country-pollutant pair, the input data for ML are a grid panel dataset, composed of $N$ grid cells and observed over $T$ time periods. For each grid cell, we randomly assign the T observations (a time series) to the train or test set. Hence, we stratify by grid cell and randomization occurs over the temporal dimension. This stratified randomization ensures equal spatial representation in both data sets. Given unobservable but time-constant characteristics of cells (such as topography) and the desire for equal spatial representation, we prefer this method to simple randomized allocation, which might allocate the entire time series for a cell to either set. We use three-fourths of the data as the training set and the remaining fourth as the test set. As for EN models, we conduct sensitivity tests on the train-test splitting ratios, achieving consistent results across splits. For more details, refer to section A.5.
2. We train the model on the training set.
3. We evaluate its performance on the test set. We report the out-of-sample $R^2$ and RMSE, calculated as in Equations 6 and 7.
4. We derive emissions-to-concentrations relationships from the extreme gradient boosting algorithm in a fashion similar to partial dependence plots (J. H. Friedman, 2001). Section 4.1 describes this step in more detail.

For ML models, we include an identifier of grid cell as an input variable, similar to what cell fixed effects would be in a regression framework. This increases the fit of models to geographical variation in emissions, concentrations, weather, and their interactions, especially in emissions scenarios that are not excessively different from the baseline. For instance, recurrent transboundary pollution can be modeled by the interaction of cell identifiers and months. The improvement in geographic precision might come at the cost, however, of more bias in the case of extreme perturbations. For robustness, we also estimate the models without the identifier.

## 3.5. Method comparison

We summarize in Table 2 the advantages and disadvantages of the methods used in the CLAQC framework. While the elastic net models do not perform well using pixel-detail data and use country-level aggregate data instead, for most of the countries, the Gradient boosting regressor method delivers reasonable results with high-resolution inputs increasing the statistical power. The pixel-based approach allows for flexible spatial aggregation, although we only discuss the country-level spatial resolution here.

| Elastic net | Extreme gradient boosting regressor |
|---|---|
| Simple equation | Non intelligible form |
| Country-level emission totals allow for direct and fast application but trade off flexibility in spatial aggregation | Flexible regional aggregation |
| Moderate sensitivity to emission changes | Low sensitivity to strong emission changes |
| Assume that historically correlated sectors will remain correlated | |

**Table 2**
Comparison summary between CLAQC framework methods.

# 4. Discussion

## 4.1. Model results — Emission scenarios

We simulate perturbations in emissions to simulate hypothetical policy scenarios. Separately for every precursor, we perturb emissions by a factor P and predict concentrations under the average monthly weather of the 5 most recent years (2017-2021). Monthly predictions are then averaged to yearly ones and, for ML, from grid-level to country-level predictions. The process is performed with perturbation from +60% to -60% at intervals of 20%. While policies generally aim to reduce emissions, including emission-increase scenarios is crucial for a comprehensive understanding of potential air quality outcomes of a wide range of possible future conditions. For example, the persistent investments in coal in India, or the investment in gas fracking. It is important to showcase that these policy interventions may lead to exposure increases. For these reasons, we have looked at decreases and increases in emissions.

We consider model predictions as baseline predictions, $\widehat{Concentrations}_{Baseline,y}$.

The predicted relative change in concentrations for a perturbation $P$ of emissions of precursor $p$ in sector $s$ is:

$$\%\Delta\widehat{Concentrations}_{A,y,s,p} = \frac{\widehat{Concentrations}_{A,y,s,p} - \widehat{Concentrations}_{Baseline,y,s,p}}{\widehat{Concentrations}_{Baseline,y,s,p}} \cdot 100 \quad (11)$$

Given that the model algorithms may include multiple emission variables within a sector, *e.g.*, both $NO_x$ and BC emissions from the Road sector, to account for the sectoral range variability we calculate the minimum and maximum annual percentage variation in predictions from perturbed emissions by perturbation and sectoral level (Equations 12 and 13).

$$\min_{A,s}\left(\Delta\widehat{Concentrations}_{A,y,s,p}\right) \quad (12)$$

$$\max_{A,s}\left(\Delta\widehat{Concentrations}_{A,y,s,p}\right) \quad (13)$$

Figures 6 and 7 plot the relative variation in annual predicted concentrations of $PM_{2.5}$ and $O_3$, respectively, against perturbations by sector ranging from -60% to +60% in the selected major economies or populous countries: Brazil (BRA), Germany (DEU), Egypt (EGY), Italy (ITA), Mexico (MEX), Nigeria (NGA), Saudi Arabia (SAU), Turkey (TUR), the United States (USA), and South Africa (ZAF). Sectors are color-coded. To consult other countries' results, refer to the CLAQC online tool at `https://datashowb.shinyapps.io/CLAQC-App/`.

In general, EN and ML models detect approximately the same number of sectors. However, the sectors selected vary according to the method. Moreover, EN models show greater variability in predictions compared to ML ones. This is expected as linear methods are less accurate, based on only a single estimator per predictor and not fully capturing non-linear relationships.

Regarding $PM_{2.5}$, in 8 out of 10 EN models from Figure 6, the Other sector is selected, affecting predictions the most in the USA, ZAF, NGA, and EGY. In ITA, DEU, the USA, and NGA, Agriculture plays a relevant role as well. Also, the Road and Residential sectors emerge as relevant in contributing to particle formation, though we find the greatest impacts in DEU, ITA, and BRA. Such sectors are often detected in ML as well.

In the case of $O_3$ predictions, positive perturbations may lead to a decrease in predicted concentration levels: this happens because $NO_x$ consumes $O_3$ in $NO_x$-rich regimes, such as for DEU's Road sector in the ML model. The industrial sector is picked up in both EN and ML models, particularly for EN models in ITA, EGY, MEX, USA, SAU, and BRA. In ML models, the agricultural sector is often associated with major variations (TUR, NGA, ITA, USA, and MEX). The Power sector is another relevant sector, appearing in all selected countries.

Mainly, we find that EN models are good for predicting the total mass of ambient pollutants, while for some countries they are not reliable for sectoral attribution. Therefore, in such a case, we suggest opting for OLS coefficients to attribute sector shares to pollutant totals (see section A.2.3 in the Appendix for further details).

See section A.5 in the Appendix to consult sensitivity tests on the train-test splitting ratio, and sections A.7.1 and A.7.2 to consult other model specifications' results.

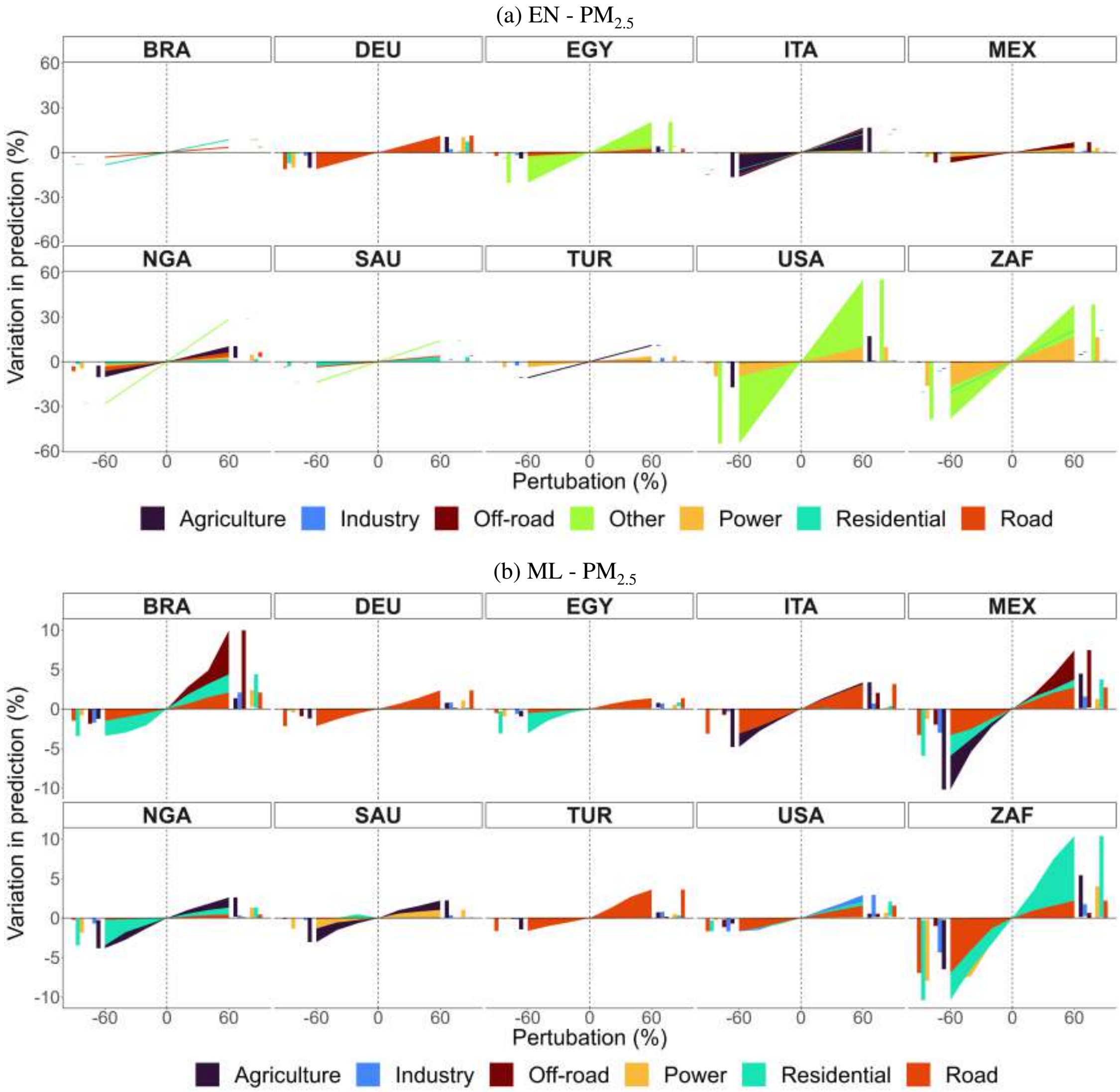

**Figure 6:** Percentage variation in predicted concentrations by sector and perturbation for selected countries in EN and ML models for weighted $PM_{2.5}$. Bar charts on the sides of each subplot help visualize overlapping variations.

## 4.2. Model internal validation results

A key aspect of predictive model evaluation is to verify if the models can reproduce events well based on past trends. We present out-of-sample validation for both methods while we perform validation against similar tools, *i.e.*, ECLIPSE-GAINS and TM5-FASST, for EN models only (for the latter one, see Appendix section A.6).

Figures 8 and 9 map the out-of-sample $R^2$ and RMSE for EN and ML models obtained from both CAMS and DACCIWA emissions. We do not advise using the models for countries with $R^2$ smaller than 0.5 or RMSE higher than 12.

Results vary by prediction target ($PM_{2.5}$ vs $O_3$) and input type (e.g., CAMS vs DACCIWA emissions), which may reflect inconsistencies in the emission or concentration data. Additionally, local factors such as unique orography and micro-meteorological conditions can significantly impact predictions in some areas, even country-level averages. Both elastic net and machine learning models are generally better at predicting $O_3$ than $PM_{2.5}$ as the former is highly correlated with incoming radiation or temperature, while predicting $PM_{2.5}$ is more challenging due to its complex secondary chemistry, local sources and particle composition. Chemistry transport models predict better $O_3$ than PM as well, due to the more complex mixture of particles and local effects from more sources of the latter one (Guérette et al., 2020).

Among the EN models with CAMS emissions for $PM_{2.5}$, 13 countries have an $R^2$ below 0.5, and 40 have RMSE above 12. Only 3 countries have an $R^2$ below 0.5, and 4 have RMSE above 12 for $O_3$. EN models for $PM_{2.5}$ with DACCIWA emissions perform comparably; $R^2$ is smaller than 0.5 in 4 countries, and RMSE is greater than 12 in 19. All EN models for $O_3$ with DACCIWA emissions have $R^2$ above 0.5 and RMSE below 12.

The ML models without pixel identifier perform poorly in 10 and 2 countries regarding $R^2$ for $PM_{2.5}$ and $O_3$, respectively; while we find an RMSE above 12 in 21 countries for $PM_{2.5}$, and none for $O_3$. As in the EN models, $PM_{2.5}$ predictions appear to be less accurate than those for $O_3$.

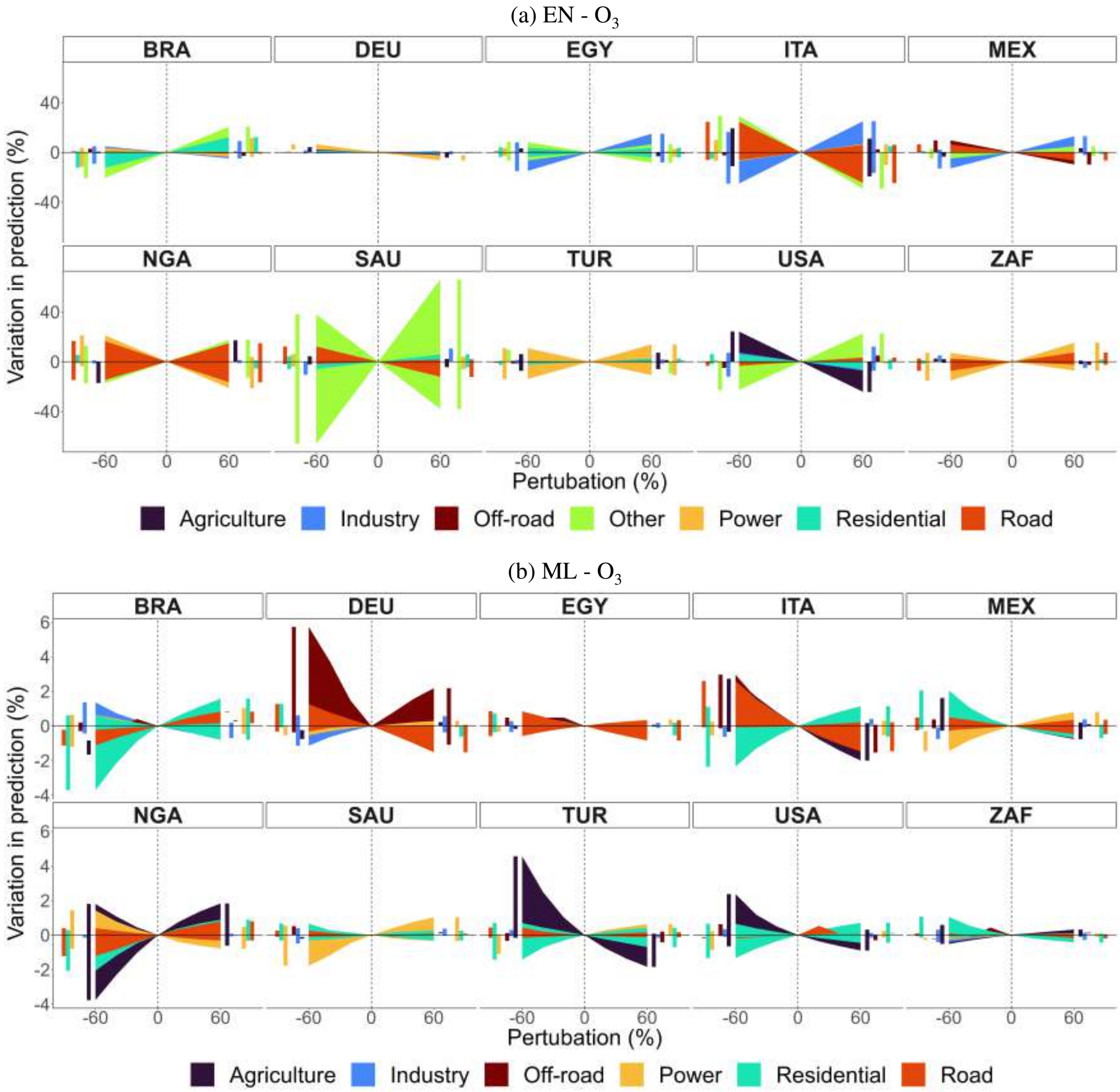

**Figure 7:** Percentage variation in predicted concentrations by sector and perturbation for selected countries in EN and ML models for $O_3$. Bar charts on the sides of each subplot help visualize overlapping variations.

While the ML models from DACCIWA emissions perform better in predicting both $PM_{2.5}$ and $O_3$ than the EN models. In general, ML models with grid cell identifiers perform better than those without them both in terms of explained linear variation and error. For more details on the ML models with grid cell identifier, see in the Appendix the A.3.1 section, while on the validation metrics of the other model specifications, see the A.3.2 section.

## 4.3. Model selection

Having estimated multiple models that rely on different algorithms and data inputs, we set out to select the ones with the best predictive power.[6] We propose a systematic model selection based on two criteria:

[6]More specifically, we consider 4 possible model specifications for elastic net and 4 for machine learning: EN × {CAMS emissions, DACCIWA emissions} × {with month fixed effects, without month fixed effects}; and ML × {CAMS emissions, DACCIWA emissions} × {with grid identifier, without grid identifier}

- The model error/reliability, measured by out-of-sample $R^2$ and RMSE.
- Reliability of emissions input data. DACCIWA is preferred to CAMS as a source of data over Africa only as the former is developed with more consistent methods, which prefer *in-situ* measurements as opposed to large data proxies and source profiles. Where DACCIWA is available, we assign a $Source$ score of 1 to models using DACCIWA and 0 to models using CAMS. Where DACCIWA is unavailable, $Source$ takes the value 0.

We re-scale all the elements of our decision criteria between 0 and 1, with 1 being the maximum score. We obtain the ensemble score, $s_v$, by weighting each of the criteria as in Equation 14. For each country, we choose the model that maximizes (Equation 14).

$$\frac{1}{6} * R^2 + \frac{1}{6} * RMSE + \frac{1}{3} * Source \tag{14}$$

We note that other weighting criteria for performance decisions are possible. Figure 10 shows which model maximizes each criteria ($R^2$, RMSE, $Source$). As expected, elastic net models have greater predictive power with month-fixed effects than without. Similarly, machine learning models perform better with pixel identifiers. On a general level, machine learning models perform better than elastic net models in Europe; in Africa, the two methods share the map; while the elastic net one is preferable in the remaining regions. We note that, by construction, DACCIWA is always preferable to CAMS as a source of input data over Africa only. To clarify further, we discuss the case of South Africa. The $PM_{2.5}$ results in Figure 10 show that the ML model (pixel version) using DACCIWA emissions maximizes $R^2$ (panel a). For RMSE, the EN model with DACCIWA emissions achieves the best performance (panel b), and DACCIWA emissions are identified as the preferred input source (panel c). For $O_3$, the EN model using CAMS-GLOB-ANT emissions maximizes $R^2$ (panel a), while the one with DACCIWA emissions minimizes RMSE (panel b). In terms of input preference, DACCIWA remains the preferred emission input (panel c).

Figure 11 maps instead the model specifications that maximize the composite criteria of Equation 14. With some exceptions, the patterns highlighted in Figure 10 are replicated.

All the maps and country-specific model results can be explored through the CLAQC-App web tool.

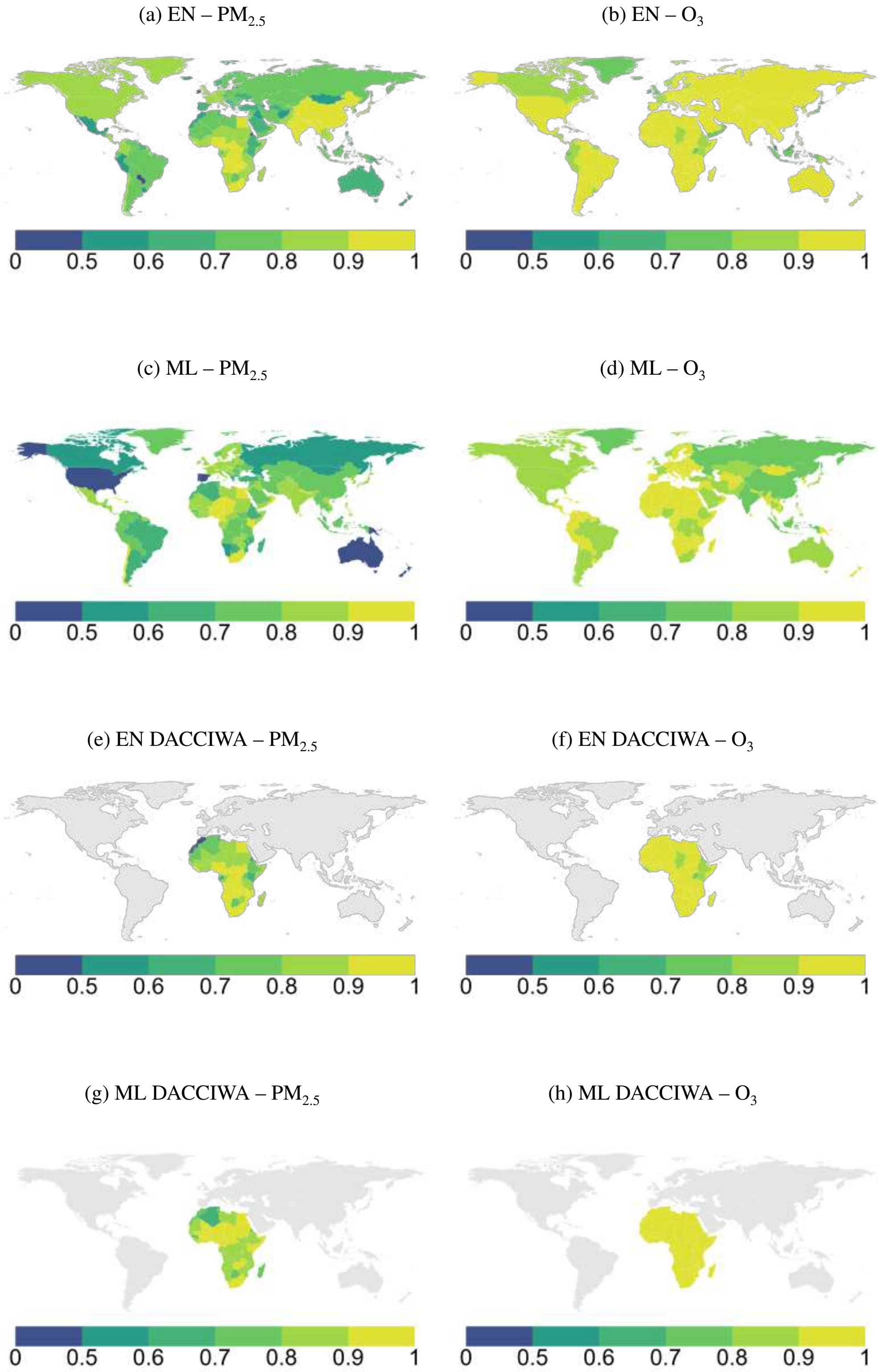

**Figure 8:** Out-of-sample performance metrics of ML (no pixel) and EN models (both from CAMS and DACCIWA data) as in Equations 8 and 9 under section 3.3: $R^2$.

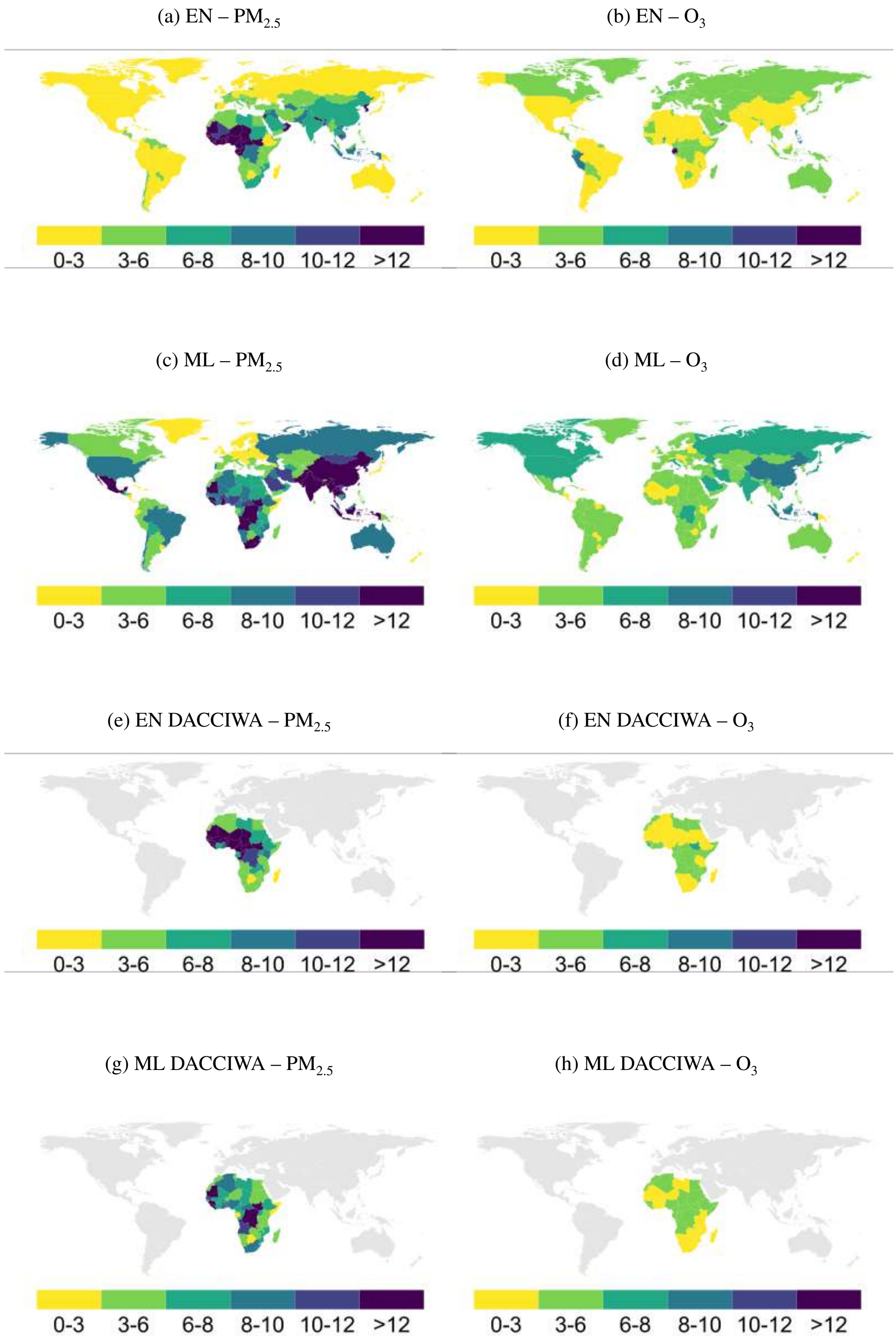

**Figure 9:** Out-of-sample performance metrics of ML and EN models (both from CAMS and DACCIWA data) as in Equation 8 and 9 under section 3.3: RMSE.

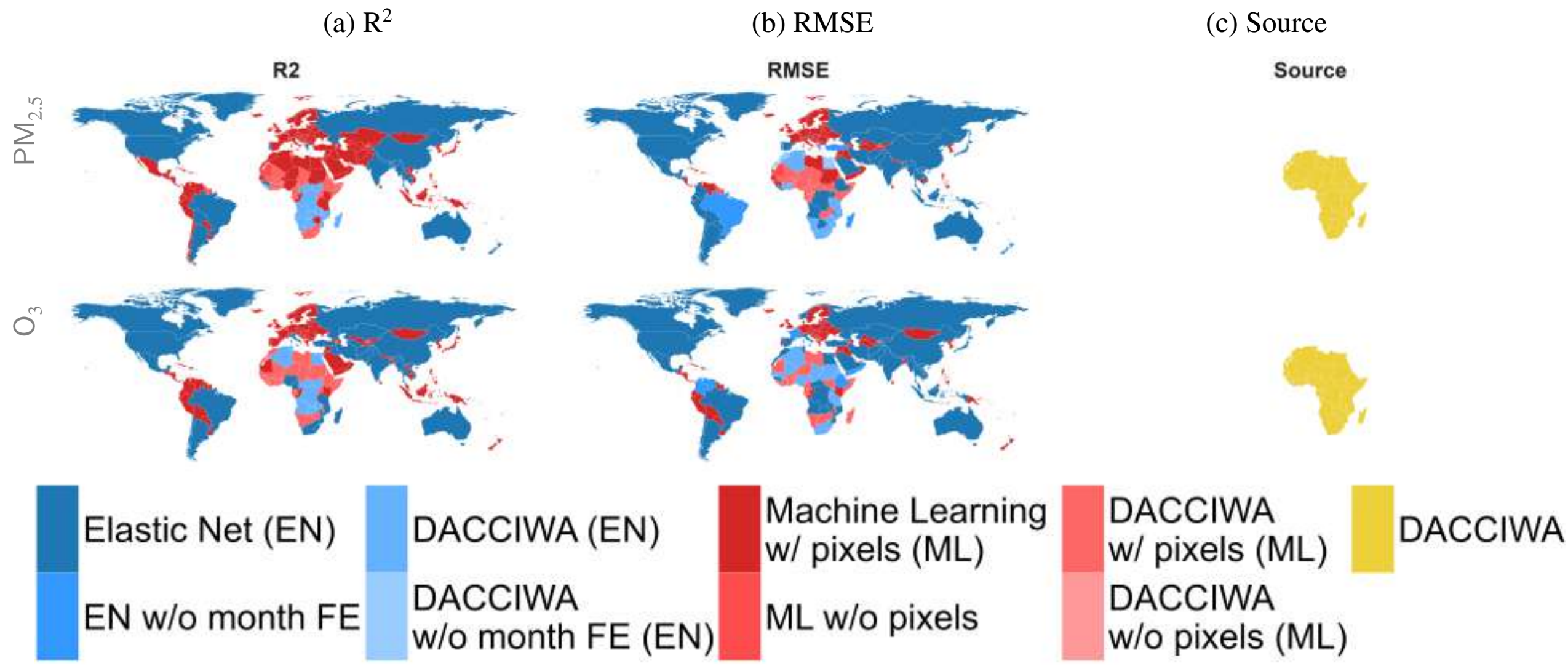

**Figure 10:** Best model score for each pollutant, country, and decision criterion.

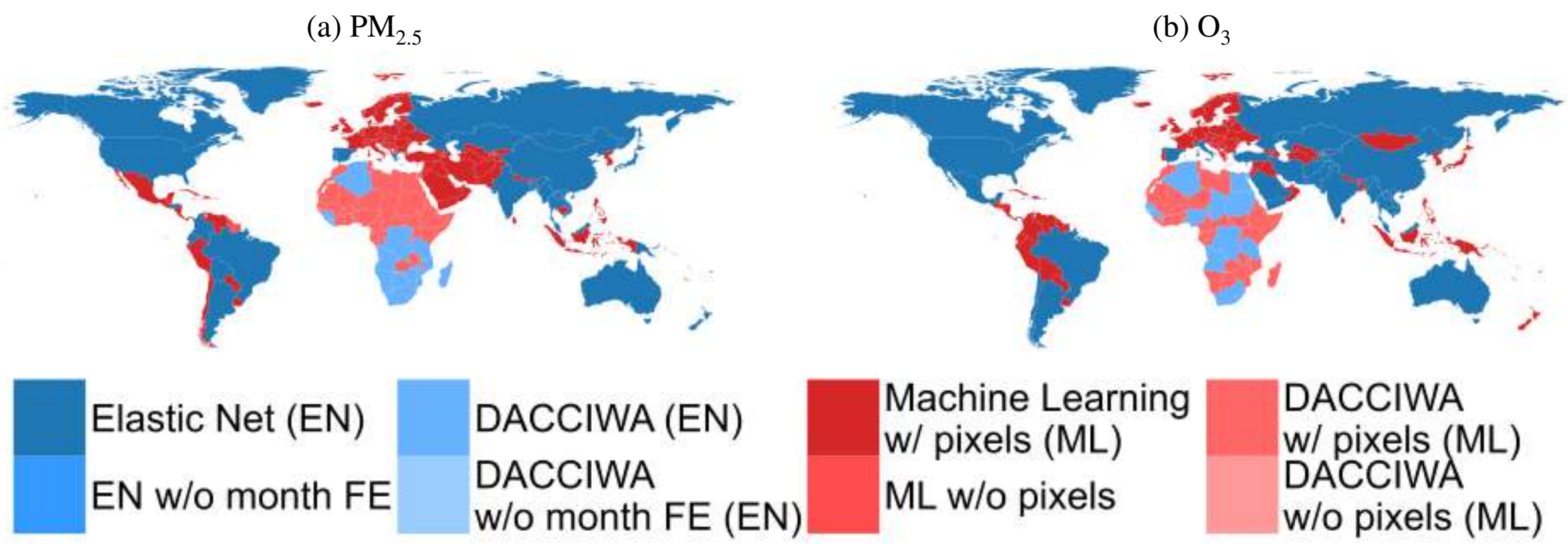

**Figure 11:** Best model score for each pollutant and each country.

# 5. Limitations

The CLAQC framework is very ambitious in terms of detail. However, it is important to note the limitations stemming from the input data, model characteristics, and specifications.

First, results are highly dependent on input data availability and quality. Using gridded data at different resolutions requires harmonization on a common global grid. As data values are estimated at locations that were not originally measured in the raw data, the interpolation process can introduce measurement errors of unknown distribution. We source data for both emissions and concentrations from the CAMS services. This choice allows for a smoother integration. However, biases in the emissions data set may propagate to the concentration ones and vice-versa. Additionally, while reanalysis is considered a state-of-the-art and complete method, it will surely not yield the same results as having institutionally approved ground monitoring stations in each grid cell, as it involves the use of data assimilation and CTM extrapolation to regions without ground or satellite monitoring. However, given the high disparities in the available ground monitoring data across the globe, we believe that CAMS reanalysis products, such as CAMS-GLOB-ANT and EAC4, are the next state-of-the-art available solution for these regions. Finally, we note that the model predictions have been evaluated with the support of observed levels of emissions and concentrations. Although we find a good out-of-sample model performance on annual concentration levels, caution is needed for the extrapolation of extreme perturbations. Furthermore, we did not analyze the reasons behind countries' poor performance, which limits our ability to interpret differences in results. In general, the model should be used under the 'fit-for-purpose' principle, i.e., for country-level policy roll-out purposes.

Second, the COVID-19 pandemic provided disruptive emission perturbations that are of key importance for this model. However, they represent only one set of large perturbations, which may differ from the real-world implementation of country-by-country policies and may differ spatially, meteorologically, or seasonally.

Third, sectoral emissions include a very large variability across publicly available databases. The results presented here are therefore sensitive to these uncertainties. If the baseline sectoral emission distribution used as input into CLAQC is substantially different from the CAMS baseline sectoral emissions, we recommend a rescaling of the total pollutant emissions to the CAMS sectoral emission profile.

Fourth, the EN model estimators may present higher variability as they rely on fewer observations relative to the ML models.

Fifth, as CLAQC was built to support policy impact evaluation, the approaches presented here do not explicitly model transboundary movements of pollution and biogenic emissions such as desert dust and sea salt. Only averages enter the models through the time and place identifiers (month-fixed effects in EN and grid identifiers in ML). Secondary inorganic aerosols (SIA) are also not directly modeled. Instead, their effects are approximated through interactions among emissions' predictors. In addition, we have not performed sensitivity analyses to assess the impact of excluding SIA on the final estimates.

# 6. Conclusions

We have developed CLAQC, a tool that provides fast simulations of emissions changes with national and sub-national resolution and global coverage. Based on statistical methods, it aims at supporting policy assessments in a timely fashion. The user sets the emission reduction for a given precursor from a given sector (or a combination thereof), and CLAQC simulates the implied change in concentrations of $PM_{2.5}$ and $O_3$. A possible application is, for instance, the calculation of co-benefits of climate policies. CLAQC can also be a tool for the scientific community and complement instruments like IAMs.

CLAQC is grounded on two different methods that trade off transparency and predictive performance. Both methods perform well, with predictive performance above reasonable levels for most countries. Elastic net models generally well estimate total annual exposure, although they are less reliable for pinpointing the contributions of individual sectors. For such a task, the machine learning approach should be preferred.

The CLAQC framework lends itself to multiple developments. It is a complementary tool to the modeling and policy scenario community, providing empirically based estimates and added value for global scale sectoral and country-level analyses. Its dynamic architecture makes it simple to update with more recent data, and the framework can be extended to both new data sources and methods. One potential evolution is to transform it into an ensemble model to enhance its accuracy, robustness, and reliability.

# Software availability

Software name: CLAQC v1.0;
Contact address: stefania.renna@cmcc.it;
Program language: R, Python;
Website: https://zenodo.org/records/14177055

# Declaration of interest

☒ The authors declare that they have no known competing financial interests or personal relationships that could have appeared to influence the work reported in this paper.
☐ The authors declare the following financial interests/personal relationships which may be considered as potential competing interests:

# Acknowledgments

The authors would like to thank Dirk Heine, Simon Black, Martin Heger, and Christian Schoder, who were part of the World Bank when this work started, for their collaboration and comments. They acknowledge and thank the RFF-CMCC European Institute on Economics and the Environment (RFF-CMCC EIEE) and Bocconi University for providing the logistical platforms to perform this work and its collaborators for all the useful comments and advice; the Emissions of Atmospheric Compounds and Compilation of Ancillary Data (ECCAD-AERIS) portal for the archiving and distribution of the emission and CONFORM data; the Tropospheric Ozone Assessment Report (TOAR) initiative for providing the

surface ozone data and analyses shown in Schultz et al. (2017); the International Institute for Applied Systems Analysis (IIASA) for providing ECLIPSE and GAINS data; and the Joint Research Centre (JRC) for TM5-FASST data. This work presents prediction models generated using modified Copernicus Atmosphere Monitoring Service information [2003-2021], downloaded from the Copernicus Atmosphere Monitoring Service (CAMS) Atmosphere Data Store (ADS), and from the Copernicus Climate Change Service (C3S) information [2003-2021], Climate Data Store (CDS) Catalogue, and using TerraClimate meteorology data. Neither the European Commission nor the ECMWF is responsible for any use that may be made of the Copernicus information or data it contains.

# Financial support

This work was supported by the World Bank. This project has received funding from the European Union's Horizon Europe research and innovation programme under grant agreement No 101069880 - AdJUST, and from the European Union - Next Generation EU, in the framework of the project GRINS - Growing Resilient, INclusive and Sustainable project (GRINS PE00000018 – CUP C93C22005270001).

# Review statement

# Code and data availability statement

All data sets used in CLAQC applications are freely available online. The modeling scripts and the output data sets are openly downloadable at `https://zenodo.org/records/14177055` (Renna, Granella, Aleluia Reis, & Schulz-Antipa, 2024).

# A. Appendix

## A.1. Multicollinearity

Differently from concentrations, emissions of pollutants and precursors are not always directly measured, but they can also be inferred using activity data and highly detailed emission factors. Emission data display a high correlation, even within grid cells, plausibly attributable to the correlation of emissions of pollutants within sectors; and in economic activity across sectors. Figure A.1 displays the cross-pollutant correlations within sectors.

## A.2. Elastic net models

### *A.2.1. Method description*

Shrinkage regression methods, such as elastic net, were developed to tackle some OLS limitations, in particular concerning the model interpretation and prediction accuracy. In OLS, the linear equation coefficients are estimated by minimizing the sum of squared residuals. Though, when there are many predictors, OLS models generally show high variance and unstable coefficients. The elastic net method minimizes such variance. In fact, shrinkage regression may improve prediction accuracy by either shrinking regression coefficients towards zero or setting them to zero, or both. However, a trade-off is produced: as the variance is reduced, the bias may increase. In this case a bias toward more conservative outcomes. Moreover, in the OLS approach, when a large number of predictors is present, it may not be straightforward to identify those representing the most relevant influence. In CLAQC elastic net models, a penalization parameter lambda ($\lambda$) is introduced, in OLS this parameter is zero. In CLAQC, $\lambda$ is selected using cross-validation to minimize divergence, so that for each country the most optimized penalization parameter of the coefficients is identified. In such a procedure, predictors are also standardized in order to identify solutions that do not depend on the unit of measurement of the features. For further details, see Zou and Hastie (2005) and Hastie, Tibshirani, and Friedman (2009).

### *A.2.2. Elastic net models with DACCIWA emission data*

The EN linear regression models obtained using DACCIWA emission data (Keita et al., 2021, 2017) take the following form for each country:

$$
\begin{aligned}
PM_{2.5t} = \alpha &+ \sum_{s,p_1} \beta_{s,p_1} E_{s,p_1,m} + \gamma_1 PPT_t + \gamma_2 TMIN_t + \gamma_3 TMAX_t + \gamma_4 VPD_t + \gamma_5 WS_t + \gamma_6 WD_t + \\
&+ \sum_{s} \delta_s E_{s,t} + \sum_{p_1} \lambda_{p_1} E_{p_1,t} + \xi E_{SO_2,t} \cdot E_{NO_x,t} + \sum_{s} \theta_s E_{s,t} \cdot WS_t \cdot WD_t + \phi_t + \varepsilon_t
\end{aligned} \tag{A.1}
$$

$$
\begin{aligned}
O_{3t} = \alpha &+ \sum_{s,p_2} \beta_{s,p_2} E_{s,p_2,t} + \gamma_1 PPT_t + \gamma_2 TMIN_t + \gamma_3 TMAX_t + \gamma_4 VPD_t + \gamma_5 WS_t + \gamma_6 WD_t + \\
&+ \sum_{s} \delta_s E_{s,t} + \sum_{p_3} \lambda_{p_3} E_{p_3,t} + \mu E_{NOx,t} \cdot E_{NMVOC,t} + \nu E_{SO_2,t} \cdot E_{NMVOC,t} + \xi E_{SO_2,t} \cdot E_{NO_x,t} + \\
&+ \sum_{s} \theta_s E_{s,t} \cdot WS_t \cdot WD_t + \phi_t + \varepsilon_t
\end{aligned} \tag{A.2}
$$

where:

$$
\begin{aligned}
s &\in \{Transport,\ Power,\ Industry,\ Residential,\ Other\} \\
p_1 &\in \left\{BC,\ NMVOC,\ NO_x,\ OC,\ SO_2\right\} \\
p_2 &\in \left\{NMVOC,\ NO_x\right\} \\
p_3 &\in \left\{NMVOC,\ NO_x,\ SO_2\right\}
\end{aligned}
$$

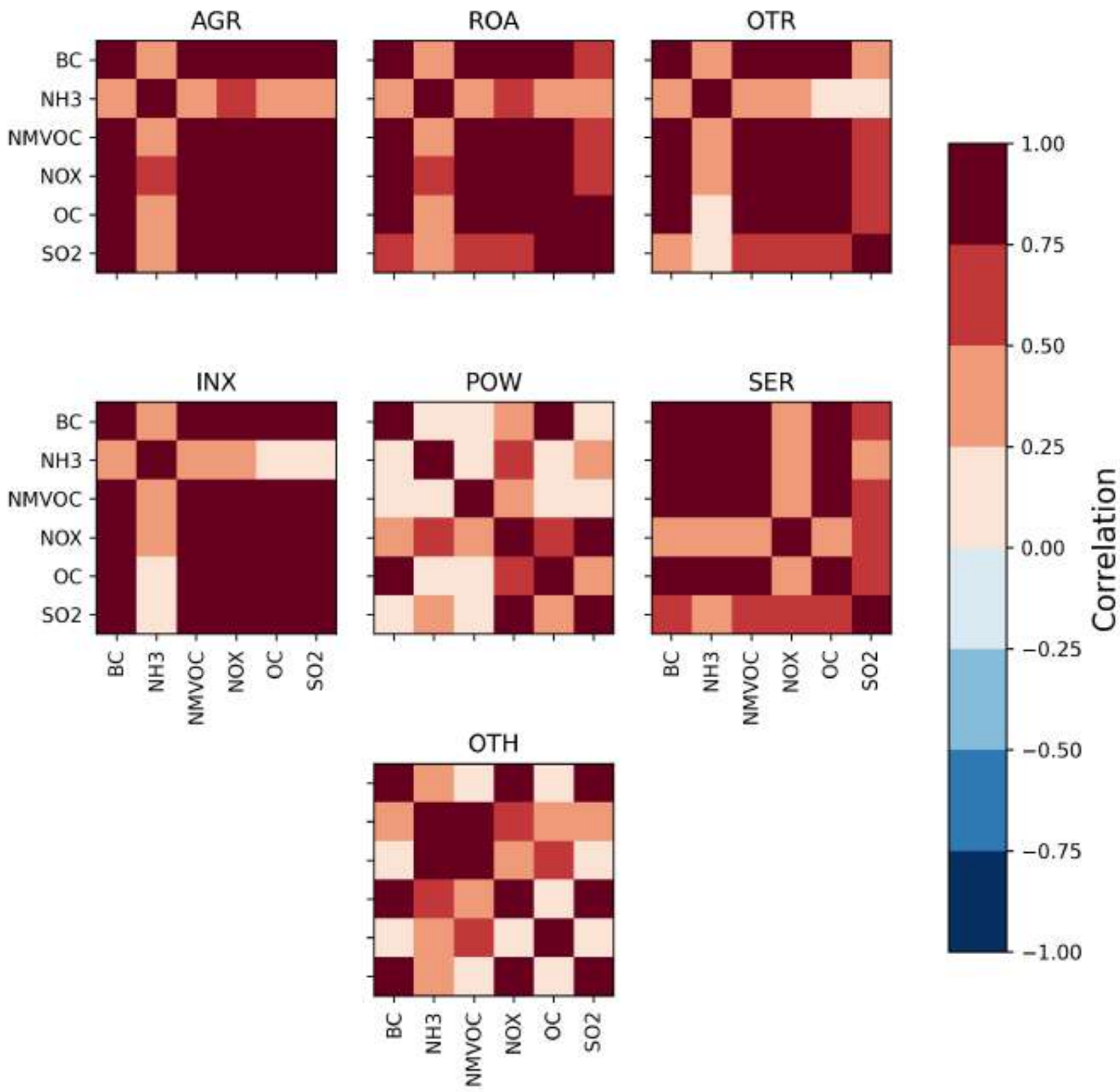

**Figure A.1:** Correlation matrix among CAMS emission predictors.

$PM_{2.5t}$ = Monthly concentration of $PM_{2.5}$ in $\mu g/m^3$ (population-weighted)
$O_{3t}$ = Monthly concentration of $O_3$ in $\mu g/m^3$
$E_{s,p_1,t}$ = Monthly emissions of sector $s$ and pollutant $p_1$ in $kg$
$E_{s,p_2,t}$ = Monthly emissions of sector $s$ and pollutant $p_2$ in $kg$
$E_{s,p_3,t}$ = Monthly emissions of sector $s$ and pollutant $p_3$ in $kg$
$PPT_t$ = Monthly accumulated precipitation in *mm*
$TMIN_t$ = Monthly minimum 2-m temperature in $°C$
$TMAX_t$ = Monthly maximum 2-m temperature in $°C$
$VPD_t$ = Monthly mean vapor pressure deficit in $kPa$
$WS_t$ = Monthly 10-m wind speed in $\frac{m}{s}$
$WD_t$ = Monthly wind direction in $degrees$
$E_{p_1,t}$ = Monthly composite index from the sum of total emissions of pollutant $p_1$ in $kg$
$E_{p_2,t}$ = Monthly composite index from the sum of total emissions of pollutant $p_2$ in $kg$
$E_{p_3,t}$ = Monthly composite index from the sum of total emissions of pollutant $p_3$ in $kg$
$E_{s,t}$ = Monthly composite index from the sum of total emissions of sectors $s$ in $kg$
$\phi_t$ = Monthly fixed effects
$\varepsilon_t$ = Error term

Unlike the Equations in 8 and 9 from section 2.1.4, in Equations A.1 and A.2, all $NH_3$ predictors[7], and Agriculture sectoral emissions are not present, while the Transport sector's predictors are not split into Road and Off-road transportation.

[7]Including interactions containing $NH_3$ totals.

$PM_{2.5}$ and $O_3$ concentration values, in $\mu g/m^3$, obtained from the models are country-level monthly concentration averages indexed by time $t$, as all the other parameters in the equation; emissions are in $kg$; weather variables' units of measurement are expressed as specified in section 3.3.

### *A.2.3. Sectoral attribution*

Given that in some cases elastic net results are not suitable for sectoral attribution, to tackle such an issue we run constrained OLS models with sector totals and other controls only. This way, elastic net results can be used to predict the total mass of our pollutants of interest, while OLS coefficients can be exploited to distribute concentration contributions by sector. We follow the procedure explained in section 3.3, with some modifications. In particular, in step 2 we first aggregate gridded sectoral emissions, $E_{s,p_{[n]},t}$, at the country, year, and month level and then normalize them to range from 0 to 1 (min-max normalization), as follows:

$$normE_{s,p,t} = \frac{E_{s,p,t} - Min_{s,p}}{Max_{s,p} - Min_{s,p}} \tag{A.3}$$

Where $normE_{s,p,t}$ are normalized monthly sectoral emission, $Min_{s,p}$ is the minimum emission level across months for pollutant $p$ and sector $s$, and $Max_{s,p}$ is the maximum. Finally, we sum them sector-wise by country, year, and month to obtain monthly sector totals, $normE_{s,t}$, as specified:

$$normE_{s,t} = \sum_{p,t} normE_{s,p,t} \tag{A.4}$$

In step 7 (model fitting), within the `glmnet` function, we set the 'lambda' parameter and all penalty factors related to the emission variables equal to 0, and set the threshold parameter for interrupting convergence to the solution, 'thresh', to $10^{-14}$ to get the OLS results. For the same reason, in step 8 (model evaluation), we set the penalty parameter lambda, 's', to 0.

For each pollutant, $poll_t$, the model specification only includes the total sectoral emissions, $E_{s,p_1,t}$, weather variables, and month fixed effects, $\phi_t$, and takes the following form:

$$poll_t = \alpha + \gamma_1 PPT_t + \gamma_2 TMIN_t + \gamma_3 TMAX_t + \gamma_4 VPD_t + \gamma_5 WS_t + \gamma_6 WD_t + \sum_s \delta_s E_{s,t} + \phi_t + \varepsilon_t \tag{A.5}$$

However, it is important to acknowledge that such OLS models may have limitations:

- Due to multicollinearity among certain sectors, it is likely that OLS models will not include all sectoral predictors;
- OLS models may introduce bias since relevant variables, such as biogenic emissions, interactions, non-linear terms, and others, are excluded from the model specification;
- Assuming non-linear relationships between sectoral emissions, weather conditions, and concentration levels, if only linear variables are used, the OLS models may incorrectly attribute sector shares;
- Mostly, OLS models will differ from elastic net models in terms of variable selection and coefficient estimation.

## A.3. Machine learning models

### *A.3.1. XGBoost additional remarks*

As stated in section 3.4, we consider two ML model specifications, with and without the grid cell feature. As expected, models with grid cell identifier perform better than those without it (Figure A.2). Figure A.3 compares the changes in concentrations predicted with and without grid cell identifier in the extreme scenario of 100% reduction in emissions. The results of both sets of models are similar. We thus prefer models with the identifier for their greater out-of-sample performance.

### *A.3.2. XGBoost models with DACCIWA emission data*

## A.4. Model implementation

The models presented in this framework follow different implementation procedures. While elastic net models can be implemented using a linear equation and temporal profiles, machine learning models can be implemented after emulation through a spreadsheet-style format.

### *A.4.1. Elastic net models*

We provide a monthly scheduling profile for each country to transform annual emissions into monthly emissions to be fed into the models. We build such a monthly schedule using the most recent years of emission data (2017-2021) for each couple of pollutant and sector, $E_{s,p}$, representing a reference monthly emission value by sector and pollutant. The monthly weights can be multiplied by the equivalent pollutant and sector total annual precursor emissions and then be used directly in Equations 8 and 9.

Additionally, we provide default meteorology fields that can be used in the models in case the input data is missing. The default meteorology variable fields are based on each country's average of the last 5 years of meteorology to represent current trends.

In health and crop impact assessments of air pollution due to $O_3$, other metrics, such as the 6-month warm season mean of daily maximum 8-hour average (6mDMA8), are more common. We provide a post-process data set that allows converting from annual average $O_3$ concentrations to a 6mDMA8 metric, obtained by using Tropospheric Ozone Assessment Report (TOAR) surface ozone data products. See Schultz et al. (2017) for more details.[8]

[8] Query instructions are available at https://join.fz-juelich.de/services/rest/surfacedata/stats/RestServicesHowTo

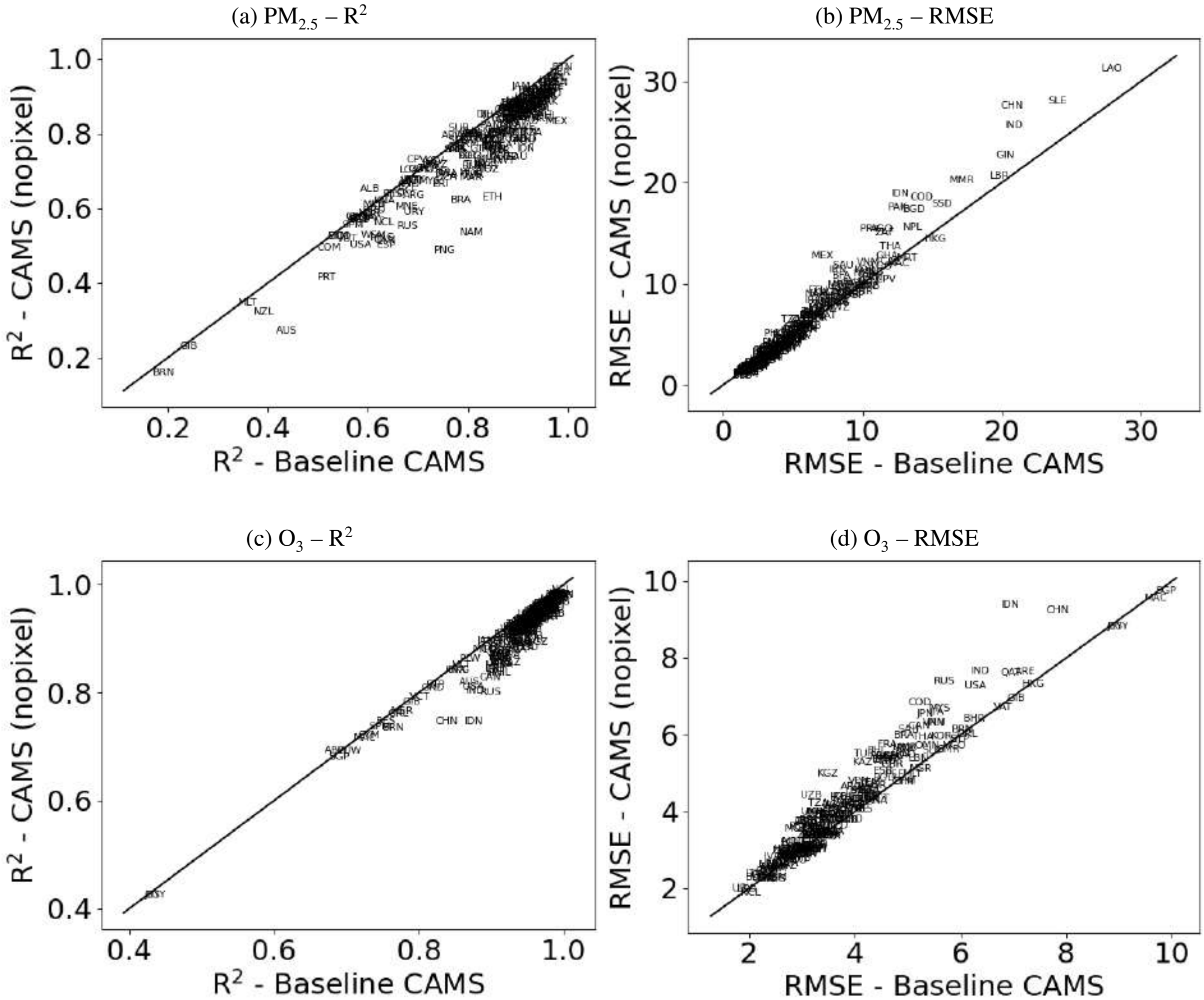

**Figure A.2:** Comparison of performance of models with (x-axis) and without (y-axis) grid cell identifier. Black lines indicate equality.

### *A.4.2. Machine learning — Guide to the Excel spreadsheet*

A machine learning model was built for every country and ambient pollutant ($PM_{2.5}$ and $O_3$), and an empirical emissions-to-concentrations relationship was constructed. The Excel spreadsheet `Results.xlsx` contains the data required to derive the changes and the levels of concentrations of pollutants under emissions scenarios supplied by the user. Each row is defined by the combination of country, pollutant, sector, and precursor. The goodness of fit of each country-pollutant model, as measured by out-of-sample $R^2$ and RMSE, is reported as well.

For easier implementation within a spreadsheet, the relationships have been approximated with piecewise linear functions that map perturbations of emissions to concentrations. A perturbation $P$ is the relative difference in emissions between the baseline scenario and a chosen scenario, expressed in 100 percentage points.

Omitting subscripts for the country and emitted pollutant for ease of notation, call $E_{Baseline,s,p}$ the baseline emissions from sector $s$ of precursor $p$ and $E_{A,s,p}$ the emissions under the alternative scenario $A$. Then, for every country, pollutant, sector, and precursor, the perturbation $P_{A,s,p}$ is

$$P_{A,s,p} = \left( \frac{E_{A,s,p}}{E_{Baseline,s,p}} - 1 \right) \cdot 100 \tag{A.6}$$

*Assuming all other emissions are constant*, the concentrations under the alternative scenario $A$ are

$$Concentrations_{A,s,p} =$$

.html#_Toc468805314

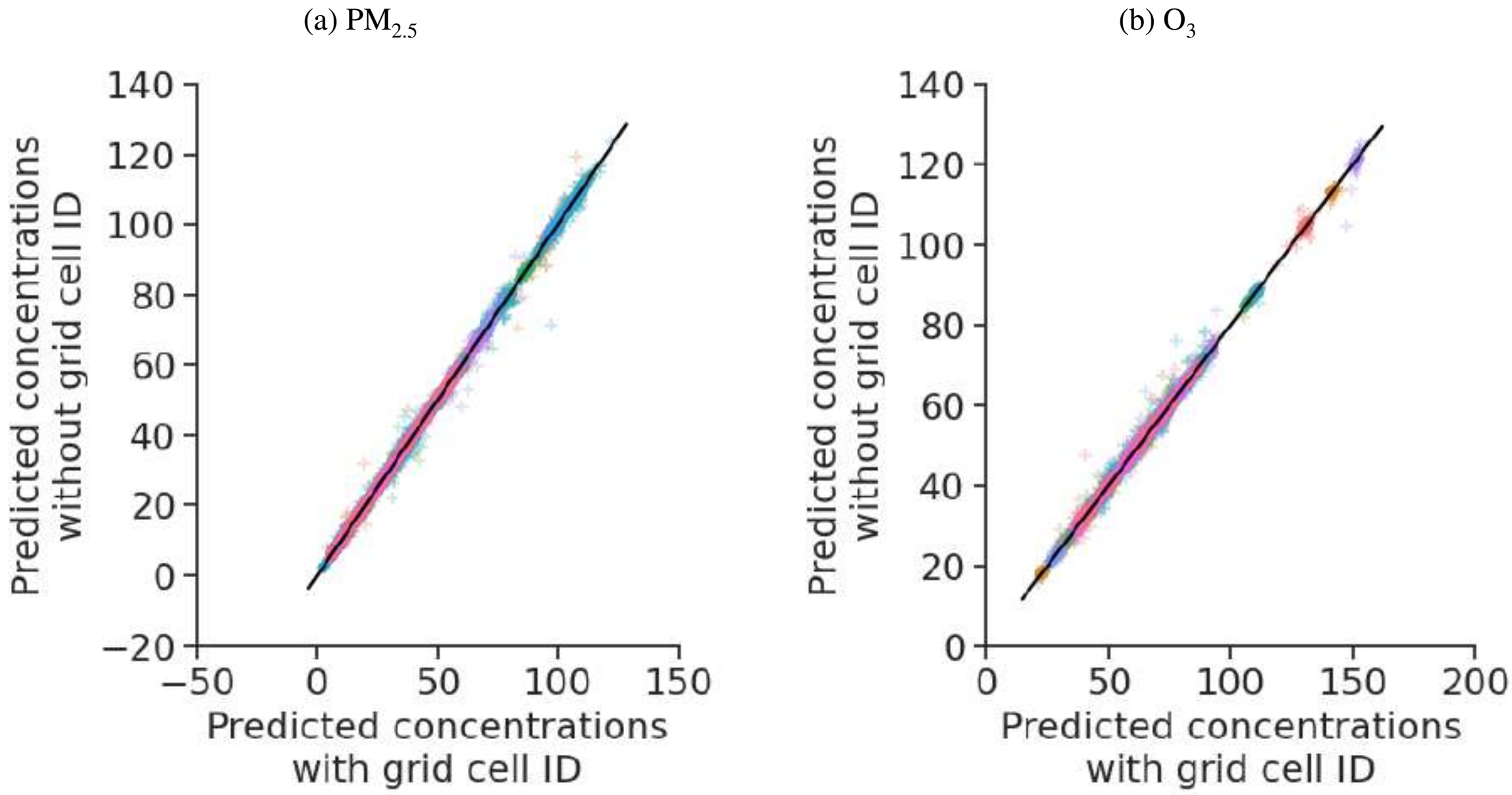

**Figure A.3:** Concentrations under the simulation of perturbations for models with and without grid cell identifier. Each cross is a country-sector-precursor-perturbation combination. Black lines indicate equality, colors indicate countries.

$$= \begin{cases} a_{-100,s,p} & \text{if } P_{A,p,t} < -100 \\ a_{-100,s,p} + b_{-100,s,p} \cdot \left(P_{A,p,t} + 100\right) & \text{if } -100 \leq P_{A,p,t} < -80 \\ a_{-80,s,p} + b_{-80,s,p} \cdot \left(P_{A,p,t} + 80\right) & \text{if } -80 \leq P_{A,p,t} < -60 \\ a_{-60,s,p} + b_{-60,s,p} \cdot \left(P_{A,p,t} + 60\right) & \text{if } -60 \leq P_{A,p,t} < -40 \\ a_{-40,s,p} + b_{-40,s,p} \cdot \left(P_{A,p,t} + 40\right) & \text{if } -40 \leq P_{A,p,t} < -20 \\ a_{-20,s,p} + b_{-20,s,p} \cdot \left(P_{A,p,t} + 20\right) & \text{if } -20 \leq P_{A,p,t} < 0 \\ a_{0,s,p} + b_{0,s,p} \cdot \left(P_{A,p,t}\right) & \text{if } 0 \leq P_{A,p,t} < 20 \\ a_{20,s,p} + b_{20,s,p} \cdot \left(P_{A,p,t} - 20\right) & \text{if } 20 \leq P_{A,p,t} < 40 \\ a_{40,s,p} + b_{40,s,p} \cdot \left(P_{A,p,t} - 40\right) & \text{if } 40 \leq P_{A,p,t} < 60 \\ a_{60,s,p} + b_{60,s,p} \cdot \left(P_{A,p,t} - 60\right) & \text{if } 60 \leq P_{A,p,t} < 80 \\ a_{80,s,p} + b_{80,s,p} \cdot \left(P_{A,p,t} - 80\right) & \text{if } 80 \leq P_{A,p,t} < 100 \\ a_{100,s,p} + b_{100,s,p} \cdot \left(P_{A,p,t} - 100\right) & \text{if } P_{A,p,t} > 100 \end{cases} \tag{A.7}$$

The coefficient $a_{j,s,p}$ is the level of concentrations when emissions of precursor $p$ from sector $s$ are perturbed by $j\%$. The coefficient $b_{j,s,p}$ is the slope of the piecewise function in the interval starting at $j$. The coefficients $a_{-100,s,p} \dots a_{100,s,p}$ and $b_{-100,s,p} \dots b_{100,s,p}$ are reported in the spreadsheet in columns L to AG. The coefficient $a_{0,s,p}$ is the value that the function takes when the perturbation is null. Thus, it is a generally close approximation of the concentration level given baseline emissions. Emissions are in $kg$, while concentrations of $PM_{2.5}$ are expressed in $\mu g/m^3$, and concentrations of $O_3$ are in 6mDMA8 $\mu g/m^3$. Baseline concentrations, in column H, are the average concentrations (over the entire country) from 2017 to 2021. In column I, baseline emissions are the average precursor emissions from a given sector over the same period.[9] Scenario emissions, in column J, are *set by the user*. The perturbation in column K is automatically computed. Concentrations under the alternative scenario are computed in column AK following Equation A.7. It should be noted that the calculation *assumes that only emissions of the row sector-precursor pair are perturbed*. All other emissions are assumed constant. The change in concentrations attributable to the perturbation $P_{A,s,p}$ is calculated in column AI as the difference between baseline concentrations and concentrations under the alternative scenario. Again, this is the change in concentrations assuming all other sectoral emissions are kept constant. The change is computed as follows:

$$\begin{aligned} \Delta\, Concentrations_{A,s,p} &= Concentrations_{Baseline} - Concentrations_{A,s,p} \\ &= a_{0,s,p} - \left[a_{j,s,p} + b_{j,s,p} \cdot (P_{A,s,p} - j)\right] \end{aligned}$$

[9] Averages are weighted by population in models for $PM_{2.5}$, but not in models for $O_3$.

$$= a_{0,s,p} - a_{j,s,p} - b_{j,s,p} \cdot (P_{A,s,p} - j) \tag{A.8}$$

where $P_{A,s,p}$ is inside an interval starting at $j$. When scenario emissions are set to zero, the change in concentrations gives the (opposite of the) estimated contribution of each sector-precursor to the total concentrations in 2017-2021. The approximation of the emissions-concentrations relationship functions is best for small and moderate perturbations and larger under scenarios of extreme perturbations. We suggest applying perturbing emissions only in the ±60% range based on the fitness-for-purpose principle and given the limitations discussed in section 5. To avoid that approximation error reverses the relationship between emissions and concentrations of $PM_{2.5}$, which is known to be positive, we impose in column P that negative perturbations cannot result in an increase in concentrations, and vice versa. The total change in concentrations under emissions scenario $A$ is computed in column AJ summing across sectors and precursors:

$$\underset{Country}{\Delta} Concentrations_{A,s,p} = \sum_{s,p} \Delta\, Concentrations_{A,s,p} = \sum_{s,p} a_{0,s,p} - a_{j,s,p} - b_{j,s,p} \cdot (P_{A,s,p} - j) \tag{A.9}$$

The level of concentrations under scenario $A$ is then reported in column AJ as:

$$Concentrations_A = Concentrations_{Baseline} + \underset{Country}{\Delta} Concentrations_{A,s,p}$$

It should be noted that, differently from the other columns, the total change in concentrations $\underset{Country}{\Delta} Concentrations_{A,s,p}$ (column AJ) and the level of concentrations $Concentrations_A$ (column AK) are invariant within a country-pollutant pair. Therefore, the same value appears in multiple rows.

### *Comparing two scenarios*

It is possible to compare concentrations in two scenarios in the following way. Consider two scenarios, $A$ and $B$. The difference in concentrations attributable to changes in precursor $p$ from sectors $s$ is:

$$\begin{aligned} \Delta\, Concentrations_{A,s,p} - \Delta\, Concentrations_{B,s,p} &= a_{0,s,p} - a_{j_A,s,p} - b_{j_A,s,p} \cdot (P_{A,s,p} - j) - \left[a_{0,s,p} - a_{j_B,s,p} - b_{j_B,s,p} \cdot (P_{B,s,p} - j)\right] \\ &= a_{j_B,s,p} + b_{j_B,s,p} \cdot (P_{B,s,p} - j) - a_{j_A,s,p} - b_{j_A,s,p} \cdot (P_{A,s,p} - j) \end{aligned} \tag{A.10}$$

Whereas the difference in the total change of concentrations (and the difference in levels of concentrations) is:

$$\begin{aligned} \underset{Country}{\Delta} Concentrations_{A,s,p} - \underset{Country}{\Delta} Concentrations_{B,s,p} &= Concentrations_A - Concentrations_B \\ &= \sum_{s,p} a_{j_B,s,p} + b_{j_B,s,p} \cdot (P_{B,s,p} - j) - a_{j_A,s,p} - b_{j_A,s,p} \cdot (P_{A,s,p} - j) \end{aligned} \tag{A.11}$$

*Example* All emissions set to zero in scenario $A$, set uniformly at 90% of baseline emissions in scenario $B$.

$$\begin{aligned} \underset{Country}{\Delta} Concentrations_{A,s,p} - \underset{Country}{\Delta} Concentrations_{B,s,p} &= Concentrations_A - Concentrations_B \\ &= \sum_{s,p} a_{-20,s,p} + b_{-10,s,p} \cdot (-10 + 20) - a_{-100,s,p} - b_{-100,s,p} \cdot (-100 + 100) \end{aligned} \tag{A.12}$$

## A.5. Train-test splitting sensitivity analysis

The training and test dataset splits differ between EN and ML models due to differences in their data samples. EN models are trained and tested on country-level aggregated data, while ML models use gridded country-level data, resulting in a larger sample size. Given these differences, we do not harmonize data splitting across methods. Instead, we ensure a sufficiently large training set for EN models to reduce variance in parameter estimates.

To validate model configurations more robustly, we conduct additional runs with varied train-test splits for models obtained from CAMS emissions: 75-25 and 70-30 for EN models, and 80-20 and 70-30 for ML models, alongside their original splits. Figures A.4 and A.5 present emission policy scenarios derived from EN models for $PM_{2.5}$ and $O_3$, comparing splits of 84-16, 75-25, and 70-30. Similarly, figures A.6 and A.7 compare policy scenarios for ML models using 80-20, 75-25, and 70-30 splits. These sensitivity analyses confirm that model predictions remain stable across different train-test splits, showing only minor variations.

(a) EN 84-16 (initial splitting) - $PM_{2.5}$

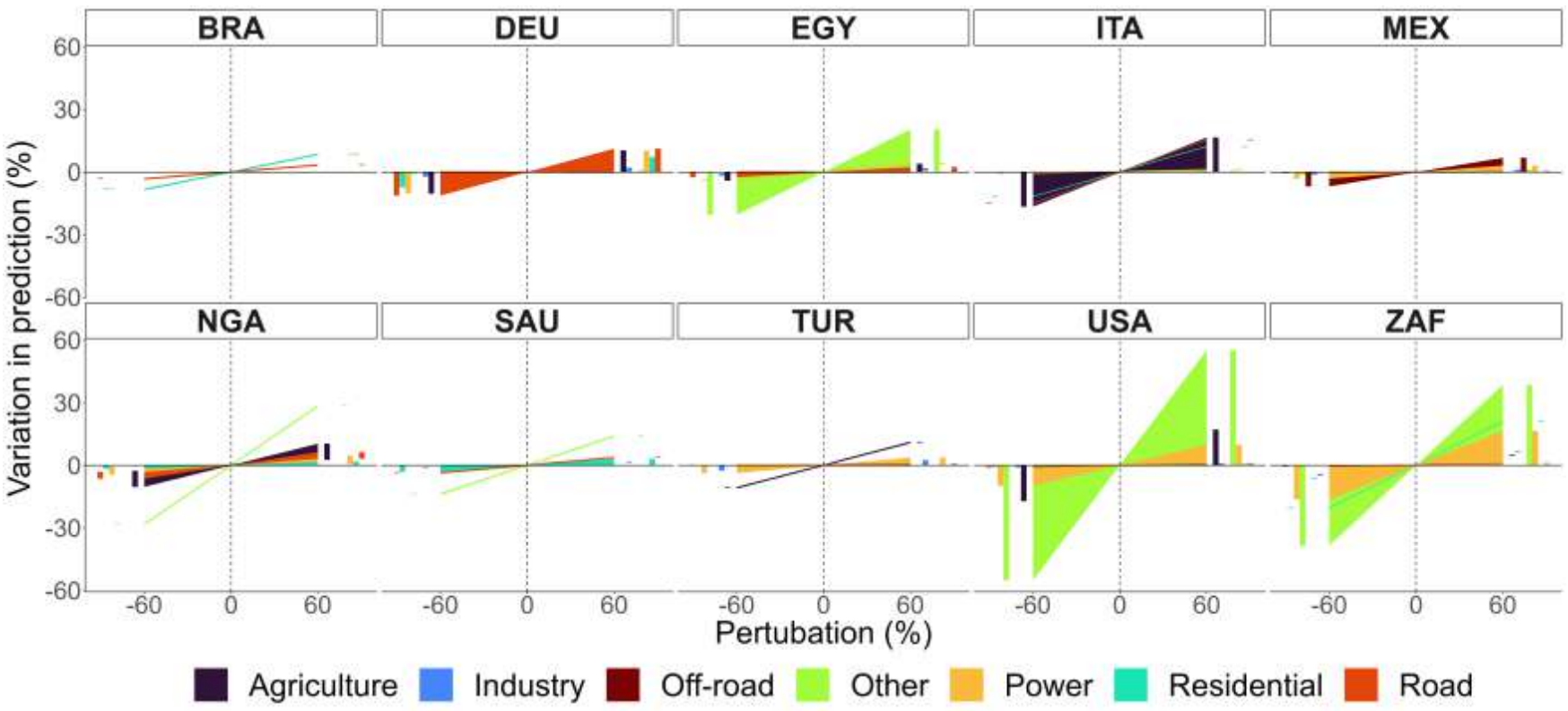

(b) EN 75-25 - $PM_{2.5}$

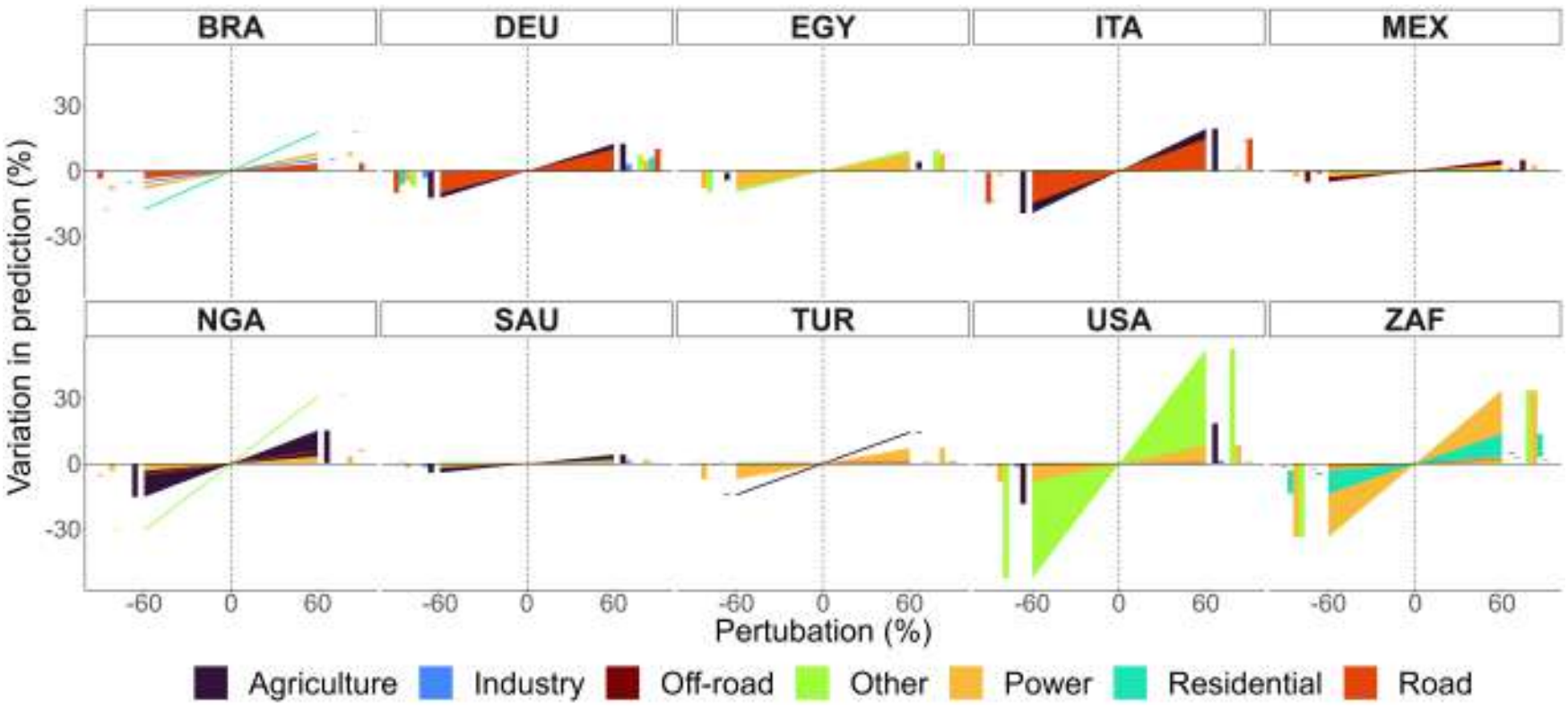

(c) EN 70-30 - $PM_{2.5}$

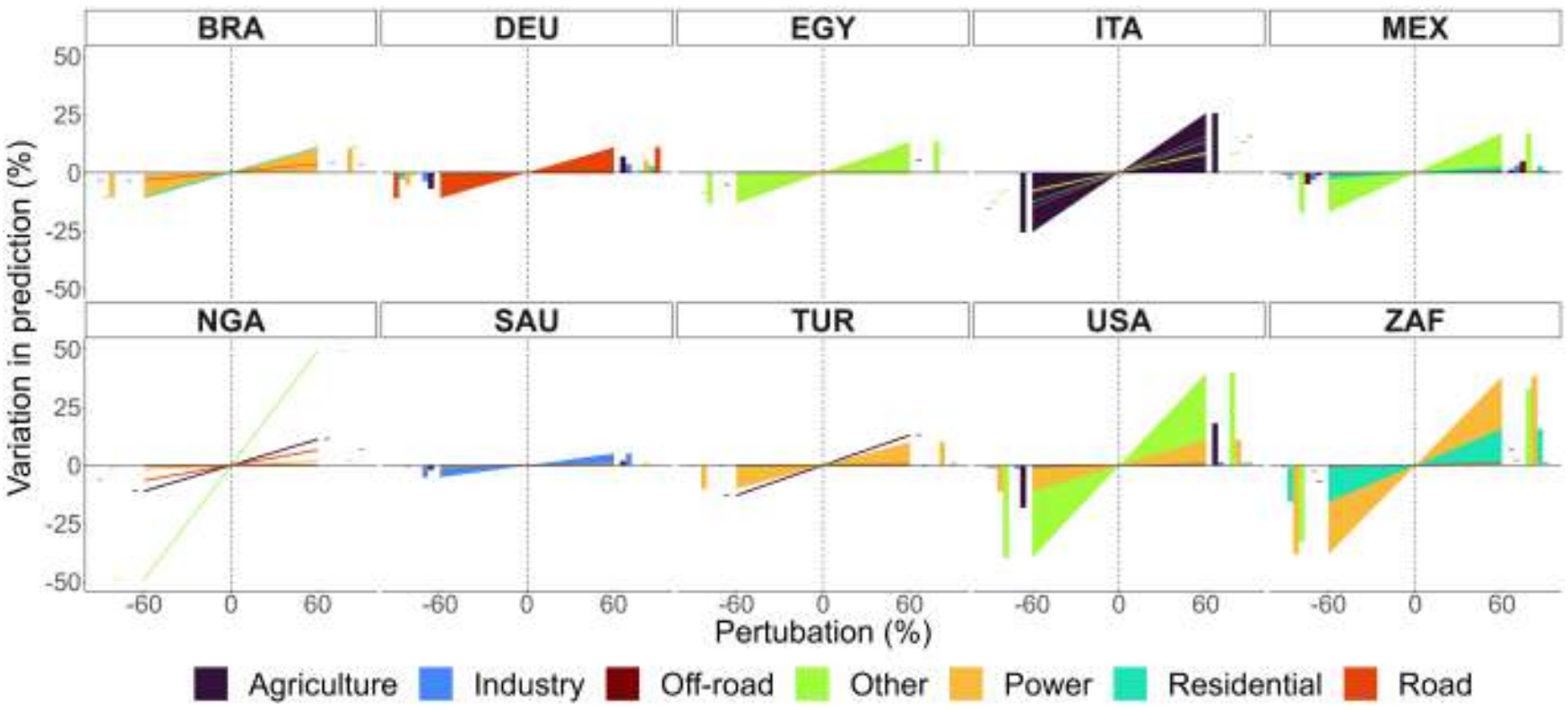

**Figure A.4:** Percentage variation in predicted concentrations of EN models for population-weighted $PM_{2.5}$ obtained from CAMS emissions, by sector and perturbation for selected countries: 84-16 (initial splitting), 75-25, and 70-30 train-test splitting. Bar charts on the sides of each subplot help visualize overlapping variations.

## A.6. Model external validation results

For the elastic net methodology only, we evaluate the previously presented CLAQC models against different global data sources: namely, the Evaluating the Climate and Air Quality Impacts of Short-Lived Pollutants (ECLIPSE) scenarios [10] (Stohl et al., 2015) provided within the GAINS model (Amann et al., 2011; Kiesewetter et al., 2015), and the TM5-FASST model (Dingenen et al., 2018).

[10] https://iiasa.ac.at/web/home/research/researchPrograms/air/ECLIPSEv5.html

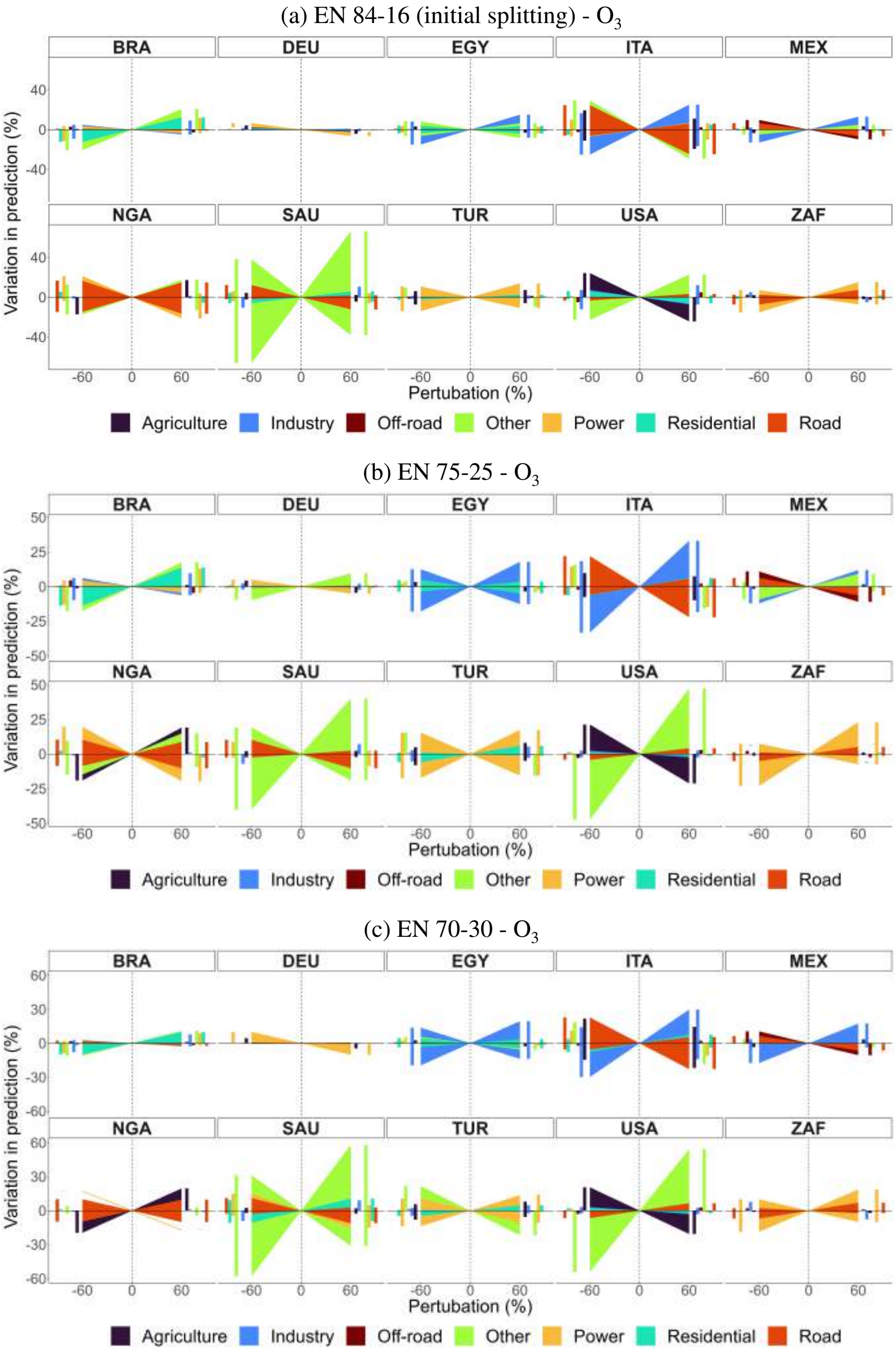

**Figure A.5:** Percentage variation in predicted concentrations of EN models for $O_3$ obtained from CAMS emissions, by sector and perturbation for selected countries: 84-16 (initial splitting), 75-25, and 70-30 train-test splitting. Bar charts on the sides of each subplot help visualize overlapping variations.

### *A.6.1. Comparison with the GAINS model*

To evaluate CLAQC models against GAINS, we obtain the anthropogenic emission data from ECLIPSE CLE (Current legislation)[11] V5, 1990-2050, quinquennial, at 0.5° spatial resolution, focusing on 2020, 2025 and 2030. Annual gridded sectoral emission data cover the following sectors and are originally expressed in $kt/year$:[12]

[11] Baseline scenario.

[12] Converted to $kg$ to implement them into CLAQC model.

- **Sectors:** Agriculture (waste burning on fields), Industry (combustion and processing), Power plants, energy conversion, extraction,[13] Residential and commercial, Waste, and Surface transportation.[14]

We download the GAINS $PM_{2.5}$ concentrations from the GAINS Online[15] tool, measured in $\mu g/m^3$.[16] To acknowledge initial differences between data sets, we compare 2015 CAMS $PM_{2.5}$ concentrations with 2015 GAINS $PM_{2.5}$ concentrations (see Figure A.8).

We compare the model outcomes after having applied population weights to the GAINS reported concentrations (see Figure A.9). The weighted CAMS concentrations from almost all considered countries are above the line of equality. Given that starting emissions and concentrations show different values between the two approaches, CLAQC and ECLIPSE-GAINS models, we expect that also their outcomes will yield different results. While CAMS concentrations range between 19.3 and 199.7 $\mu g/m^3$ (median of 54.3 $\mu g/m^3$), GAINS concentrations span between 9.5 and 74.9 $\mu g/m^3$ (median of 30.6 $\mu g/m^3$).

Regarding the comparison of CAMS and GAINS emissions, we find that while annual pollutant totals in the two data sets reflect similar magnitudes, annual sectoral emissions diverge substantially in their order of magnitude, *e.g.*, up to the order of thousands. More broadly, such heterogeneity is confirmed when comparing other sectoral emission sources present in the literature (Kurokawa & Ohara, 2020; Li et al., 2017). Thus, sector-specific air quality models face the infamous problem of drastic input source uncertainty when it comes to delivering sectoral detail. Importantly, even a few orders of magnitude differences between the emissions input data and the emissions data used during the model training may generate non-realistic concentrations as the resulting coefficients are scaled to the order of magnitude of the underlying training data. There are two ways of overcoming this issue: i) using CLAQC for scenario comparison, *i.e.*, a reference scenario and a policy scenario are simulated and the difference between the two scenarios may be used for policy analysis instead of the absolute values; ii) implementing the emissions into CLAQC by applying the CAMS sectoral emission shares to the total of the emissions inputted. Here we apply the latter.

We then input ECLIPSE emissions into the CLAQC model. Firstly, we pre-process the data to match the input requirements of CLAQC, in particular:

- We aggregate sectoral emissions to make them match CLAQC's variables.
- We re-scale the surface transportation sector into Off-road and Road transportation based on CAMS shares.
- We re-scale ECLIPSE CLE V5 annual sectoral emissions to monthly sectoral emissions by applying CLAQC monthly emission profiles based on 2014-2018 data.
- We use the resulting gridded emissions with typical meteorology data[17] and aggregate them at the country-level, to be implemented into the CLAQC model.

Finally, after running ECLIPSE emissions into CLAQC, we aggregate the obtained country-level monthly concentrations at the annual level to compare them with GAINS country-level annual concentrations.

As shown in Figure A.10, the GAINS model underestimates population-weighted concentrations of $PM_{2.5}$ in several Asian countries for the year 2020.[18] CLAQC predicted values range between 2.9 and 154.2 $\mu g/m^3$ (median of 67.7 $\mu g/m^3$), as opposed to GAINS concentrations ranging between 8.5 and 80.6 (median of 27.2 $\mu g/m^3$). These differences can be explained by the different sectoral emissions aggregations, spatial and temporal resolutions, and the different approaches followed in calculating concentrations. While CAMS concentrations are derived from a combination of multiple sources, including measurements taken from monitoring stations, satellite observed data, and modeled atmospheric data (from an ensemble of models), GAINS uses relationships from EMEP and CHIMERE models (see section 2.3). Since the CAMS data uses satellite imagery, it includes many natural sources that may not be easily observed by models, such as sea salt, desert dust, and wildfires.

### *A.6.2. Comparison with the TM5-FASST model*

The TM5-FASST model[19] is a reduced-form air quality source-receptor model at the global scale constructed by the JRC. CLAQC model comparison is applied only to TM5-FASST single-country regions among the 56 regions available.

TM5-FASST concentrations are expressed as population-weighted $PM_{2.5}$ in $\mu g/m^3$, including dust and sea salt. Thus, they are directly comparable with CAMS-CLAQC yearly population-weighted concentrations, *i.e.*, CLAQC model's outcomes aggregated at the annual level. We use TM5-FASST annual pollutant total emissions[20].

We implement the 2015 CAMS emissions into the TM5-FASST scenario aggregating the CAMS monthly sectoral emissions by pollutant and year and comparing its predictions with CAMS emissions inputted into CLAQC models. As a result, in the case of $PM_2.5$ exposure, TM5-FASST with CAMS-CLAQC emissions predicts lower values compared to CLAQC emissions into the CLAQC model (see figure A.11a). The latter ones range between 5.7 and 55.7 $\mu g/m^3$, with a median of 26.6 $\mu g/m^3$, while predictions from CAMS-CLAQC emissions into TM5-FASST range between 1.3 and 24 $\mu g/m^3$, with a median of 7.4 $\mu g/m^3$. Differently, in the case of $O_3$ exposure,[21] CAMS-CLAQC emissions into CLAQC model predict lower concentrations compared to TM5-FASST model, as detailed in Figure A.11b. Specifically, CLAQC predicted exposures from CAMS-CLAQC emissions range between 5.2 and 55.7 $\mu g/m^3$ (median of 22.1 $\mu g/m^3$), while TM5-FASST values between 47.8 and 140 $\mu g/m^3$ (median of 114 $\mu g/m^3$).

[13] Including gas flaring.

[14] Shipping sectoral emissions are not considered.

[15] https://gains.iiasa.ac.at/models/index.html (IIASA, 2009)

[16] Notice that for $O_3$ GAINS reports a different metric, *i.e.*, $SOMO35$, therefore we make such evaluation for $PM_{2.5}$ only.

[17] An average of all years.

[18] The same pattern is repeated in the years 2025 and 2030, so we omit such graphs.

[19] Derived from `"spreadsheet FASST V1.2 NORMALIZED"`.

[20] Consisting in IPCC Fifth Assessment's Representative Concentration Pathways (RCP) (Lamarque et al., 2010).

[21] Converted from 6mDMA8 expressed in *ppb* into $O_3$ mean exposure in $\mu g/m^3$.

## A.7. Emission scenarios

### A.7.1. Emission scenarios of models from DACCIWA emissions

In this section, we present the stylized emission scenarios generated by using DACCIWA emissions in both EN and ML models for a subset of countries: the Democratic Republic of the Congo (COD), Egypt (EGY), Ethiopia (ETH), Kenya (KEN), Nigeria (NGA), Tanzania (TZA), Uganda (UGA), and Sudafrica (ZAF) (see Figures A.12, A.13, and A.14).

As in the case of models derived from the CAMS emissions, EN models from DACCIWA emissions shown in Figure A.12 exhibit greater sectoral variability compared to ML models. In both methods, the Residential sector emerges as a significant contributor to $PM_{2.5}$ concentrations. The Transport and Industry sectors are consistently present in all ML models considered, though with relatively lower weights than the Residential sector. Also, the Power sector has a minimal impact on concentrations, except for ZAF. Among the EN models, the Power sector is influential in five out of eight countries, while in ML models, it notably contributes to concentrations in only one country (ZAF). Regarding $O_3$, the Industry and Power sectors exhibit higher contributions in COD, EGY, KEN, and ZAF in EN models. While ML models consistently include the Residential, Transport, and Industry sectors. In most cases, both EN and ML models capture the same relationship between emissions and concentrations, though with different magnitudes. However, there are instances where ML and EN models present contrasting associations between sectors and concentrations. For example, in the EN model for COD, an increase in industrial emissions corresponds to an increase in concentrations, while the ML model indicates a decrease.

In Figures A.13 and A.14, we show the variation in % in predicted concentrations of EN and ML models (without pixel detail) obtained from CAMS emissions, not only by sector and perturbation but also by precursor, for selected countries.

### A.7.2. Emission scenarios of ML models with pixel detail

In Figures A.15 and A.16, we show perturbation plots for ML models with sub-national detail. When considering $PM_{2.5}$, it is observed that both versions of the ML models generally select the same precursors. However, there are slight variations in the prediction outcomes based on changes in these precursors. For instance, in the USA model, all available sectors are chosen in both model versions. However, in the pixel-level model, the Residential and Road sectors carry more weight compared to the aggregate version, particularly for positive perturbations. This pattern holds true for models predicting $O_3$ levels as well. Continuing with the example of the USA case, while for most precursors the contribution seems to be similar among models, the pixel-level model exhibits a reduced relevance of the Residential and Power sectors in comparison to the country-level model, especially for negative perturbations. On the other hand, the Agriculture sector makes a similar contribution in both model versions. Overall, it appears that the ML aggregate version serves as a reliable approximation of the spatially heterogeneous models with pixel detail, indicating its effectiveness in capturing the underlying dynamics.

(a) ML 80-20 splitting - $PM_{2.5}$

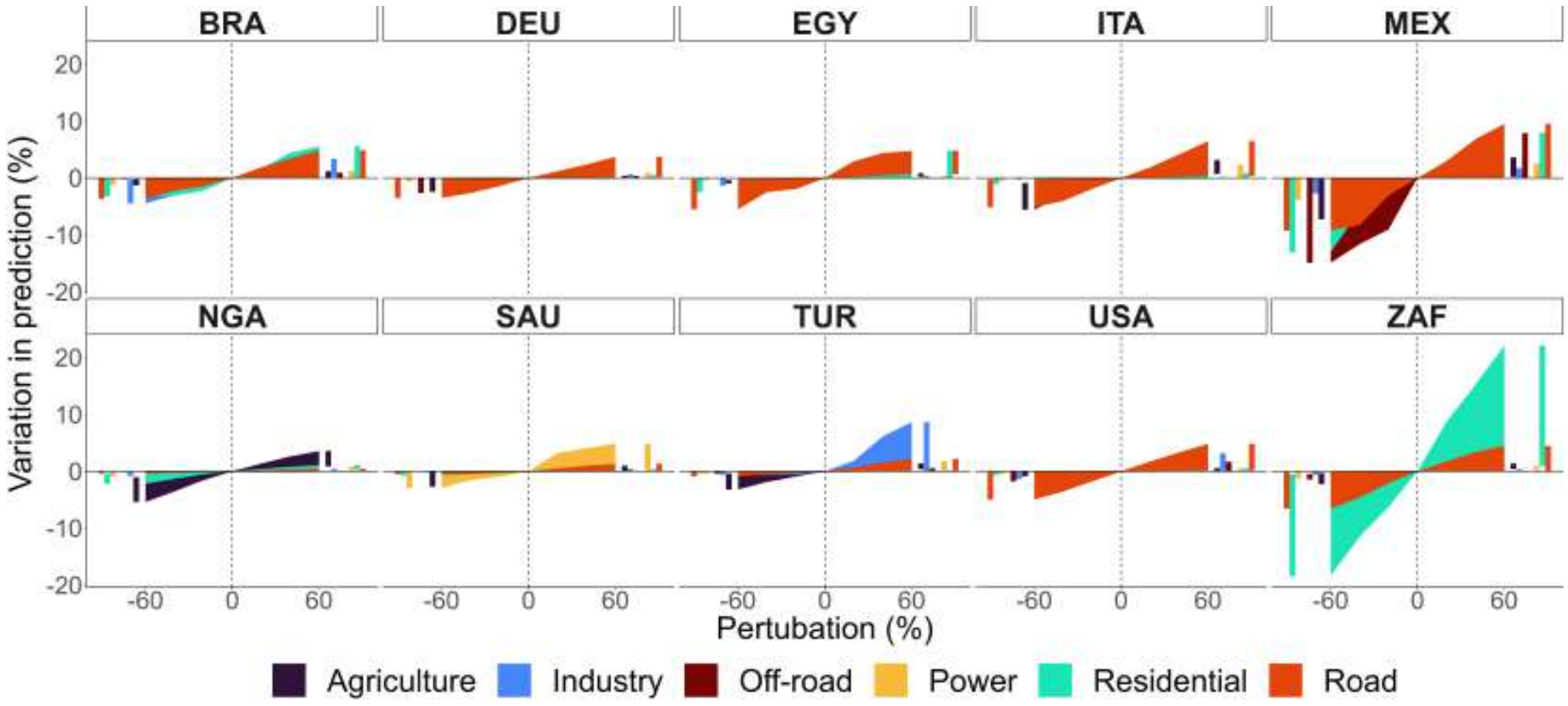

(b) ML 75-25 splitting (initial splitting) - $PM_{2.5}$

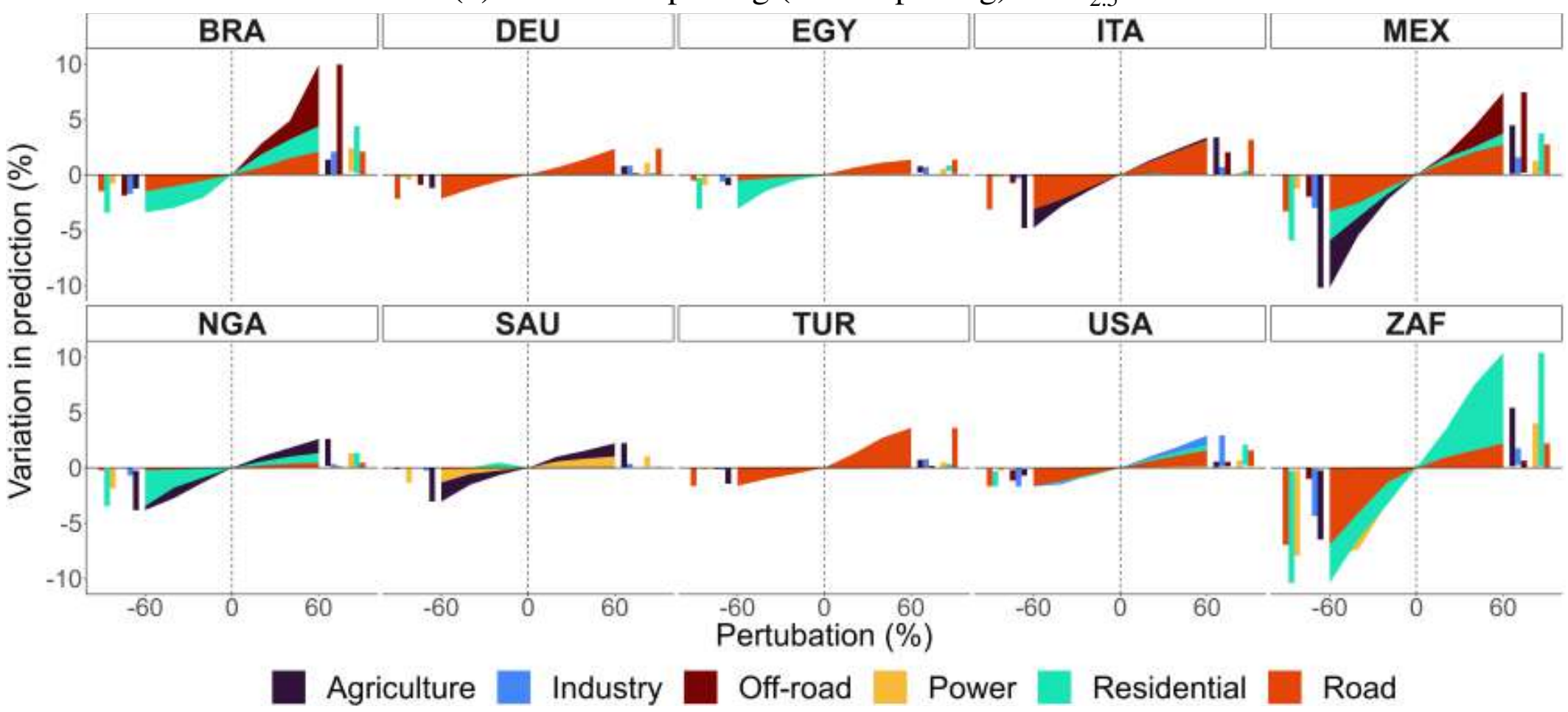

(c) 70-30 splitting - $PM_{2.5}$

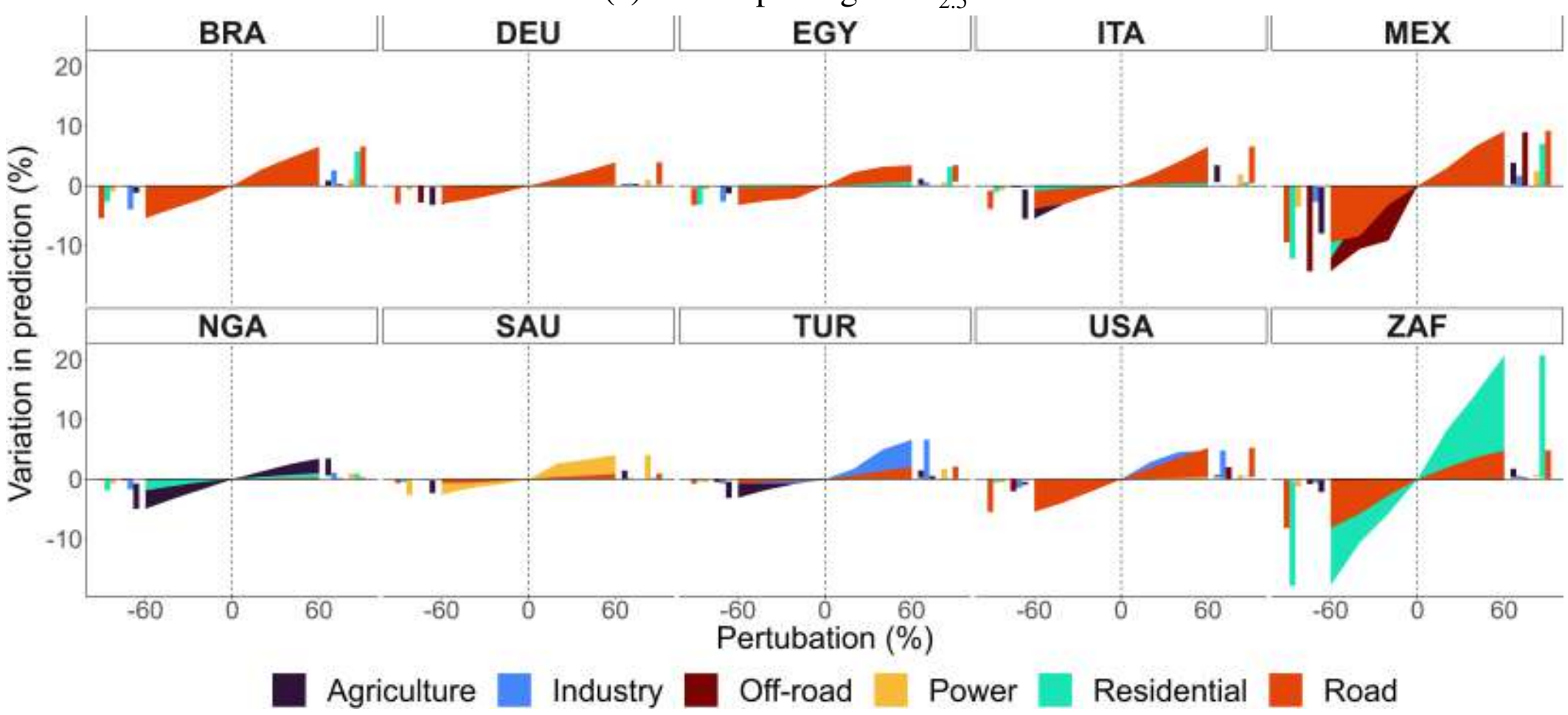

**Figure A.6:** Percentage variation in predicted concentrations of ML models for weighted $PM_{2.5}$ obtained from CAMS emissions, by sector, pollutant, and perturbation for selected countries: 80-20, 75-25 (initial splitting), and 70-30 train-test splitting.

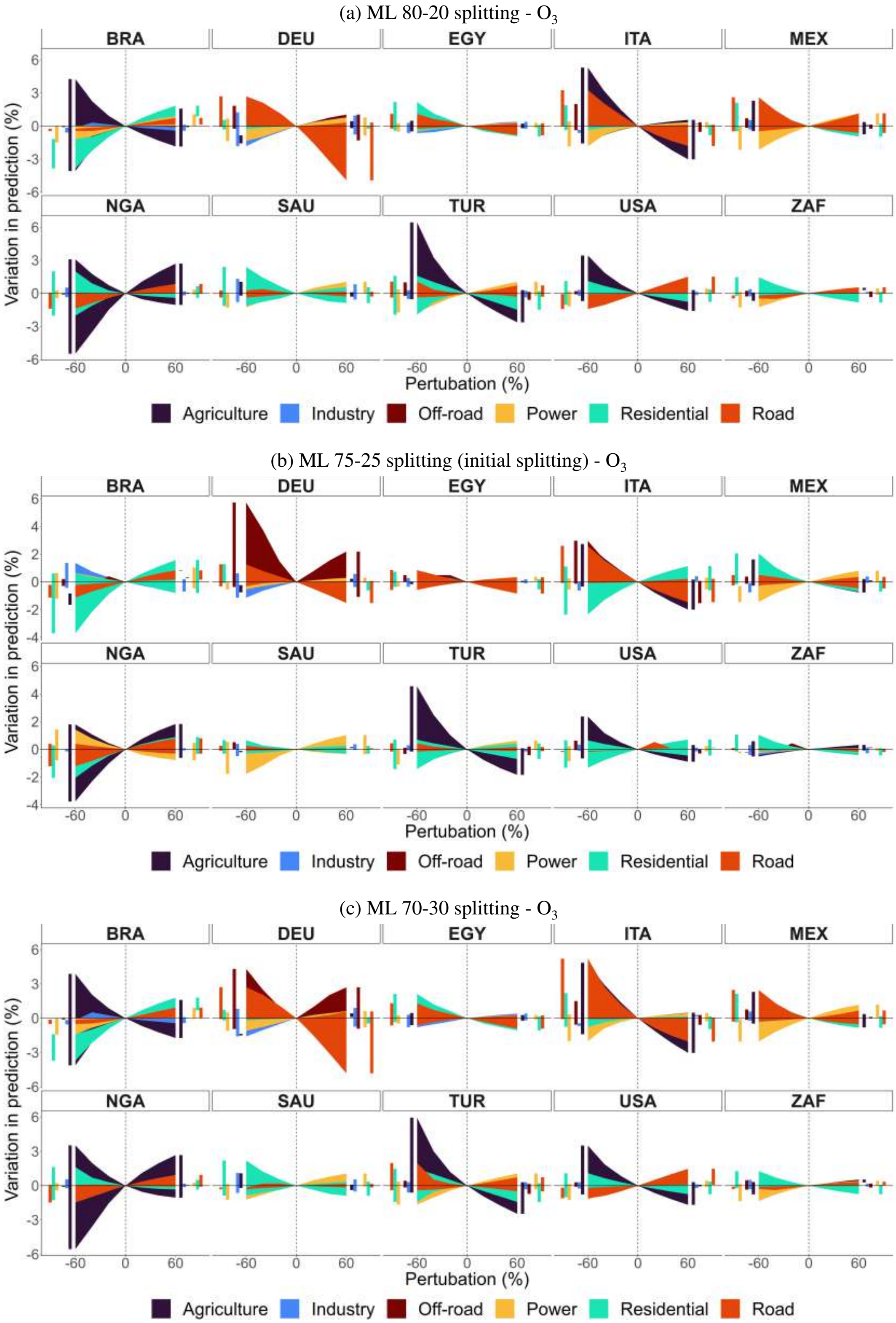

**Figure A.7:** Percentage variation in predicted concentrations of ML models for $O_3$ obtained from CAMS emissions, by sector, pollutant, and perturbation for selected countries: 80-20, 75-25 (initial splitting), and 70-30 train-test splitting.

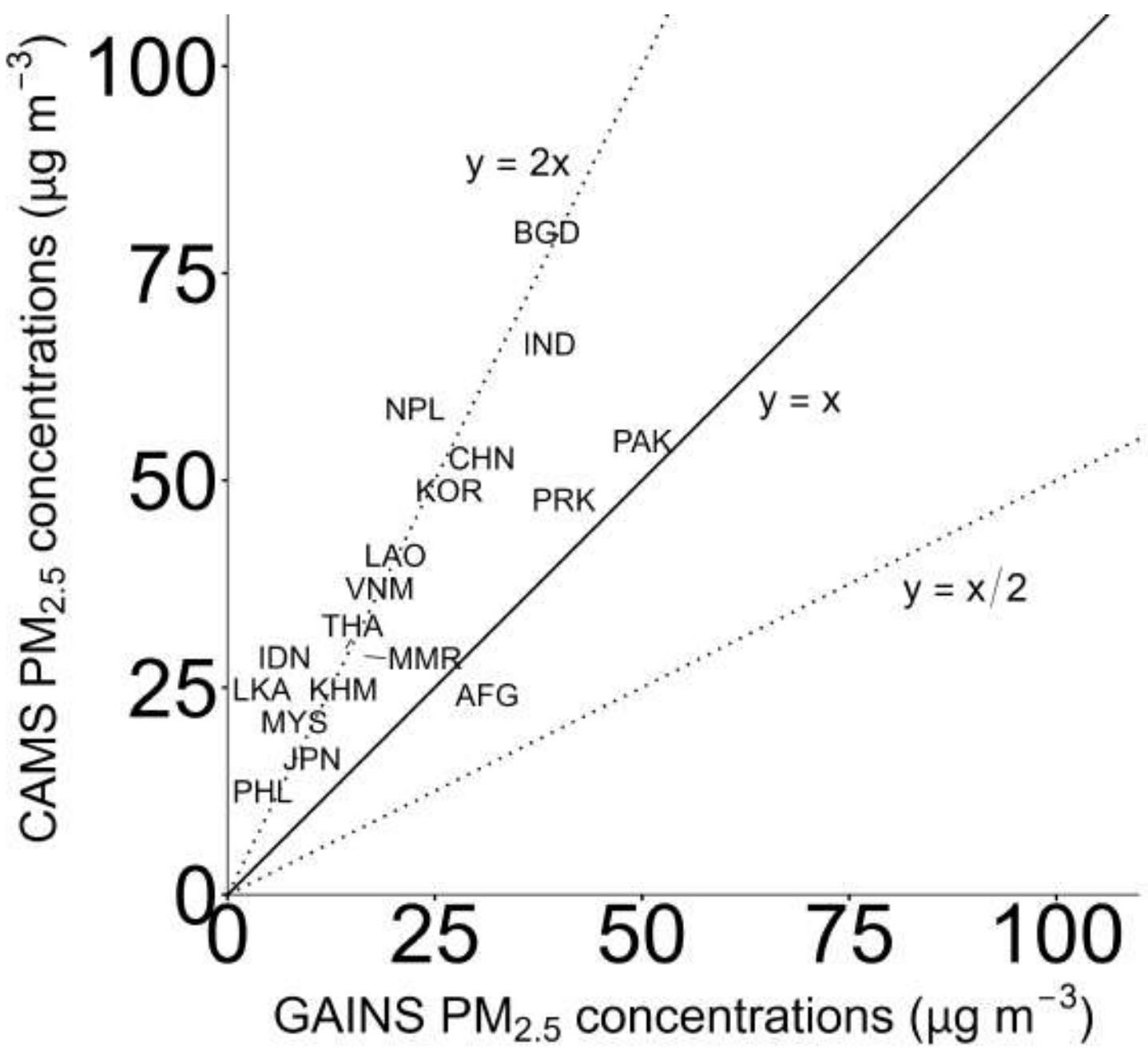

**Figure A.8:** 2015 country-level annual concentrations of $PM_{2.5}$ in Asia from CAMS and GAINS data sets ($\mu g/m^3$). The dotted lines represent the following factor differences between models: $y = 2x$ and $y = \frac{x}{2}$.

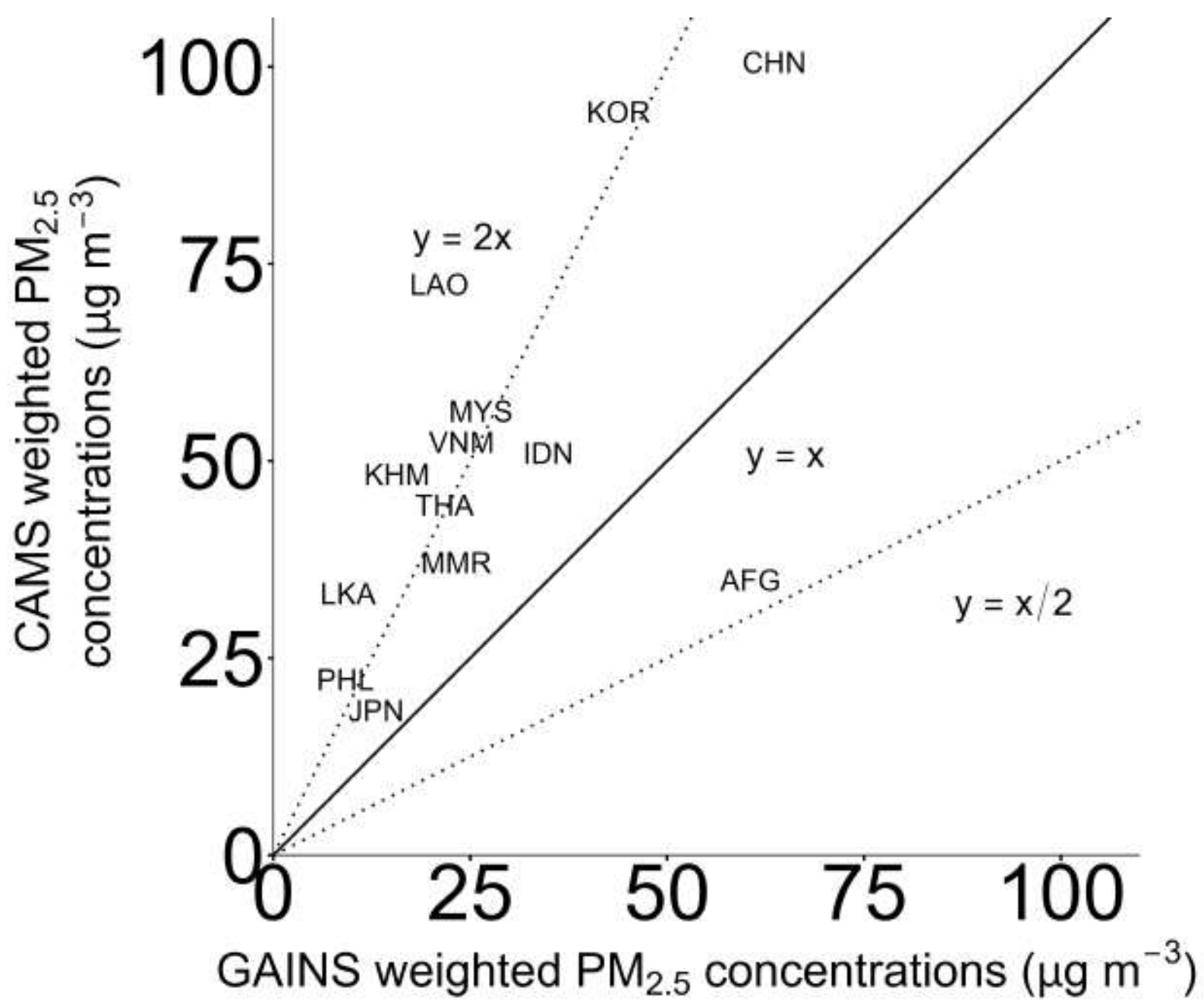

**Figure A.9:** 2015 country-level annual population-weighted concentrations of $PM_{2.5}$ in Asia from CAMS and GAINS data sets ($\mu g/m^3$).

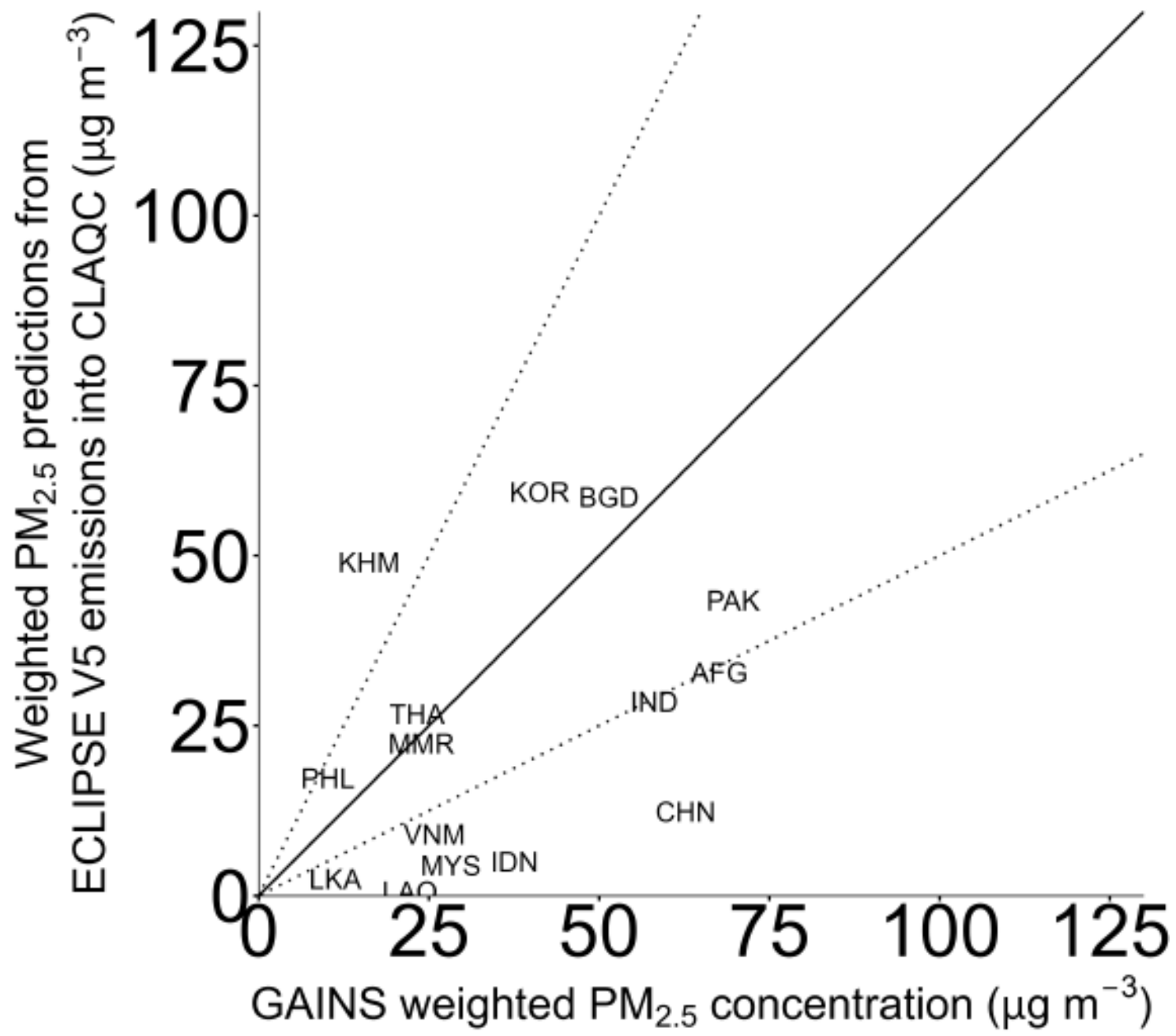

**Figure A.10:** 2020 country-level annual population-weighted concentrations of $PM_{2.5}$ in Asia from CLAQC and re-scaled GAINS data sets ($\mu g/m^3$).

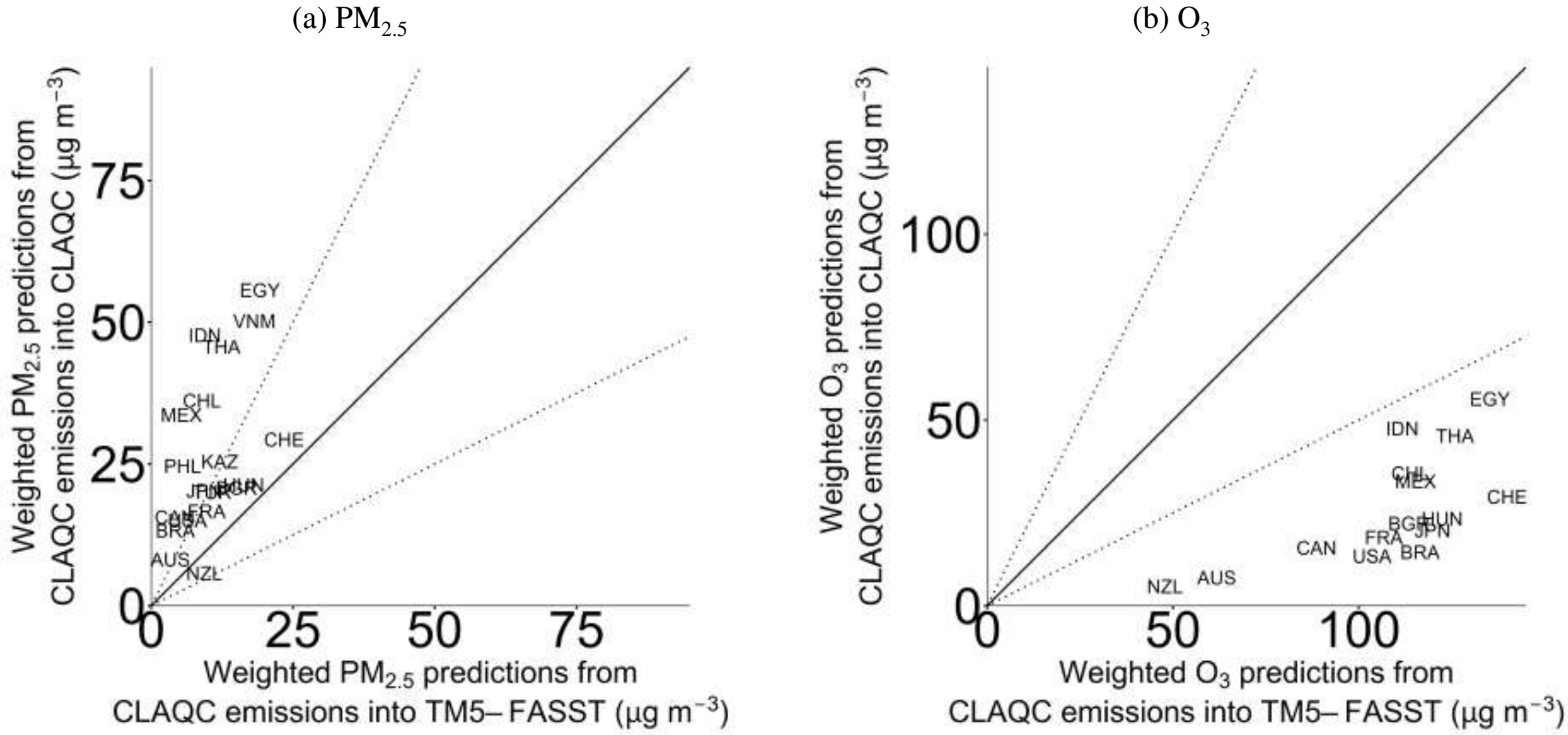

**Figure A.11:** Country-level annual population-weighted concentrations of $PM_{2.5}$ and $O_3$ from CLAQC emissions into CLAQC and CLAQC emissions into TM5-FASST ($\mu g/m^3$).

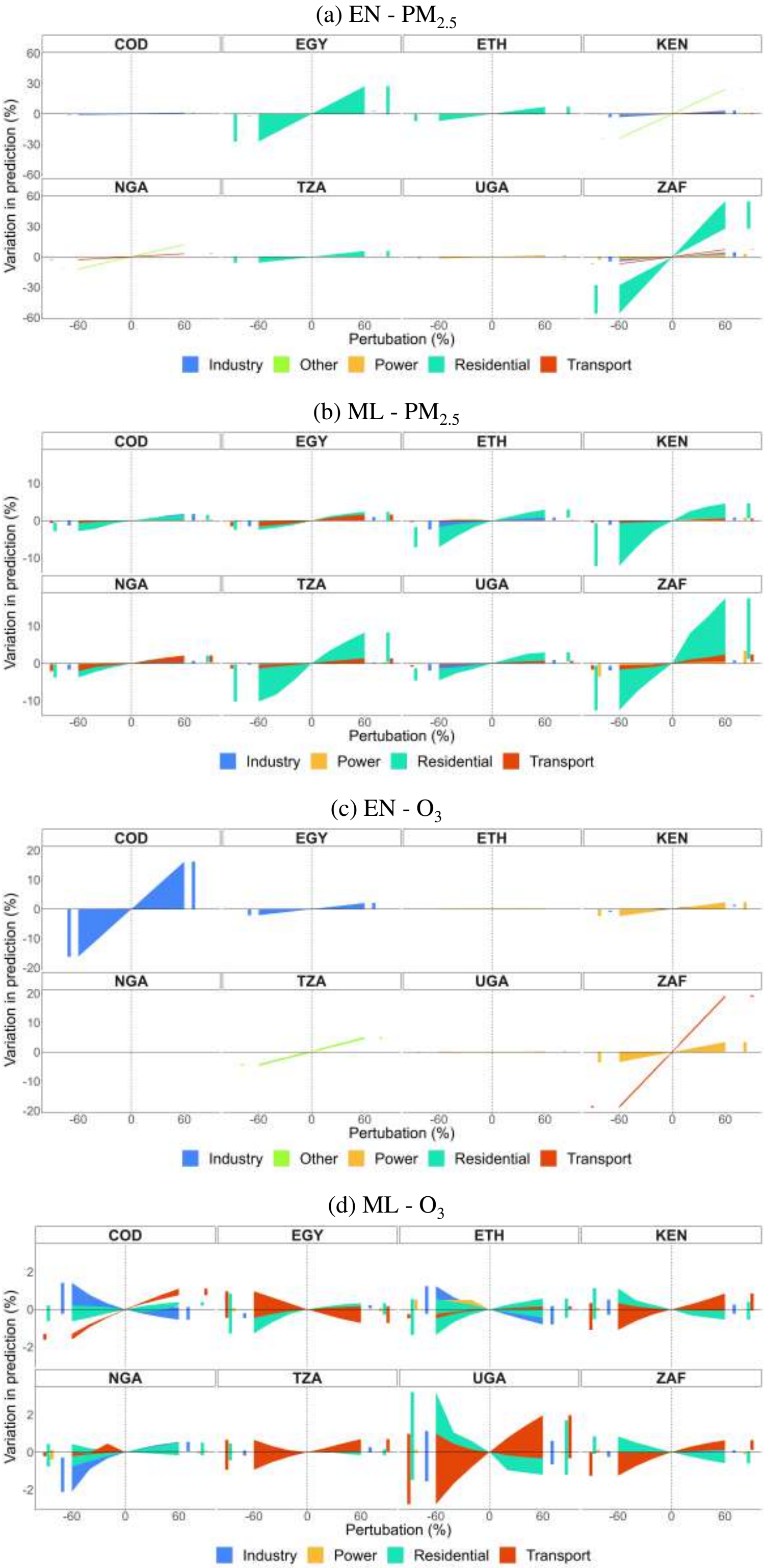

**Figure A.12:** Percentage variation in predicted concentrations by sector and perturbation for selected countries in EN and ML models from DACCIWA emissions for $PM_{2.5}$ and $O_3$ (without pixel detail). Bar charts on the sides of each subplot help visualize overlapping variations.

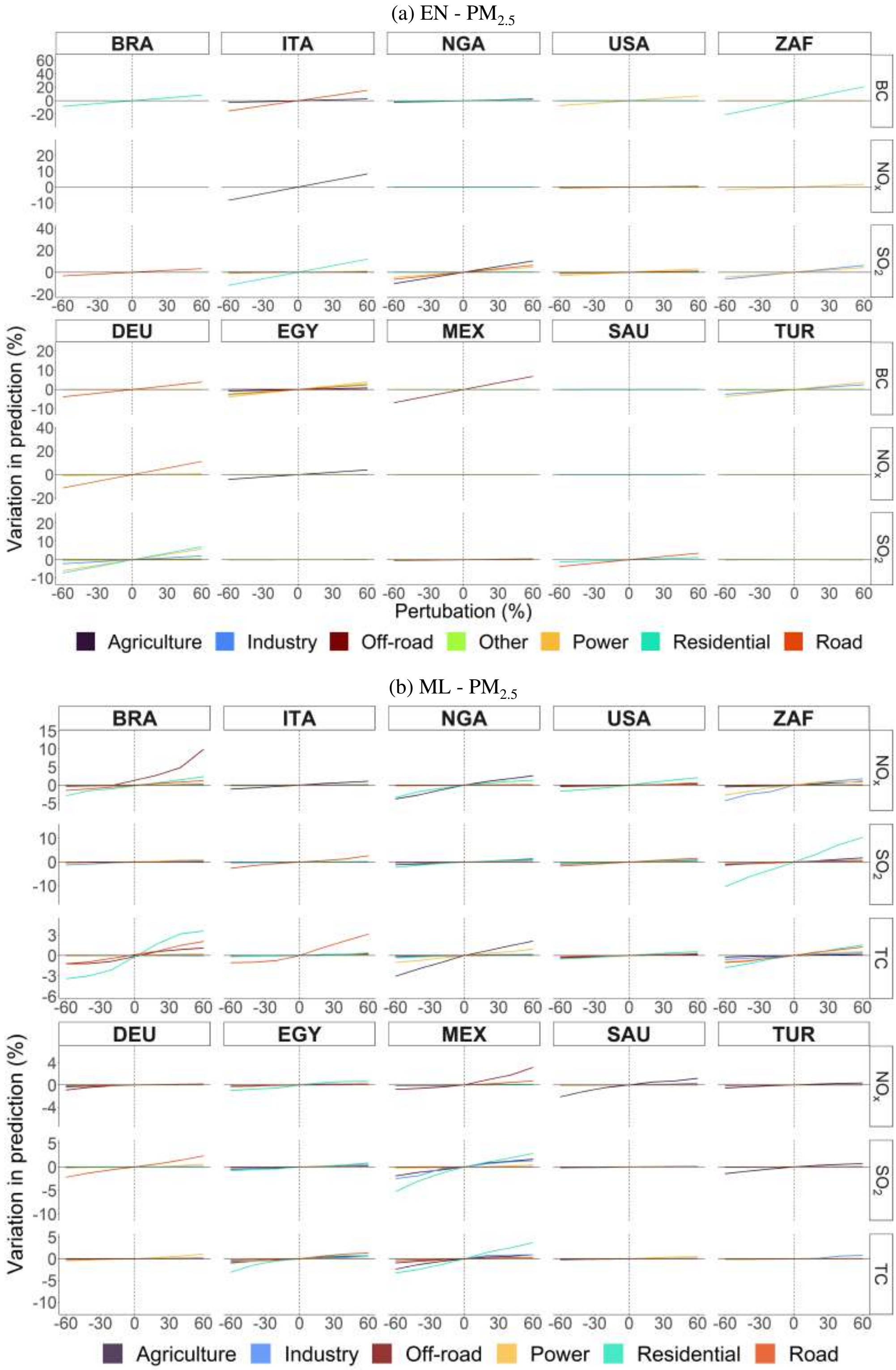

**Figure A.13:** Percentage variation in predicted concentrations of EN and ML models (without pixel detail) for $PM_{2.5}$ obtained from CAMS emissions, by sector, pollutant, and perturbation for selected countries.

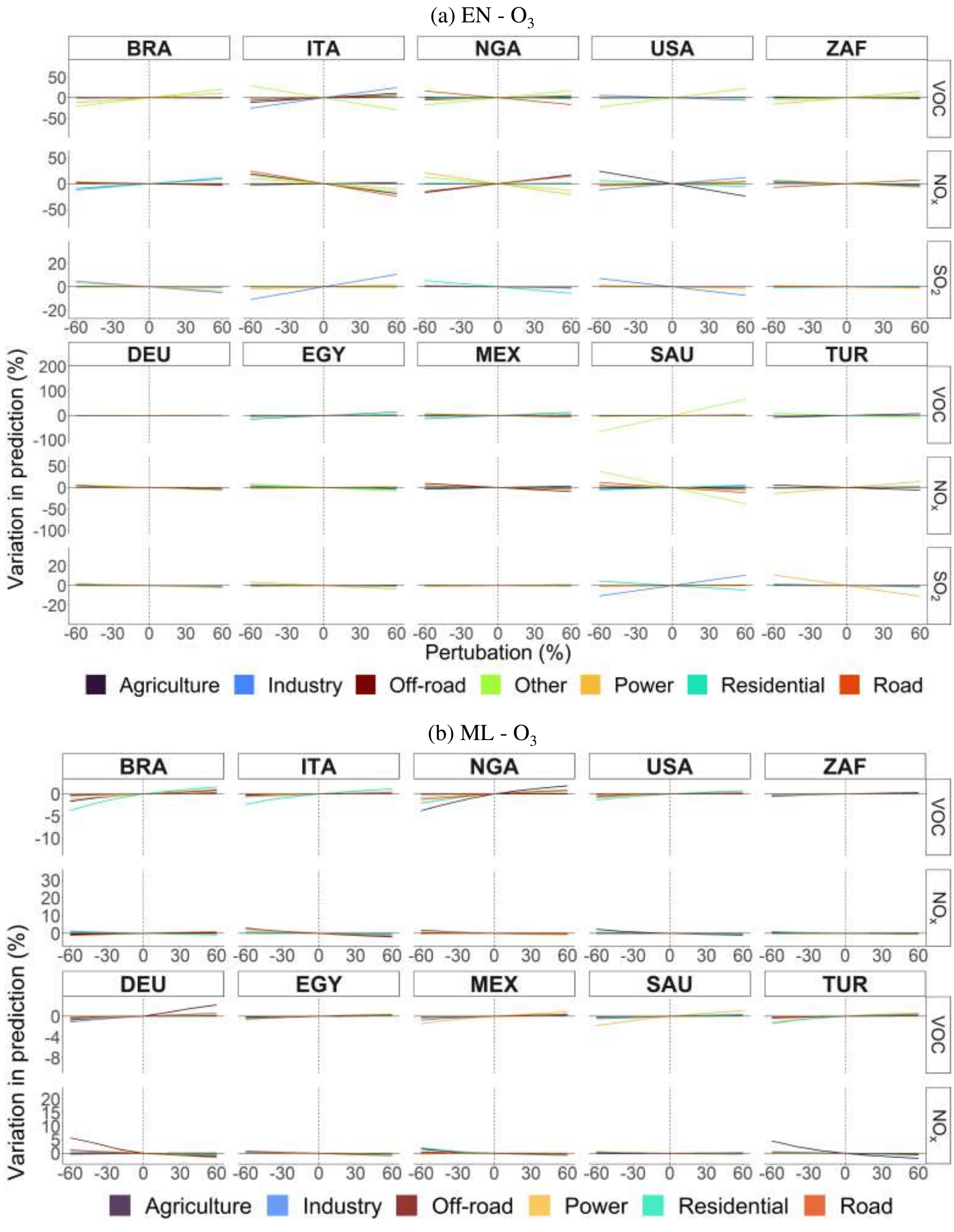

**Figure A.14:** Percentage variation in predicted concentrations of EN and ML models (without pixel detail) for $O_3$ obtained from CAMS emissions, by sector, pollutant, and perturbation for selected countries.

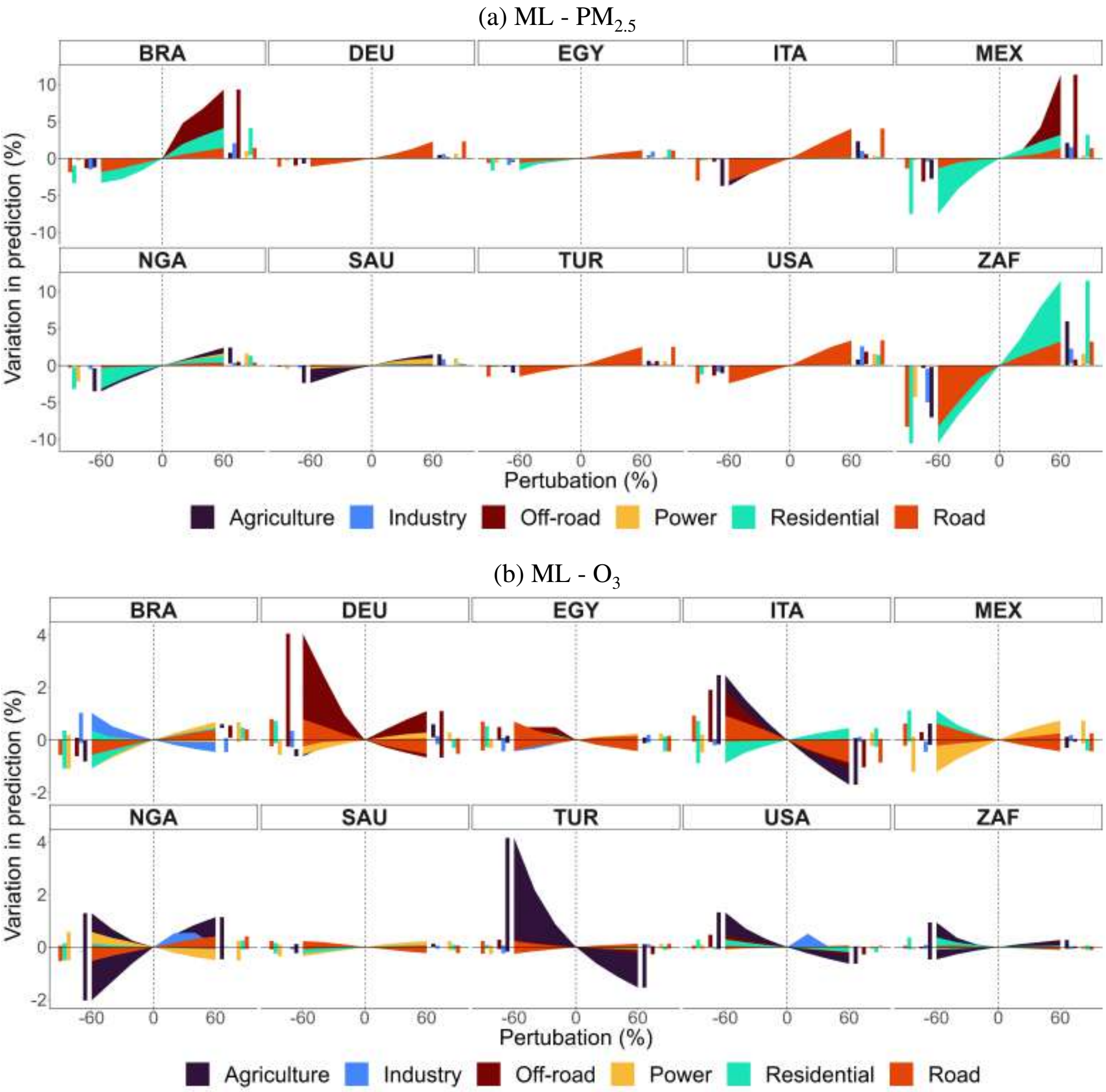

**Figure A.15:** Percentage variation in predicted concentrations by sector and perturbation for selected countries in ML models (with pixel detail) obtained from CAMS emissions. Bar charts on the sides of each subplot help visualize overlapping variations.

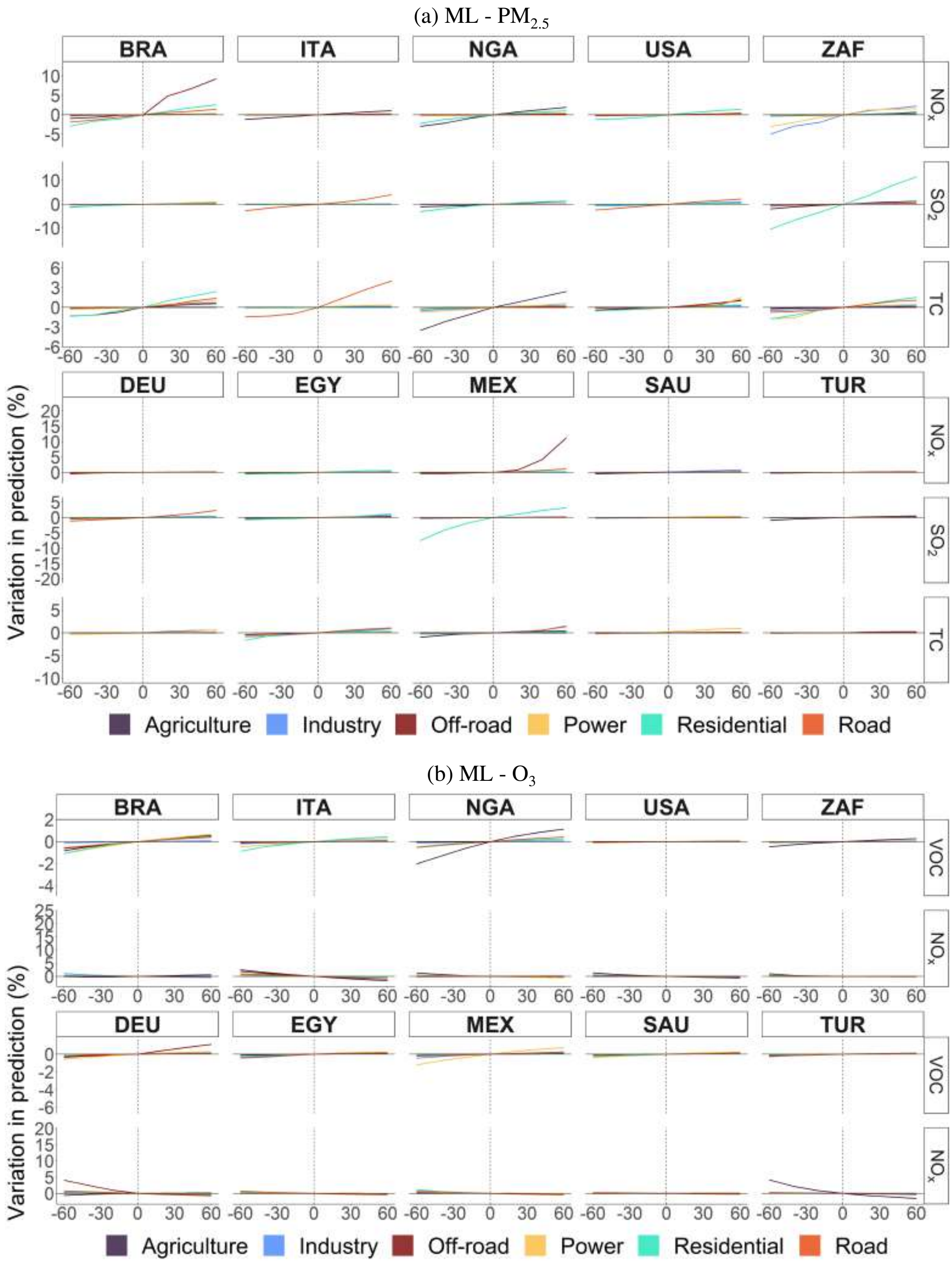

**Figure A.16:** Percentage variation in predicted concentrations of ML models (with pixel detail) obtained from CAMS emissions, by sector, pollutant, and perturbation for selected countries.

# CRediT authorship contribution statement

**Stefania Renna:** Investigation, Data curation, Methodology, Software, Validation, Writing — Review & Editing. **Francesco Granella:** Investigation, Data curation, Methodology, Software, Validation, Writing — Review & Editing. **Lara Aleluia Reis:** Conceptualization of this study, Investigation, Project administration, Supervision, Data curation, Methodology, Software, Validation, Writing — Original draft preparation, Writing — Review & Editing. **Paulina Schulz-Antipa:** Investigation, Data curation, Software, Validation, Writing — Review & Editing.

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
