# Peer review of "Graphical Abstract"

_EGUsphere, 2024_

## Author Comment (AC1)

**Authors' response to Reviewer 1**

We sincerely thank the Reviewer for taking the time to thoroughly review our manuscript and provide relevant feedback to improve it further. Below we address their concerns and suggestions point by point.
* * *
**Reviewer 1**

**General comment:**

*This article aims at proposing two methodologies for assessing impact of policy scenarios on monthly and annual pollutant concentrations ($PM_{2.5}$ and $O_3$) at country level on a global scale. The first approach is based on Elastic Net models and the second one on machine learning models (XGboost). Those methodologies used several datasets as inputs such as pollutant emissions, weather data, and concentrations data that are harmonized to a common a grid of 0.5° x 0.5° from 2003 to 2021. Results show that EN models are performing well for annual total exposure, but ML models are better for evaluating the contribution of individual sector of emissions. I would like to congratulate the authors of the paper for their interesting work. The article is very well written and clearly structured. The scientific topic is of a great interest and the methodologies proposed in the paper are quite innovative considering the published approaches. However, some clarifications still need to be made in the text as highlighted in the specific comments concerning the consideration of secondary inorganic aerosols in the models and the composition of the "Other" sector of emissions for $PM_{2.5}$ estimations. Some sensitivity tests could also be performed to address the impact of these features on the final estimation as well as the effect of the proportion of the train and test sets when applying the models.*

**Specific comments:**

**Reviewer 1 Point 1** — *Page 1, line 33: The following change should be made: "fine particulate matter (particles with a diameter less than 2.5 µm, PM2.5)"*

**Reply**: We thank the Reviewer for this comment and have added the suggested part in the text.

**Reviewer 1 Point 2** — *Page 2, line 4: The following change should be made: "chemistry-transport models (CTMs) are tools for calculating the impact of emissions on pollutant concentration levels"*

**Reply**: Thank you, we agree that it would be better to specify that concentration levels refer to pollutants. We have changed the sentence accordingly.

**Reviewer 1 Point 3** — *Page 2, lines 7 to 10: ACT tool (Air Control Toolbox, Colette et al., 2022) should be mentioned here. ACT is a surrogate model to explore mitigation scenarios in air quality forecasts. This is designed for estimation at the European level on a daily basis.*

*Colette, A., Rouïl, L., Meleux, F., Lemaire, V., and Raux, B.: Air Control Toolbox (ACT_v1.0): a flexible surrogate model to explore mitigation scenarios in air quality forecasts, Geosci. Model Dev., 15, 1441–1465, https:doi.org10.5194 gmd-15-1441-2022, 2022.*

**Reply**: We thank the Reviewer for suggesting adding the ACT tool. Indeed, it is a relevant tool to discuss; we have mentioned it in the 1.0 "Introduction" Section: "The most commonly used models to evaluate air pollution policies, such as the Greenhouse Gas - Air Pollution Interactions and Synergies (GAINS) model (Amann et al., 2011; Kiesewetter et al., 2015), the SHERPA tool (Thunis et al., 2016), the TM5-FAst Scenario Screening Tool (TM5-FASST) model (Dingenen et al., 2018), and the Air Control Toolbox (ACT) tool (Colette et al., 2022), rely on a variety of methods."

**Reviewer 1 Point 4** — *Page 2, line 19: "The latter one is the most detailed, up-to-date reduced form air pollution model", the following clarification should be made: "on a global scale".*

**Reply**: We agree that we should have specified the global scale reference, so we have changed the text accordingly: "The latter one is the most detailed, up-to-date reduced-form air pollution model on a global scale."

**Reviewer 1 Point 5** — *Page 3, line 9: Please explain further what you mean by "factors" and the impact of the monitoring of pollutant concentrations in ambient air. Perhaps, it should be mentioned that monitoring stations are used for air quality assessment and the lack of measurement points in an area is a strong constraint in that objective.*

**Reply**: We thank the Reviewer for this comment and suggestion that we have clarified in the text. We changed the paragraph as follows: "Monitoring stations are used for air quality assessment. However, the lack of a spatially consistent large ground monitoring network in a given area is a strong constraint to achieving this objective. Despite the recent harmonization and open-access advancements in air pollution data, most publicly available global ground-level monitoring databases (e.g., OpenAQ (2024)) provide reasonable territorial coverage of the population only in developed countries, in particular in the United States and Europe. The network of ground-level monitors is growing in emerging economies such as China and India, yet urban and rural areas are largely unmonitored in middle- and low-income countries. The uneven ground-level monitoring geographical coverage is problematic, as factors driving the emissions-concentrations relationship differ between monitored and unmonitored areas (e.g., population density, distance from industrial sources, GDP per capita)."

**Reviewer 1 Point 6** — *Page 3, line 13: "Global, gridded reanalysis data combine and harmonize satellite air pollution measurements with ground-level monitors." The following clarification should be made: CTM estimates are also used.*

**Reply**: We thank the Reviewer for highlighting this imprecision. We have included a reference to chemistry-transport models and changed the text to: "Global, gridded reanalysis data combine and harmonize satellite air pollution measurements and CTM output with ground-level monitors."

**Reviewer 1 Point 7** — *Page 3, lines 24 and 25: The following clarification should be made: "the need to homogenize different grids in terms of spatial resolution"*

**Reply**: We agree with this suggestion and have updated the text accordingly: "Among the disadvantages, the need to homogenize different grids in terms of spatial resolution may lead to approximations during the data manipulation process."

**Reviewer 1 Point 8** — *Page 4, line 24: Is the odd-road transportation corresponds to shipping and aviation? Could you please clarify.*

**Reply**: We thank the Reviewer for the opportunity to clarify this point. The "Off-road transportation" sector we employ in CLAQC includes railways and other types of non-road transports not typically used on public roads, such as agricultural machinery, construction equipment, and certain types of off-road vehicles used in industrial operations (e.g., tractors, telehandlers, excavators). It corresponds to the "Non-road transportation" (TNR) sector in the CAMS-GLOB-ANT data, to the IPCC Sector Name "Non-road ground transportation" (IPCC Sectors "Rail transport", 1A3c, and "Other transport", 1A3e), and to "SNAP 8, non-shipping and non-aviation transportation" (Granier et al., 2019, 2021; Krey et al., 2014). Therefore, it does not correspond to shipping and aviation. In general, pollutant emissions from the shipping and aviation sectors are not taken into account in CLAQC because they are beyond the scope of our analyses. We made this clearer in the text under Subsection 2.1.4. "Sectoral aggregation": "We do not include biogenic and sectoral emissions from shipping and aviation. We build the following 7 sectors from CAMS-GLOB-ANT data: Agriculture, Industry, Other (including the emissions not considered in the other sectors), Off-road transportation, Energy power generation, Road transportation, and Residential (including buildings, commercial and services). Note that the Off-road transportation sector includes railways and other types of non-road transports not typically used on public roads, such as agricultural machinery, construction equipment, and certain types of off-road vehicles used in industrial operations (e.g., tractors, telehandlers, excavators)."

**Reviewer 1 Point 9** — *Page 5, line 4: Are the natural emissions included in the "Other" sector? Could you please clarify.*

**Reply**: No, the CLAQC "Other" sector does not contain biogenic emissions. CLAQC is based on anthropogenic emissions derived from the CAMS-GLOB-ANT sectors, meaning that in terms of emissions, only anthropogenic sources are included in the models. In particular, the "Other" sector contains anthropogenic emissions from the following sectors: "Solvents application and production" (SLV), and "Solid waste and wastewater handling" (SWD). Such anthropogenic emissions are summed up within

pollutants' sectoral emissions, into this residual sector, that we call "Other". We have not included biogenic emissions as CLAQC was built aiming to support policy impact. Policies generally are not likely to target non-anthropogenic sources. However, you are right in stating that those emissions in some regions of the globe are major contributors to exposure. We have clarified this in the manuscript and integrated it as a limitation of our tool subject to future improvement. In the "Limitation" Section, we have stated: "Fifth, as CLAQC was built to support policy impact evaluation, the approaches presented here do not explicitly model transboundary movements of pollution and biogenic emissions such as desert dust and sea salt". Given that earlier in the text this aspect was not discussed, we have clarified this under Subsection 2.1.4. "Sectoral aggregation" as well: "We do not include biogenic and sectoral emissions from shipping and aviation. Furthermore, natural emissions, such as desert dust and sea salt, are not taken into account since they are less likely to be subject to policy interventions".

**Reviewer 1 Point 10** — *Page 5, line 19: How did you manage to downscale the concentrations data to 0.5°? Please explain further.*

**Reply**: Thank you for asking for this clarification. We have further explained the rescaling process updating Subsection 2.4. "Grid definition" as follows: "All gridded data sources are rescaled to the same $0.5° \times 0.5°$ coordinate grid through linear interpolation, based on the population grid, and merged into a single data set. For instance, concentration data originally at $0.75° \times 0.75°$ spatial resolution are downscaled to $0.5° \times 0.5°$, generating intermediate values that align with the reference grid. The interpolation is implemented using the `interp_like` function from the `xarray` Python package."

**Reviewer 1 Point 11** — *Page 5, lines 20 and 21: I don't understand why you must change the unit here. Aren't ECMWF data already in $\mu g/m^3$? Please clarify.*

**Reply**: Regarding ECMWF data, the original unit of measure for GEOM Ozone is $kg/kg$, while for fine particulate matter is $kg/m^3$. So, in both cases, we convert the unit to $\mu g/m^3$ to improve results' interpretability. We have clarified this in the text under Subsection 2.3. "Concentrations". The documentation on the unit of measurement of the ozone concentration data is available at: `https://ads.atmosphere.copernicus.eu/datasets/cams-global-reanalysis-eac4-monthly?tab=overview`

**Reviewer 1 Point 12** — *In addition, Figure 2.3 is called in the main text, but the figure numbering is Figure 1. In order to improve reading of the figure, the ticks should be added in the two colorbars to make the correspondence with the tick labels in panels a) and b).*

**Reply**: Regarding Figure 2.3, we thank the Reviewers for noticing the mismatch in the figure naming, we have solved this issue. We also agree that label ticks would improve the readability of plots in Figure 1, so we have updated the color scales as follows.

[Figure]

Figure 1: Level plots of EAC4 concentrations of PM$_{2.5}$ (January 2018) and O$_3$ (July 2018) in $\mu g/m^3$ with color bar in logarithmic scale.

**Reviewer 1 Point 13** — *Figure 2 title: The expression "weighted by the population" should be mentioned in the title.*

**Reply**: To improve clarity, as suggested, we have changed the title as follows: "EAC4 concentration inputs of PM$_{2.5}$ weighted by the population and O$_3$ in $\mu g/m^3$ aggregated at the country level (2018)."

**Reviewer 1 Point 14** — *Page 7, line 25 to 30: It is a choice of simplicity to not considered secondary aerosols in the model. Have sensitivity tests carried out on the impact of this choice on the final estimation?*

**Reply**: We apologize for not being clear on this point and have tried to explain it better in the text under Subsection 3.2. "Coefficient constraints". Secondary inorganic aerosols are not directly included in the models predicting $PM_{2.5}$ concentrations. To some extent, they can be captured through the interaction terms we had included. We had included the interactions between total pollutant emissions of $PM_{2.5}$ precursors as a proxy of secondary reactions among them. In particular, interactions between $NH_3$, $NO_x$ and $SO_2$ enter the models (see Eq. 8, pages 9-10). We had constrained such relationships to be positive, i.e., an increase in emissions cannot lead to a decrease in $PM_{2.5}$ concentrations. This is to avoid the models, that are designed to optimize the best predictor variables, to choose a predictor that will predict a policy to increase precursor emissions, or to avoid decreasing emissions because of small increases in secondary $PM_{2.5}$. While in specific regions a decrease in $NO_x$ can lead, for example, to an increase in secondary inorganic aerosols, these are very local effects that tend to not occur at the country-level scale (Thunis et al., 2019; Granella et al., 2024). However, if they were to occur, the models would foresee a zero decrease instead of an increase in predicted concentrations. This means that the models would still predict the best policy path through the reduction of other precursors. We have discussed this in the 5. "Limitations" Section as follows:

"Fifth, as CLAQC was built to support policy impact evaluation, the approaches presented here do not explicitly model transboundary movements of pollution and biogenic emissions such as desert dust and sea salt. Only averages enter the models through the time and place identifiers (month-fixed effects in EN and grid identifiers in ML). Secondary inorganic aerosols (SIA) are also not directly modeled. Instead, their effects are approximated through interactions among emissions' predictors. In addition, we have not performed sensitivity analyses to assess the impact of excluding SIA on the final estimates."

**Reviewer 1 Point 15** — *Page 9, line 13: Have sensitivity tests been carried out on the splitting of the training and test data sets? If not, these tests should be considered.*

**Reply**: The splitting of the training and test data sets is different across models as they have different data samples. While EN models are trained and tested on country-level aggregated data, ML models use the country-level gridded data, with a bigger sample available. Due to these data differences, we did not harmonize the data splitting across methods. Instead, we prioritized ensuring a sufficiently large training set for the EN models to minimize the variance in parameter estimates. However, we acknowledge the importance of conducting a sensitivity analysis on the choice of train-test split. Therefore, we welcome this suggestion. We would like to clarify that while we initially reported a train-test split of 80-20 for the EN models, the actual splitting used was 84-16 for models based on CAMS emissions and 80-20 for models based on DACCIWA emissions.

To provide a more robust validation of our model configurations, we have performed additional

tests by varying the train-test splits to 75-25 and 70-30, alongside the original split for EN. These sensitivity analyses confirm that our model predictions remain stable across different train-test splits. For conciseness, we present the results for the 10 countries originally provided in the manuscript. In most cases, the dominant sectors remain consistent across splits, with slight variations in the range of prediction changes. The results show that certain countries, such as Germany (DEU), Turkey (TUR), and the United States (USA), exhibit particularly robust predictions across splitting ratios for $PM_{2.5}$. Figures 2 and 3 illustrate these scenarios for EN models, comparing train-test splits of 84-16, 75-25, and 70-30 ratios. While there are minor variations, the overall findings remain consistent. Sectors such as Industry, Agriculture, and Residential remain stable across splits, while less dominant sectors like Off-road or Other display higher variability.

[Figure]

Figure 2: Percentage variation in predicted concentrations of EN models for population-weighted PM$_{2.5}$ obtained from CAMS emissions, by sector and perturbation for selected countries: 84-16 (initial splitting), 75-25, and 70-30 train-test splitting. Bar charts on the sides of each subplot help visualize overlapping variations.

[Figure]

Figure 3: Percentage variation in predicted concentrations of EN models for O$_3$ obtained from CAMS emissions, by sector and perturbation for selected countries: 84-16 (initial splitting), 75-25, and 70-30 train-test splitting. Bar charts on the sides of each subplot help visualize overlapping variations.

**Reviewer 1 Point 16** — *Page 9, line 13: It is not clear if the whole period of the dataset (from*

*2003 to 2021) is used to train de model? Please clarify in the main text what is the periods of the train set and the test set. It is mentioned that the perturbations of emissions are applied to the last 5 years of data (page 9 line 27), thus is the model trained from 2017 to 2021?*

**Reply**: We confirm that the whole period of the dataset (2003-2021) is used both to train and test the models for both methods. While, the last 5 years of data (2017-2021) are applied for calculating emission scenarios. We have changed the Subsection 3.3. "Elastic net models" to make this clearer: "We randomly split the 2003-2021 data into training (80% of observations) and test set (20%) stratifying by month."

**Reviewer 1 Point 17** — *Page 10, line 23: If the Other sector includes Natural emissions, it could have a significant impact of $PM_{2.5}$ concentrations (from desert dust and sea salt). That may bias the estimate of the machine learning model. This sector should be considered.*

**Reply**: As specified in a previous reply, the Other sector does not consider biogenic emissions. It instead includes anthropogenic emissions derived from the following CAMS-GLOB-ANT sectors: Solvents application and production (SLV), and Solid waste and wastewater handling (SWD). We have not included biogenic emissions as CLAQC was built to support policy impact evaluation. In general, policies are unlikely to target non-anthropogenic sources. We recognize that natural emissions such as desert dust and sea salt may have a significant impact on $PM_{2.5}$ concentrations in some regions, though such emissions indirectly enter the model through the time and place identifiers. In the 5. "Limitations" Section, we have stated the following: "Fifth, as CLAQC was built to support policy impact evaluation, the approaches presented here do not explicitly model transboundary movements of pollution and biogenic emissions such as desert dust and sea salt. Only averages enter the models through the time and place identifiers (month-fixed effects in EN and grid identifiers in ML)."

**Reviewer 1 Point 18** — *Page 11, line 7: I'm a bit confused with the consideration of the secondary inorganic aerosol's formation in the model. It is explained in Subsection 3.2: "It is crucial to understand that in situations where secondary reactions substantially affect the overall mass of $PM_{2.5}$ within a country, our models are designed to omit these precursors from the list of predictors, thereby not reflecting a decrease in $PM_{2.5}$ levels." Please clarify if the secondary inorganic aerosols are excluded or not.*

**Reply**: We apologize for not being clear on this point. We have addressed this concern in a previous reply to your comment, and have tried to clarify it in the text under Subsection 3.2. "Coefficient constraints". We have also added this to the 5. "Limitations" Section, where we make explicit that the model should be used under the 'fit-for-purpose' principle, i.e., for country-level policy roll-out purposes. Secondary inorganic aerosols are not directly included in the models predicting $PM_{2.5}$ concentrations. Instead, we include interactions between specific total pollutant emissions of PM precursors as proxies

for secondary reactions. In particular, interactions between $NH_3$, $NO_x$ and $SO_2$ enter the models (see Eq. 8).

We have integrated the text as follows:

"While at the local scale, reducing certain precursors of secondary inorganic aerosols might not always lead to a decrease in $PM_{2.5}$ levels — due to nonlinear atmospheric reactions noted by Thunis et al. (2019); Ding et al. (2021) — our national-scale models focus on broader trends. To avoid giving undue importance to cases where local emissions reductions might result in increased levels of inorganic $PM_{2.5}$, we apply monotonic constraints between emissions and concentrations.

Rather than directly including secondary inorganic aerosols, the models incorporate interactions between PM precursors — specifically $NH_3$, $NO_x$, and $SO_2$ — as proxies for secondary reactions.

It is crucial to understand that in situations where secondary reactions substantially affect the overall mass of $PM_{2.5}$ within a country, our models are designed to omit these precursors from the list of predictors, thereby not reflecting a decrease in $PM_{2.5}$ levels."

**Reviewer 1 Point 19** — *Page 11, line 29: Please clarify what you mean by: "We randomly split a gridded data set stratifying by grid cell. Hence, randomization occurs over the temporal dimension."*

**Reply**: We apologize for not being clear on this point. We have changed the text as follows: "For each country-pollutant pair, the input data for ML are a grid panel dataset, composed of $N$ grid cells and observed over $T$ time periods. For each grid cell, we randomly assign the T observations (a time series) to the train or test set. Hence, we stratify by grid cell and randomization occurs over the temporal dimension. This stratified randomization ensures equal spatial representation in both data sets. Given unobservable but time-constant characteristics of cells (such as topography) and the desire for equal spatial representation, we prefer this method to simple randomized allocation, which might allocate the entire time series for a cell to either set. We use three-fourths of the data as the training set and the remaining fourth as the test set."

**Reviewer 1 Point 20** — *Page 11, line 31: Why the splitting of the training and the test data set is different from the EN model? Same question as for the EN model, were sensitivity tests on the choice of the training / test sats carried out?*

**Reply**: We have tried to address this concern in a previous reply. To provide a more robust validation of our model configurations, we are performing additional tests by varying the train-test splits to the 80-20 and 70-30 ratios, alongside the original 75-25 split for ML as well. As for EN scenario runs, results are robust to different train-test splittings for both $PM_{2.5}$ and $O_3$, with small variations. See figures 4 and 5 for variations in ML scenarios for $PM_{2.5}$ and $O_3$, respectively.

[Figure]

Figure 4: Percentage variation in predicted concentrations of ML models for weighted PM$_{2.5}$ obtained from CAMS emissions, by sector, pollutant, and perturbation for selected countries: 80-20, 75-25 (initial splitting), and 70-30 train-test splitting. 12

[Figure]

Figure 5: Percentage variation in predicted concentrations of ML models for O₃ obtained from CAMS emissions, by sector, pollutant, and perturbation for selected countries: 80-20, 75-25 (initial splitting), and 70-30 train-test splitting.

**Reviewer 1 Point 21** — *Page 12, line 2: Why not to say "Emission scenarios" instead of "Stylized scenarios" in section 4.1 title?*

**Reply**: We welcome the Review's suggestion to simplify the Subsection 4.1. title to "Emission scenarios".

**Reviewer 1 Point 22** — *Page 12, line 5: Why are emission perturbations ranging to +60%, when we would expect policy scenarios to necessarily seek to reduce precursor emissions?*

**Reply**: We thank the Reviewer for asking about this point. We have added the following paragraph under Subsection 4.1. "Model results — Stylized scenarios":

"While policies generally aim to reduce emissions, including emission-increase scenarios is crucial for a comprehensive understanding of potential air quality outcomes of a wide range of possible future conditions. For example, the persistent investments in coal in India, or the investment in gas fracking. It is important to showcase that these policy interventions may lead to exposure increases. For these reasons, we have looked at decreases and increases in emissions."

**Reviewer 1 Point 23** — *Page 13, line 4: Move Figure 6 in the main text. The panels of this figure are very small, and it is very difficult to read the figure correctly. It would be preferable to prepare one figure per model and per pollutant to be mor readable.*

**Reply**: We agree on moving Figure 6 to the main text. We also welcome the suggestion of plotting one figure per pollutant and model, increasing their size to improve readability (see Figures 6 and 7). We have also corrected the x-axis plot related to EN models for predicting $PM_{2.5}$: the subplot was mistakenly displaying the range -50 to +50 instead of -60 to +60.

[Figure]

Figure 6: Percentage variation in predicted concentrations by sector and perturbation for selected countries in EN and ML models for weighted PM$_{2.5}$. Bar charts on the sides of each subplot help visualize overlapping variations.

[Figure]

Figure 7: Percentage variation in predicted concentrations by sector and perturbation for selected countries in EN and ML models for O$_3$. Bar charts on the sides of each subplot help visualize overlapping variations.

**Reviewer 1 Point 24** — *Page 13, line 26: Move figures 7 and 8 to the main text.*

**Reply**: We welcome this suggestion and have changed the manuscript accordingly.

**Reviewer 1 Point 25** — *Page 13, line 28: Could you explain why models work better for O3 than for PM2.5?*

**Reply**: The authors thank the Reviewer for allowing this point to be discussed. We have updated the text: "Both elastic net and machine learning models are generally better at predicting O$_3$ than PM$_{2.5}$ as

the former is highly correlated with incoming radiation or temperature, while the secondary chemistry of $PM_{2.5}$ is harder to grasp. Chemistry transport models predict better $O_3$ than PM as well, due to the more complex mixture of particles and local effects from more sources of the latter (Guérette et al., 2020)."

**Reviewer 1 Point 26** — *Page 13, line 42: The following clarification should be made: DACCIWA is preferred to CAMS over Africa only. (same page 14, lines 1 and 2).*

**Reply**: We thank the Reviewer for this useful comment, on which we fully agree. We have specified in both lines that we refer to Africa only.

**Reviewer 1 Point 27** — *Page 13, lines 3 and 4: Move figures 9 and 10 in the main text.*

**Reply**: We welcome this suggestion and move the figures accordingly.

**Reviewer 1 Point 28** — *Page 14, line 10: What you mean by "measurement error of unknown distribution"? Could you please clarify.*

**Reply**: When harmonizing grids with different spatial resolutions, the interpolation process itself can introduce errors. Such errors can arise because data values are estimated in locations that were not originally measured in the raw data, therefore we have no means of knowing the type of distribution of these errors. We have clarified as follows: "First, results are highly dependent on input data availability and quality. Using gridded data at different resolutions requires harmonization on a common global grid. As data values are estimated in locations that were not originally measured in the raw data, the interpolation process can introduce measurement errors of unknown distribution."

**Reviewer 1 Point 29** — *Page 14, line 39: The evolution of the approach by the consideration of an ensemble of the models is a quite good perspective of work to improve the final estimate.*

**Reply**: We thank the Reviewer for this comment.

Moreover, we have integrated the acknowledgement of financial support as follows:

"This work was supported by the World Bank. This project has received funding from the European Union's Horizon Europe research and innovation programme under grant agreement No 101069880 - AdJUST, and from the European Union - Next Generation EU, in the framework of the project GRINS - Growing Resilient, INclusive and Sustainable project (GRINS PE00000018 – CUP C93C22005270001)."

**References**

Amann, M., Bertok, I., Borken-Kleefeld, J., Cofala, J., Heyes, C., Höglund-Isaksson, L., Klimont, Z., Nguyen, B., Posch, M., Rafaj, P., Sandler, R., Schöpp, W., Wagner, F., Winiwarter, W., 2011. Cost-effective control of air quality and greenhouse gases in europe: Modeling and policy applications. Environmental Modelling & Software 26, 1489–1501. doi:10.1016/j.envsoft.2011.07.012.

Colette, A., Rouïl, L., Meleux, F., Lemaire, V., Raux, B., 2022. Air Control Toolbox (ACT_v1.0): a flexible surrogate model to explore mitigation scenarios in air quality forecasts. Geoscientific Model Development 15, 1441–1465. doi:10.5194/gmd-15-1441-2022.

Ding, D., Xing, J., Wang, S., Dong, Z., Zhang, F., Liu, S., Hao, J., 2021. Optimization of a nox and voc cooperative control strategy based on clean air benefits. Environmental Science amp; Technology 56, 739–749. doi:10.1021/acs.est.1c04201.

Dingenen, R.V., Dentener, F., Crippa, M., Leitao, J., Marmer, E., Rao, S., Solazzo, E., Valentini, L., 2018. TM5-FASST: a global atmospheric source–receptor model for rapid impact analysis of emission changes on air quality and short-lived climate pollutants. Atmospheric Chemistry and Physics 18, 16173–16211. doi:10.5194/acp-18-16173-2018.

Granella, F., Renna, S., Aleluia Reis, L., 2024. The formation of secondary inorganic aerosols: A data-driven investigation of Lombardy's secondary inorganic aerosol problem. Atmospheric Environment 327, 120480. doi:10.1016/j.atmosenv.2024.120480.

Granier, C., Darras, S., Denier van der Gon, H., Doubalova, J., Elguindi, N., Galle, B., Gauss, M., Guevara, M., Jalkanen, J.P., Kuenen, J., Liousse, C., Quack, B., Simpson, D., Sindelarova, K., 2019. The copernicus atmosphere monitoring service global and regional emissions (april 2019 version) doi:10.24380/D0BN-KX16.

Granier, C., Denier van der Gon, H., Gauss, M., Arellano, S., Darras, S., Dellaert, S., Guevara, M., Jalkanen, J.P., Kuenen, J., Liousse, C., Markova, J., Quack, B., Simpson, D., Sindelarova, K., Soulie, A., 2021. The copernicus atmosphere monitoring service global and regional emissions (november 2021 version) URL: https://aerocom-classic.met.no/DATA/download/for_cams81/CAMS81_2020SC1_D81.5.2.5_202111_Documentation_v1.pdf.

Guérette, E.A., Chang, L.T.C., Cope, M.E., Duc, H.N., Emmerson, K.M., Monk, K., Rayner, P.J., Scorgie, Y., Silver, J.D., Simmons, J., Trieu, T., Utembe, S.R., Zhang, Y., Paton-Walsh, C., 2020. Evaluation of Regional Air Quality Models over Sydney, Australia: Part 2, Comparison of PM2.5 and Ozone. Atmosphere 11, 233. doi:10.3390/atmos11030233.

Kiesewetter, G., Borken-Kleefeld, J., Schöpp, W., Heyes, C., Thunis, P., Bessagnet, B., Terrenoire, E., Fagerli, H., Nyiri, A., Amann, M., 2015. Modelling street level PM10

concentrations across europe: source apportionment and possible futures. Atmospheric Chemistry and Physics 15, 1539–1553. doi:`10.5194/acp-15-1539-2015`.

Krey, V., Masera, O., Blanford, G., Bruckner, T., Cooke, R., Fisher-Vanden, K., Haberl, H., Hertwich, E., Kriegler, E., Mueller, D., Paltsev, S., Price, L., Schlömer, S., Ürge Vorsatz, D., van Vuuren, D., Zwickel, T., 2014. Annex ii: Metrics & methodology, in: Edenhofer, O., Pichs-Madruga, R., Sokona, Y., Farahani, E., Kadner, S., Seyboth, K., Adler, A., Baum, I., Brunner, S., Eickemeier, P., Kriemann, B., Savolainen, J., Schlömer, S., von Stechow, C., Zwickel, T., Minx, J. (Eds.), Climate Change 2014: Mitigation of Climate Change. Contribution of Working Group III to the Fifth Assessment Report of the Intergovernmental Panel on Climate Change. Cambridge University Press, Cambridge, United Kingdom and New York, NY, USA, pp. 1281–1328.

OpenAQ, 2024. OpenAQ. URL: `https://openaq.org/#/`.

Thunis, P., Clappier, A., Tarrason, L., Cuvelier, C., Monteiro, A., Pisoni, E., Wesseling, J., Belis, C., Pirovano, G., Janssen, S., Guerreiro, C., Peduzzi, E., 2019. Source apportionment to support air quality planning: Strengths and weaknesses of existing approaches. Environment International 130, 104825. doi:`10.1016/j.envint.2019.05.019`.

Thunis, P., Degraeuwe, B., Pisoni, E., Ferrari, F., Clappier, A., 2016. On the design and assessment of regional air quality plans: The SHERPA approach. Journal of Environmental Management 183, 952–958. doi:`10.1016/j.jenvman.2016.09.049`.

---

## Author Comment (AC2)

**Authors' response to Reviewer 2**

We sincerely thank the Reviewer for taking the time to thoroughly review our manuscript and provide relevant feedback to improve it further. Below we address their concerns and suggestions point by point.
* * *
**Reviewer 2**

**General**

*The authors developed the tool CLAQC to quickly assess the impact of policy scenarios on Air Quality using 2 methods: elastic net modelling (EN) and an extreme gradient boosting regressor (ML). CLAQC can be used to attribute sectoral and country specific emissions changes to changes in $PM_{2.5}$ and $O_3$ concentrations without great computation burden. It is a useful too for policy makers and other stakeholders. The authors evaluate the performance of both models on a country level and find that the model performance differs depending on country and model used, while generally both models are better at predicting $O_3$ than $PM_{2.5}$. The paper is excellently written, the language is easy to understand and the paper well structured. The figures were unfortunately of low resolution and should be improved for publication. In several sections a more detailed explanation or discussion on top of the description of results would be useful.*

**Comments:**

(Abbreviations used: PXX-LYY − > page xx, line YY; EQZ − > Equation Z)

**Reviewer 2 Point 1**  —  *P2-L28f Did you mean "secondary $O_3$ formation and secondary PM formation"?*

**Reply**:  Yes, we meant "secondary $O_3$ formation and secondary PM formation". Thank you very much for noticing this. We have updated the text as suggested.

**Reviewer 2 Point 2**  —  *P2-L40ff The 2 sentences starting with "As new data" and "As new and better data" seem repetitive. Please consolidate these sentences.*

**Reply**:  Thank you for this suggestion. We have changed the text as follows: "As new and better data come in every year, the emulator can be updated, and a higher detail level may be possible at lower trade-off costs".

**Reviewer 2 Point 3**  —  *P5-L7 Why did you choose TerraClimate over ERA5 for the majority of variables used? Is TerraClimate's rain product more accurate than the one from ERA5? Since*

*you're aggregating all data to 0.5° x 0.5°, you don't seem to make use of TerraClimate's higher horizontal resolution.*

**Reply**: It is true that the ERA5 dataset is also a very complete product. Nevertheless, the TerraClimate product is a peer-reviewed dataset published in Scientific Data. It has a spatial resolution of about 4 km x 4 km and a monthly temporal resolution and was validated against weather measurement station data from several different networks, e.g., the Global Historical Climatology Network (GHCN) database, the Snow Telemetry network (SNOTEL), and the RAWS USA Climate Archive (Abatzoglou et al., 2018). Indeed, we could have used other reanalysis products such as the ERA5 ones for the weather variables included in the models. However, TerraClimate is both spatially and temporally suitable for our purpose and allows us to keep flexibility for further improvements in spatial and temporal resolution.

**Reviewer 2 Point 4** — *P5-L23ff You are describing how a reanalysis again here when you had already described it in greater detail in the introduction to section 2 (P3-L4ff). This seems repetitive.*

**Reply**: We thank the Reviewer for this suggestion. We have removed the first sentence, "EAC4 reanalysis combines model data with observations, *in-situ* and satellite, from all over the world into a globally complete and consistent data set using a model of the atmosphere based on the laws of physics and chemistry", and moved the remaining part of the paragraph to the introductory part of Section 2.

**Reviewer 2 Point 5** — *P7-L21 Please elaborate on the general purpose of monotonic constraints for the benefit of the reader not too familiar with that kind of modelling.*

**Reply**: We thank the Reviewer for this suggestion: we have briefly introduced the general purpose of monotonic constraints, explaining their purpose and providing an example. We have changed the text as follows: "We impose monotonic constraints on certain model coefficients to align with expected physicochemical relationships. These constraints specify how input variables should affect the target, ensuring interpretable and physically plausible results. For instance, a positive monotonic constraint enforces a non-negative relationship, ensuring that as an input variable increases, the predictor output does not decrease.

In the presence of noise, complex interactions in the data, or predictor cross-correlation, models may otherwise learn patterns that are not realistic or physically plausible. Additionally, monotonic constraints help prevent overfitting, enhancing robustness when input data are limited or uncertain. For example, it is not expected that an increase in BC emissions would lead to a decrease in $PM_{2.5}$ concentrations.

While at the local scale, reducing certain precursors of secondary inorganic aerosols might not always lead to a decrease in $PM_{2.5}$ levels — due to nonlinear atmospheric reactions noted by Thunis et al. (2019); Ding et al. (2021) — our national-scale models focus on broader trends. To avoid giving undue importance to cases where local emissions reductions might result in increased levels of inorganic $PM_{2.5}$, we apply monotonic constraints. In scenarios where secondary reactions substantially affect the overall

mass of $PM_{2.5}$, our models are designed to exclude such precursors from the predictor list, thereby not reflecting a decrease in $PM_{2.5}$ levels."

**Reviewer 2 Point 6** — *EQ5 Please define $\beta$ and $\beta_0$.*

**Reply**: We have included the definition of $\beta$ and $\beta_0$ as follows: "It solves the following minimization problem for the model parameters $\beta_0$ and $\beta$, where $\beta_0$ is the model's intercept and $\beta$ represents the coefficients of the input variables:"

**Reviewer 2 Point 7** — *EQ7 Should this be $n_{test}$ instead of $n^{test}$? In P9-L24 test is a sub- not a superscript.*

**Reply**: Thank you for pointing out this inconsistency. The superscripts in EQ6&7 are used as a reference to the test set. Regarding the $n$ constant, we have updated it in EQ7 to $n_{test}$, as in P9-L24.

**Reviewer 2 Point 8** — *EQ8&9 Several parameters are not defined, e.g. $\lambda$, $\beta$, $\gamma_i$, $\delta$, $\mu$, $\nu$, $\epsilon$, $\theta$. Are the $\alpha$ and $\beta$ the same as in EQ5?*

**Reply**: We thank the Reviewer for pointing out this. The mentioned parameters ($\lambda$, $\beta$, $\gamma_i$, $\delta$, $\mu$, $\nu$, $\epsilon$, $\theta$) are the predictors' coefficients. The $\beta$ parameters in EQ5, 8, and 9 have theoretically the same meaning. Instead, the $\alpha$ parameter in EQ5 is a regularization parameter that sets up the elastic net setting while in EQ8 and 9 it represents the linear regression intercept. In EQ5, $\alpha$ corresponds to a value of 0.5. If the $\alpha$ parameter was 1, then a LASSO shrinkage regression would be performed. If the $\alpha$ parameter was 0, a Ridge regression. We have added these explanations in the text.

**Reviewer 2 Point 9** — *EQ8&9 In both equations emissions are used multiple times: with emissions depending on sector & pollutant ($\beta$-term), just sector ($\delta$-term) and just pollutant ($\lambda$-term). Please clarify what the purpose of the multiple emission terms is. In the P11-L7ff you describe how the terms in EQ8&9 mimic secondary production, transport and dispersion. It would be useful for the reader to also understand what processes or dependencies the multiple emission terms are a proxy for.*

**Reply**: We thank the Reviewer for this comment. To better clarify, we have updated the text as follows: "In both Equations, we include multiple emission terms to increase the chances that models capture variations in emissions. In Equation 8, to model the formation of secondary inorganic aerosol, we interact emissions of $NO_x$ and $NH_3$, $SO_2$ and $NH_3$, and $NO_x$ and $SO_3$, respectively. This approach helps to represent the formation of secondary inorganic aerosols, such as ammonium salts, which result from reactions between these precursors. Similarly, in equation 9, we interact emissions of $NO_x$ and NMVOC, $SO_2$ and NMVOC, and $SO_2$ and $NO_x$. As before, this attempts to capture the reactions between the precursors of $O_3$, since the presence of at least two of these precursors is necessary for its

formation. In both Equations, we also interact sectoral emissions with wind speed and direction to proxy transport and dispersion of pollutants. We include total sectoral emissions to reflect that sector-specific policies typically impact multiple pollutants through dedicated emission offset protocols. Additionally, we consider total emissions from individual pollutants because variations in total pollutant emissions may result not only from specific sectors but also from inter-sector changes, transported emissions, and chemical reactions."

**Reviewer 2 Point 10** — *EQ9 The emission terms for the $O_3$ equations are slightly different to the ones for $PM_{2.5}$. For both pollutants, there are terms depending on sector & pollutant at the same time ($\beta$-term) and then just on the sector ($\delta$-term). In EQ9, term depending on just the pollutant ($\lambda$) is for a different pollutant ($p_3$) than the $\beta$-term. Why do the $\beta$- and $\lambda$-term for $PM_{2.5}$ (EQ8) depend on the same pollutants, but the $\beta$- and $\lambda$-terms for $O_3$ (EQ9) do not: namely the $\lambda$-term depends on one additional pollutant ($SO_2$).*

**Reply**:   Correct, the emission terms for the $O_3$ equations are slightly different with respect to the ones for $PM_{2.5}$ and the $\lambda$-term depends on one additional pollutant ($SO_2$), due to their different atmospheric reactions. Regarding emissions, the $O_3$ model includes sectoral emissions related to NMVOC and $NO_x$, which are the main precursors of $O_3$ (Baird and Cann, 2013; John H. Seinfeld, 2016). It also includes sector totals, total emissions of NMVOC, $NO_x$, and $SO_2$, and their interactions (specifically, $NO_x \times$ NMVOC, $SO_2 \times$ NMVOC, and $SO_2 \times NO_x$). Plus, interactions between sectoral emissions and wind speed and direction. While NMVOC and $NO_x$ are $O_3$ main precursors, reacting in the presence of solar radiation, $SO_2$ plays an indirect role in $O_3$ formation. $SO_2$ is typically emitted by industrial sources. It is involved in secondary PM formation, which can reduce the radiative properties and oxidative capacity of the atmosphere, indirectly affecting $O_3$ formation. We have tried to clarify this further in the text under Subsection 3.3. Elastic net models:

"Note that the emission terms in the Equations differ due to their different atmospheric reactions. In Equation **??**, to model the secondary inorganic aerosol formation, we interact total emissions of $NO_x$ and $NH_3$, $SO_2$ and $NH_3$, and $NO_x$ and $SO_2$, respectively. Similarly, in equation **??**, we interact total emissions of $NO_x$ and NMVOC, $SO_2$ and NMVOC, and $SO_2$ and $NO_x$. While NMVOC and $NO_x$ are $O_3$ main precursors, reacting in the presence of solar radiation, $SO_2$ plays an indirect role in $O_3$ formation (Baird and Cann, 2013; John H. Seinfeld, 2016). $SO_2$ is typically emitted by industrial sources. It is involved in secondary PM formation, which can reduce the radiative properties and oxidative capacity of the atmosphere, indirectly affecting $O_3$ formation. In both Equations, we also interact sectoral emissions with wind speed and direction to proxy transport and dispersion of pollutants. Refer to section **??** for the EN model specifications with DACCIWA emissions."

**Reviewer 2 Point 11** — *P11-L29 You say that randomisation occurs over the temporal dimension. Does that mean that the concentration fields calculated by the ML model do not depend on the*

*previous time step (month in this case)? Is there an initialisation of the pollutant concentrations or is the assumption essentially that the ML model can estimate the pollutant concentration of the current month based on the emissions and meteorological conditions of the current month only, without knowledge of previous atmospheric conditions and pollutant concentrations?*

**Reply**: We apologize for not being clear on this point. Your interpretation is correct. Regarding ML estimation through the XGBoost architecture, it takes place without initialisation and without imposing any functional form, except for monotonic constraints. The model estimates the pollutant concentration for the current month based on emissions, weather, and other conditions of that same month (pixel identifier and seasonal identifiers), without knowledge of previous atmospheric conditions. The model captures interactions between features as well, without explicitly defining them.

**Reviewer 2 Point 12** — *P13-L12f The road and residential sectors are named as having the greatest impact in DEU, ITA and BRA. Is that referring to Fig 6a? I cannot see that in the figure. Italy seems to only have impact from Agriculture. The resolution of the plot is quite low so it's hard to see details.*

**Reply**: Correct, this statement is referring to Figure 6a. Unfortunately, due to the size constraints of the uploaded preprint document, the plots' resolution is low, sometimes not allowing to accurately see plot details. Though, all our plots have a resolution of 300 dpi. In Figure 6, both ranges and line plots are displayed. Given that the model algorithms may include multiple emission variables within a sector, e.g., both $NO_x$ and BC emissions from the Agriculture sector, to account for the sectoral range variability we calculate the minimum and maximum annual percentage variation in predictions from perturbed emissions by perturbation and sectoral level. When the model chooses only one sectoral predictor, the minimum and maximum annual percentage variation in predictions from perturbed emissions is the same, and a simple line is displayed in the plot, as in the case of the Road and Residential sectors for DEU, ITA, and BRA.

[Figure]

Figure 6: Percentage variation in predicted concentrations by sector and perturbation for selected countries in EN models for PM$_{2.5}$. Bar charts on the sides of each subplot help visualize overlapping variations.

**Reviewer 2 Point 13** — *P13-L28ff Please discuss why the models perform so poorly in some countries. Is it inconsistencies in either emission or concentration data for that country? Are there important mechanisms occurring in this countries that are missed by the models? Are there pollutants missing from the emission data sets that are important in those countries? Does the model perform poorly because of some of the inputs or is it something in the model that you could change to improve the performance?*

**Reply**: It is true that results vary by prediction target (PM$_{2.5}$ vs O$_3$) and input type (e.g., CAMS vs DACCIWA emissions), which may reflect the inconsistencies in emission data mentioned. Generally, both modeling methods are better at predicting O$_3$, as O$_3$ concentrations are highly correlated with incoming radiation or temperature, while predicting PM$_{2.5}$ is more challenging due to its complex secondary chemistry, local sources and particle composition. Chemistry transport models predict better O$_3$ than PM as well, due to the more complex mixture of particles and local effects from more sources of the latter one (Guérette et al., 2020). Many of the potential issues you noted can contribute to country-level performance variations. Additionally, local factors such as unique orography and micro-meteorological conditions can significantly impact predictions in some areas, even country-level averages. This is suggested by the stronger performance of ML models incorporating pixel identifiers in Figure 10, highlighting the importance of local features in achieving better performance. Although we did not dive into the reasons for some countries performing better than others, Figures 7 to 9 can help us identify some sources of poor performance. For instance, results indicate that Europe and some African countries, plus some countries in Central and South America perform better with ML models, implying that their

variability is better explained by nonlinear relationships, since their $R^2$ is higher in ML models (Figure 9). However, we also observe an increase in error in some of these models. While EN models seem to outperform ML ones in North America for both pollutants.

In response to your suggestion, we have extended the discussion in subsection 4.2. Model internal validation results, as below:

"Figures 7 and 8 map the out-of-sample $R^2$ and RMSE for EN and ML models obtained from both CAMS and DACCIWA emissions. We do not advise using the models for countries with $R^2$ smaller than 0.5 or RMSE higher than 12.

Results vary by prediction target ($PM_{2.5}$ vs $O_3$) and input type (e.g., CAMS vs DACCIWA emissions), which may reflect inconsistencies in the emission or concentration data. Additionally, local factors such as unique orography and micro-meteorological conditions can significantly impact predictions in some areas, even country-level averages. Generally, both modeling methods are better at predicting $O_3$ than $PM_{2.5}$, as $O_3$ concentrations are highly correlated with incoming radiation or temperature, while predicting $PM_{2.5}$ is more challenging due to its complex secondary chemistry, local sources and particle composition. Chemistry transport models predict better $O_3$ than PM as well, due to the more complex mixture of particles and local effects from more sources of the latter one (Guérette et al., 2020)".

We have also added to the 5. "Limitations" Section the fact of not having analyzed the reasons behind countries' poor performance.

**Reviewer 2 Point 14** — *P14-L1 In P13-L43f it sounds like DACCIWA is used everywhere where available, so in Africa CAMS is never used, correct? In fig 9c then, are the runs using DACCIWA actually being compared with runs using CAMS in Africa?*

**Reply**: We thank the Reviewer for this comment and apologize for not being clear on this aspect. We clarify a bit here and in the text. While CAMS-GLOB-ANT emissions are available for all countries worldwide, DACCIWA emissions are only available for Africa. Note that we construct and provide independent modeling versions with both CAMS-GLOB-ANT and DACCIWA emissions for Africa. In Figure 9, regarding the performance metrics RMSE and $R^2$, runs using DACCIWA are compared with runs using CAMS in Africa in subplots 9a and 9b, respectively. While, as the Reviewer correctly pointed out, in subplot 9c, we underline that we recommend DACCIWA over CAMS-GLOB-ANT as the preferred input source for emissions in Africa. Therefore, models built with DACCIWA emissions for African countries maximize the *Source* criterion, as shown in 9c. To clarify further, we discuss the case of South Africa. In figure 9, regarding $PM_{2.5}$, $R^2$ is maximized for South Africa with ML runs (pixel version) on DACCIWA emissions (9a), and regarding RMSE the model and input version that minimizes RMSE is elastic net with DACCIWA emissions (9b), while in terms of source DACCIWA is preferred (9c). Regarding $O_3$ instead, the EN run with CAMS-GLOB-ANT maximizes $R^2$ (9a), and the EN run with DACCIWA minimizes RMSE (9b), while DACCIWA is the preferred source (9c). We have improved clarity on this in the text.

**Reviewer 2 Point 15** — *P14-L15 Is the CAMS reanalysis you mention here the EAC4 reanalysis*

*you introduced in 2.3? If yes, it is confusing for the reader to refer to the same product with different names. If no, please introduce the CAMS reanalysis.*

**Reply**: With "CAMS reanalysis" we meant both CAMS emission and concentration reanalysis products employed for CLAQC, so the data described under Subsections 2.1.2. CAMS emissions (CAMS GLOB-ANT reanalysis) and 2.3. Concentrations (EAC4 reanalysis), respectively. Although the EAC4 reanalysis product is operated by ECMWF, it is part of the services offered by the Copernicus Atmosphere Monitoring Service (CAMS). Therefore, it is also known as the CAMS global reanalysis EAC4 product. As this may create confusion, we have tried to make this clearer in the text, as follows: "However, given the high disparities in the available ground monitoring data across the globe, we believe that CAMS reanalysis products, such as CAMS-GLOB-ANT and EAC4, are the next state-of-the-art available solution for these regions".

**Reviewer 2 Point 16** — *P14-L36 The sentence starting with "It is a complimentary model" is confusing:*
- *"A [. . . ] model to the model [. . . ] community" is repetitive. Maybe "A complimentary tool"?*
- *Which scenario community? The policy scenario community?*
- *Did you mean "providing empirically based estimates"?*

**Reply**: We thank the Reviewer for these suggestions. We have updated the text as follows: "The CLAQC framework lends itself to multiple developments. It is a complementary tool to the modeling and policy scenario community, providing empirically based estimates and added value for global scale sectoral and country-level analyses."

**Reviewer 2 Point 17** — *P14-L39 Unless there is a second paper planned describing the CLAQC tool's functionality, it would be useful to have a short overview over the kind of scenarios that can be run. I.e. is the 60% perturbation fixed or can the user have some control over the scenario selection (apart from country, model, specification, etc used).*

**Reply**: We thank the Reviewer for this comment. Regarding the scenarios, we simulate perturbations of emissions from -60% to +60% at 20% steps based on the last 5 years of data (2017-2021). This means that emission perturbations are fixed to those levels. However, the user has control over the scenarios as they can select the following parameters: country, model, specification, sector, precursor pollutant, baseline concentration, baseline emissions, and perturbation level. We discussed scenarios under Subsection "Comparing two scenarios" in the Appendix Subsection A.4.2. Machine learning — Guide to the Excel spreadsheet.

**Reviewer 2 Point 18** — *P14-L44 Link is broken. Is the code embargoed until the paper is published?*

**Reply**: We had reserved a DOI (10.17632/wt25vt6ycr.1) through a Mendeley Data repository but unfortunately we are having problems accessing it again to make it public. This might be due to some changes in the Mendeley services that have occured recently. We have set up a new frozen repository with the original code through Zenodo at the following URL: `https://zenodo.org/records/14177055`.

**Figures**

**Reviewer 2 Point 19** — *Fig 1 Please include more labels for the colour scale in b).*

**Reply**: We thank the Reviewer for this suggestion that we have implemented as in Figure 1. Note that concentration levels are displayed on a logarithmic scale as before.

[Figure]

(a) PM$_{2.5}$

PM$_{2.5}$

(b) O$_3$

O$_3$

Figure 1: Level plots of EAC4 concentrations of PM$_{2.5}$ (January 2018) and O$_3$ (July 2018) in $\mu g/m^3$ with color bar in logarithmic scale.

**Reviewer 2 Point 20** — *Fig 6a In the plots for BRA, NGA, SAU and TUR there are line plots instead of filled areas for some of the sectors. Is that a plotting error or does that signify something?*

**Reply**: Line plots in Figure 6 are not a plotting error. Given that the model algorithms may include multiple emission variables within a sector, e.g., both NO$_x$ and BC emissions from the Road sector, to

account for the sectoral range variability we calculate the minimum and maximum annual percentage variation in predictions from perturbed emissions by perturbation and sectoral level. When the model chooses only one sectoral predictor, the minimum and maximum annual percentage variation in predictions from perturbed emissions is the same, and a simple line is displayed in the plot.

**Reviewer 2 Point 21** — *Fig 7 It is difficult to see which countries are below 0.5 with a continuous colour scale. Maybe include a colour break at 0.5?*

**Reply**: We thank the Reviewer for this suggestion. We have included a colour break at 0.5 in the plots as proposed. See below Figure 7.

[Figure]

Figure 7: Out-of-sample performance metrics of ML (no pixel) and EN models (both from CAMS and DACCIWA data): $R^2$.

**Reviewer 2 Point 22** — *Fig 8 Similar to Fig 7, it is not possible to see the cut-off of 12 with the colour scale used.*

**Reply**: We thank the Reviewer for this suggestion: we have updated the plots accordingly. See below Figure 8.

[Figure]

Figure 8: Out-of-sample performance metrics of ML and EN models (both from CAMS and DACCIWA data): RMSE.

**Reviewer 2 Point 23** — *Fig 9 "with" should be abbreviated with "w" or "w/" not "w/t".*

**Reply**: Thank you for highlighting this imprecision. We have updated the plot legend as suggested. See Figure 9.

[Figure]

Figure 9: Best model score for each pollutant, country, and decision criterion.

**Reviewer 2 Point 24** — *Fig 9 To make the best performing model "group" (EN vs ML) more obvious, you could use one hue per model group, i.e. all EN models in shades of blue and all ML models in shades of red.*

**Reply**: We thank the Reviewer for this suggestion that we have considered for improving the plot's readability. See Figure 9.

**Reviewer 2 Point 25** — *There are some small inconsistencies in notation the authors may want to address, e.g.*

*P7-L24ff PM2.5 not subscripted for some occurrences in this paragraph.*

*P2-L31 $O_3$ is cursive here but nowhere else.*

*L39 Remove gap between T and g for Tg.*

*P10 in description for TMINt and TMAXt, use °C, not degC just as elsewhere in the text*

**Reply**: We thank the Reviewer for noticing these inconsistencies. We have corrected the $PM_{2.5}$ notation and the cursive $O_3$. Regarding the unit of measure on P3-L39, there is no gap between T and g for $Tg$: it may appear as a gap just due to the cursive font. We have harmonized the abbreviation for degrees Celsius to °C throughout the text.

**References**

Abatzoglou, J.T., Dobrowski, S.Z., Parks, S.A., Hegewisch, K.C., 2018. TerraClimate, a high-resolution global dataset of monthly climate and climatic water balance from 1958-2015. Scientific Data 5, 170191. URL: https://www.nature.com/articles/sdata2017191, doi:10.1038/sdata.2017.191.

Baird, C., Cann, M., 2013. Chimica ambientale. Terza edizione italiana condotta sulla quinta edizione americana.

Ding, D., Xing, J., Wang, S., Dong, Z., Zhang, F., Liu, S., Hao, J., 2021. Optimization of a nox and voc cooperative control strategy based on clean air benefits. Environmental Science amp; Technology 56, 739–749. doi:10.1021/acs.est.1c04201.

Guérette, E.A., Chang, L.T.C., Cope, M.E., Duc, H.N., Emmerson, K.M., Monk, K., Rayner, P.J., Scorgie, Y., Silver, J.D., Simmons, J., Trieu, T., Utembe, S.R., Zhang, Y., Paton-Walsh, C., 2020. Evaluation of Regional Air Quality Models over Sydney, Australia: Part 2, Comparison of PM2.5 and Ozone. Atmosphere 11, 233. doi:10.3390/atmos11030233.

John H. Seinfeld, S.N.P., 2016. Atmospheric Chemistry and Physics: From Air Pollution to Climate Change. WILEY. URL: https://www.ebook.de/de/product/25599491/john_h_seinfeld_spyros_n_pandis_atmospheric_chemistry_and_physics_from_air_pollution_to_climate_change.html.

Thunis, P., Clappier, A., Tarrason, L., Cuvelier, C., Monteiro, A., Pisoni, E., Wesseling, J., Belis, C., Pirovano, G., Janssen, S., Guerreiro, C., Peduzzi, E., 2019. Source apportionment to support air quality planning: Strengths and weaknesses of existing approaches. Environment International 130, 104825. doi:10.1016/j.envint.2019.05.019.